# BiMoE: Pushing the Limit of Post-Training Quantization for MoE-based LLMs

## Abstract

Large language models (LLMs) with Mixture-of-Experts (MoE) architectures have achieved remarkable progress in natural language processing, yet their massive memory and compute costs hinder practical deployment. Binarization, which compresses model weights to 1 bit, yields extreme efficiency, offers an extreme efficiency advantage. However, existing methods that primarily target dense LLMs are not well suited to address MoE-specific quantization challenges, including redundant expert representations, task-unaware weight-importance scoring, and quantization-induced expert-shift. To this end, we propose BiMoE, the first binarization framework tailored for MoE-based LLMs. BiMoE is built on three core innovations: 1) using joint SVD decomposition to reduce cross-expert redundancy; 2) integrating global loss gradients into local Hessian metrics to enhance weight importance estimation; 3) introducing an error constraint guided by the input null space to mitigate routing distortion. Notably, BiMoE achieves these optimizations while incurring no additional storage overhead, striking a balance between efficiency and model performance. Extensive experiments demonstrate that BiMoE consistently outperforms state-of-the-art binary methods across multiple MoE-based LLMs and benchmarks. For example, on Qwen3-30B-A3B, BiMoE reduces perplexity by 52.2%, improves average zero-shot performance by 43.4%, achieves over $2\times$ inference speedup, and further shortens quantization time. The code is available at https://anonymous.4open.science/r/BiMoE.

## 1 Introduction

Mixture-of-Experts (MoE) has emerged as a powerful architecture for scaling large language models (LLMs) (DeepSeek-AI et al., 2025; Yang et al., 2025; Muennighoff et al., 2025). MoE-based LLMs couple expert specialization with a learned router that selects only a few experts per token. Replacing monolithic feed-forward layers with a pool of smaller experts and activating them sparsely delivers higher compute efficiency than dense LLMs. However, deploying MoE-based LLMs introduces significant memory overhead since all experts must reside in memory, even though only a small fraction are active during inference. Take GPT-OSS-120B (OpenAI et al., 2025) as an example: while only 5 billion parameters are active at decoding, the full 120 billion parameters (more than 240GB in FP16) must remain in memory continuously, regardless of expert activation, leading to inflated inference memory overhead. This underscores the need to compress MoE-based LLMs to reduce inference costs and enable deployment on resource-constrained devices.

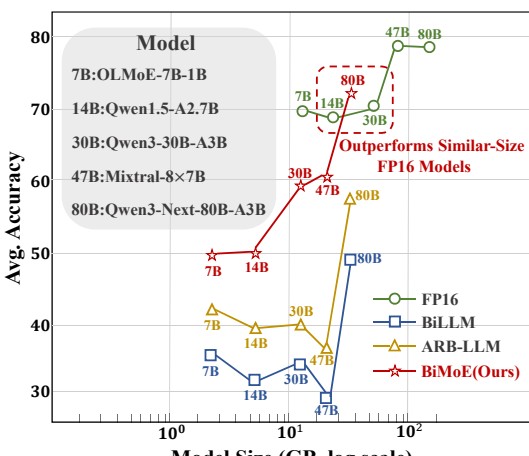

Figure 1: Five different MoE-based LLMs performance on 7 zero-shot datasets. Our BiMoE outperforms the similar-size FP16 models.

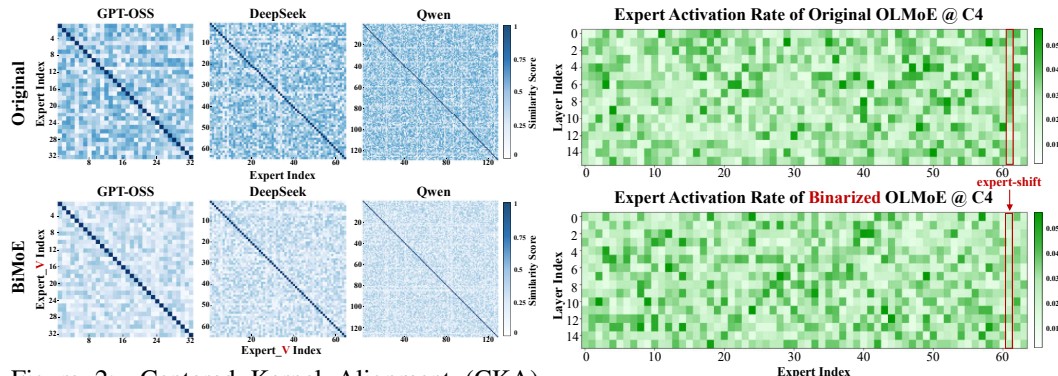

Figure 2: Centered Kernel Alignment (CKA) similarity of expert weights (**Top**: original weights. **Bottom**: expert-specific $V$ matrices after joint SVD in BiMoE) across GPT-OSS-20B, DeepSeek-V2-Lite, and Qwen3-30B-A3B. After applying our method, the off-diagonal CKA decreases, implying less cross-expert redundancy.

Figure 3: Token–expert mappings of OLMoE on C4 (**Top**: original model. **Bottom**: binarized model). Binarization leads to noticeable expert-shift, where token assignments migrate across experts compared to the original distribution, indicating routing mechanism degradation.

Quantization is a key compression technique, with binarization being particularly attractive for reducing weights to one bit. Recent works like BiLLM (Huang et al., 2024) and ARB-LLM (Li et al., 2024) compute the weight importance of task-unaware local Hessians and allocate more bits accordingly, the lack of task alignment often limits downstream performance. Further advancement, such as that in (Boža & Macko, 2025), represent weights as products of scaled binary matrices, outperforming single-matrix schemes. Although these methods are successful on dense LLMs, they transfer poorly to MoE-based LLMs: direct application causes marked degradation (Fig. 1). Consequently, binarization tailored to MoE-based LLMs remains unexplored.

In this paper, we investigate and identify three fundamental challenges in binarizing MoE-based LLMs: ❶ **Expert redundancy:** MoE-based LLMs typically manage tens to hundreds of experts, but existing methods treat them independently, ignoring cross-expert similarities (Fig. 2 (top)). This causes unnecessary storage redundancy in binarized MoE-based LLMs. ❷ **Task-unaware weight-importance scoring:** existing methods rely on local (per-layer) Hessian-based scores that are misaligned with the end-task loss, limiting downstream effectiveness. ❸ **Quantization-induced expert-shift:** binarization perturbations distort routing distributions, dispatching tokens to suboptimal experts (Fig. 3), which undermines generalization and amplifies quantization noise.

To this end, we propose **BiMoE**, the first customized binarization framework for MoE-based LLMs, which is built on three core innovations: ❶ **By using the joint decomposition of SVD**, BiMoE extracts a shared high-precision basis $U\Sigma$ and binarizes only the specific projection of experts $V$, reducing the redundancy of expert parameters to support subsequent binarization. ❷ **By aligning Hessian-based importance with downstream loss via global loss gradient injection**, it safeguards critical weights under binarization. ❸ **By confining noise to routing-insensitive null-spaces through lightweight row/column scaling (fused into binarization factors)**, it mitigates expert-shift while retaining 1-bit storage benefits. Across large-scale MoE-based LLMs, BiMoE consistently outperforms state-of-the-art binary PTQ baselines: for example, in Qwen3-30B-A3B it cuts perplexity by more than 50% and delivers more than $2\times$ faster inference, highlighting its effectiveness and practicality. Our key contributions can be summarized as follows:

- We present the first systematic study of binarizing MoE-based LLMs, identifying three distinct challenges: expert redundancy, limitations of task-unaware Hessian-based scores, and quantization-induced expert-shift.

- We introduce **BiMoE**, a tailored binarization framework that eliminates redundancy through joint SVD, aligns significance with global objectives via gradient-enhanced Hessians, and suppresses expert-shift with null-space constraints at minimal overhead.

- Extensive experiments show BiMoE outperforms state-of-the-art binary PTQ methods on MoE-based LLMs with superior efficiency. And we will release our code to advance research in this area.

## 2 RELATED WORK

**Mixture-of-Experts Large Language Models.** The Mixture-of-Experts (MoE) model was originally proposed by (Jacobs et al., 1991; Jordan & Jacobs, 1994) and has since been widely explored across domains (Deisenroth & Ng, 2015; Aljundi et al., 2017). In modern LLMs, each MoE layer comprises multiple experts and a gating network. The gate, typically a linear softmax layer, routes inputs to a few experts and aggregates their outputs. Designs vary to balance efficiency and performance: SwitchTransformer (Fedus et al., 2022) uses top-1 gating; Mixtral-8x7B (Jiang et al., 2024) activates two experts per layer; DeepSeekMoE (Dai et al., 2024) partitions FFN dimensions and adds shared experts to reduce redundancy; DeepSeek-v2 (DeepSeek-AI et al., 2024) and DeepSeek-v3 (DeepSeek-AI et al., 2025) further refine this design. Qwen-MoE (Team, 2024) replaces FFN layers with MoE, using 4 shared plus 4 selected from 60 experts. Kimi-K2 (Team et al., 2025) employs MuonClip to stabilize training and reaches strong performance with 32B active parameters. These advances highlight the need for compression tailored to the sparse activation of MoE, and while several methods for quantizing MoE (Li et al., 2025; Hu et al., 2025) have emerged, binarization remains unexplored.

**Binarization for Large Language Models.** Binarization is an extreme quantization technique that constrains weights and activations to binary values (e.g., -1/+1 or 0/1), reducing memory footprint and computation for deployment on resource-constrained devices. Applying binarization to LLMs is challenging due to accuracy sensitivity, especially in attention and embeddings. Prior work mitigates this by retaining selective high precision or partial binarization: BinaryBERT (Bai et al., 2021) binarizes BERT with selective retention; PB-LLM (Shang et al., 2023) keeps salient weights at higher precision. Recent methods leverage sparsity and importance (Dong et al., 2024) and use alternating fine-grained and column-group binarization (Li et al., 2024). Other work decomposes weights into products of binary matrices with scaling (Boža & Macko, 2025). Despite progress, these studies target dense LLMs. For MoE-based LLMs, related efforts such as mixed precision in MxMoE (Duanmu et al., 2025) address parameter scale rather than binarization. The unique routing and distributed parameters in MoE differ from dense models, making direct transfer difficult and underscoring the need for MoE-tailored binarization.

## 3 PRELIMINARIES

**Mixture-of-Experts.** MoE-based LLMs replace standard feed-forward (MLP) layers with MoE modules, each consisting of a router $G$ and $M$ experts $E = \{E_1, \ldots, E_M\}$. Given hidden state $x \in \mathbb{R}^d$, the output is

$$\text{MoE}(x) = \sum_{E_i \in S_{k,x}} G(x)_i \cdot E_i(x), \tag{1}$$

where $G(x)$ gives routing scores, and $S_{k,x}$ is the top-$k$ selected experts. The final output is a weighted sum of these experts.

**Binarization.** Binarization compresses full-precision (FP) weights $W \in \mathbb{R}^{n \times m}$ into 1-bit representations $\{\pm 1\}$ to reduce storage and accelerate inference. During the forward pass, binarized weights are obtained via the sign function:

$$B = \alpha \cdot \text{Sign}(W), \tag{2}$$

$$\text{sign}(w) = \begin{cases} +1, & w \geq 0, \\ -1, & \text{otherwise,} \end{cases} \tag{3}$$

where $B \in \mathbb{R}^{n \times m}$ is the binarized output. The scaling factor $\alpha = \frac{\|W\|_1}{m}$ preserves the magnitude of the original weights, with $n$ and $m$ denoting the dimensions of the weight matrix.

**Null Space.** For activations $X \in \mathbb{R}^{b \times d}$, the null space is $\{v : Xv = 0\}$, obtainable via SVD of $XX^\top$. Perturbations in this subspace vanish after multiplication with $X$. In quantization, if $\Delta W = W - W_q$ lies in the null space, then $X(W - W_q) \approx 0$, meaning its effect on outputs is suppressed. Thus, the null space defines natural *tolerant directions* for quantization errors.

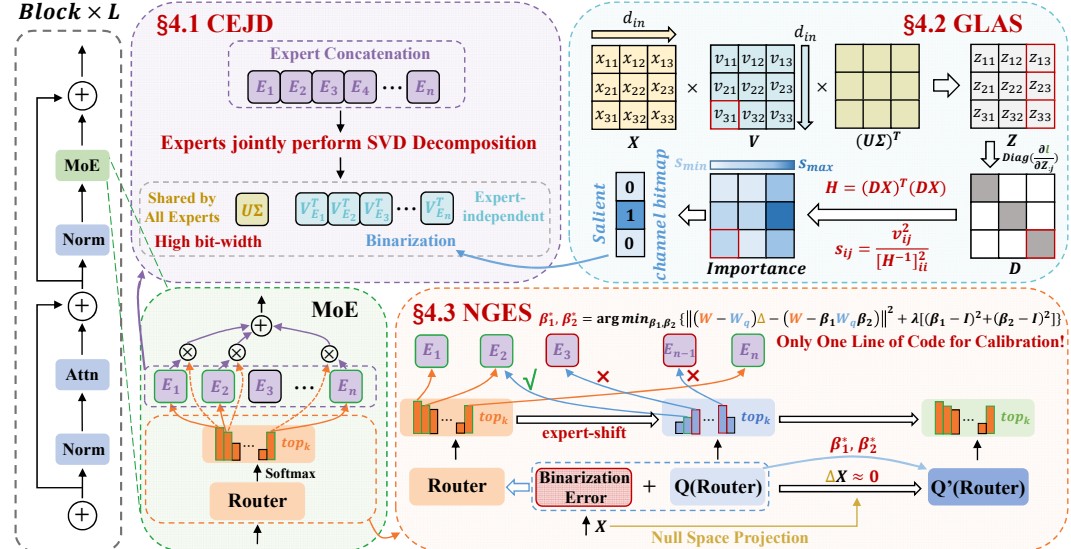

Figure 4: Overall architecture of BiMoE. To address the three challenges of binarizing MoE-based LLMs, BiMoE integrates three components: Cross-Expert Joint Decomposition **(CEJD)** for reducing redundancy, Global Loss-Aligned Saliency **(GLAS)** for correcting saliency misalignment, and Null-Space Guided Expert-Shift Suppression **(NGES)** for mitigating expert-shift.

# 4 METHOD

**Overview.** To address the three challenges of binarizing MoE-based LLMs identified in Sec. 1, we propose BiMoE, a three-stage framework (Fig. 4). First, to tackle expert redundancy, Cross-Expert Joint Decomposition (CEJD) extracts a shared high-precision backbone and binarizes only expert-specific projections (Sec. 4.1). Second, to correct misaligned saliency metrics, Global Loss-Aligned Saliency (GLAS) integrates task-level gradients into Hessian-based saliency, aligning saliency with downstream performance (Sec. 4.2). Third, to address expert shift, Null-Space Guided Expert-Shift Suppression (NGES) confines perturbations to routing-insensitive subspaces via lightweight scaling (Sec. 4.3). The general workflow and pseudocode are provided in Appendix Sec. A.3.

## 4.1 CROSS-EXPERT JOINT DECOMPOSITION (CEJD)

In MoE-based LLMs, a central challenge in binarization is the substantial accuracy degradation observed when directly binarizing full-precision expert weights, largely because existing methods do not exploit cross-expert structural redundancy.

To investigate this, we analyze the original expert weights. As shown in Fig. 2 (top), the CKA similarities typically range from 0.3 to 0.6, indicating moderate overlap across experts. From the perspectives of manifold learning and representation alignment, these observations suggest a low-dimensional structure shared across experts. Motivated by these observations, we propose Cross-Expert Joint Decomposition (CEJD, Fig. 4), which jointly decomposes expert weight matrices to extract a shared backbone while disentangling expert-specific variations. This design mitigates binarization-induced accuracy loss and reduces cross-expert redundancy (see Fig. 2 (bottom)). Concretely, within a layer we horizontally concatenate the expert weight matrices and compute the SVD:

$$\text{SVD}\left(\left[\boldsymbol{W}^{(1)}, \boldsymbol{W}^{(2)}, \cdots, \boldsymbol{W}^{(n)}\right]\right) = \underbrace{(\boldsymbol{U\Sigma})_{\text{shared}}}_{\text{High bit-width}} \underbrace{\left[\boldsymbol{V}^{(1)}, \boldsymbol{V}^{(2)}, \cdots, \boldsymbol{V}^{(n)}\right]^{\top}}_{\text{Binarization}}, \quad (4)$$

where $r = \min(O, nI)$; $\boldsymbol{U} \in \mathbb{R}^{O \times r}$ contains the left singular vectors forming an orthonormal output basis; $\boldsymbol{\Sigma} \in \mathbb{R}^{r \times r}$ holds the singular values that rank the basis dimensions by importance; and $\boldsymbol{V}^{\top} \in \mathbb{R}^{r \times nI}$ is partitioned into $n$ contiguous blocks (each of size $r \times I$), each associated with one expert. Hence, $\boldsymbol{U}$ provides a shared semantic basis, $\boldsymbol{\Sigma}$ ranks its dimensions, and $\boldsymbol{V}$ encodes the expert-specific projections.

We retain the shared backbone $U\Sigma$ in high precision and binarize only the expert-specific blocks $V^{(i)\top}$ (see Appendix Sec. A.1.1). Compared with direct binarization, this yields two benefits: (i) the high-precision backbone preserves cross-expert collaboration and helps prevent feature collapse; and (ii) because most of the energy is captured by the backbone, quantization noise is confined to a stable orthogonal basis, thereby substantially reducing error. In MoE-based LLMs, a single high-precision backbone per MLP layer (shared by all experts) adds negligible storage overhead and a negligible increase in active parameters while preserving expressiveness, thereby offering a superior compression–accuracy trade-off (see Appendix Sec. A.2).

## 4.2 GLOBAL LOSS-ALIGNED SALIENCY (GLAS)

Prior studies (Shang et al., 2023) show that effective binarization critically depends on accurately identifying and exploiting salient weights. A common practice is to construct a saliency matrix $s$ via a local Hessian criterion, with each element defined as follows:

$$s_{ij} = \frac{w_{ij}^2}{[\boldsymbol{H}^{-1}]_{ii}^2}, \tag{5}$$

where $\boldsymbol{H}$ is the Hessian of the layerwise reconstruction loss used in GPTQ (Frantar et al., 2023) and serves as the saliency criterion. The Hessian is derived as the product of the activation matrix $\boldsymbol{X}$ and its transpose, scaled by a factor of 2:

$$\boldsymbol{H} = 2\boldsymbol{X}\boldsymbol{X}^\top, \tag{6}$$

with $\boldsymbol{X}$ denoting the current layer's activations. Because this construction relies solely on local activations, the resulting saliency primarily reflects reconstruction sensitivity rather than task-level impact. To address this limitation, we propose Global Loss-Aligned Saliency (GLAS), which explicitly incorporates downstream loss gradients into the Hessian construction, thereby making the scoring task-aware (derivations in Appendix Sec. A.1.2). This alignment better captures how features influence the global objective and improves saliency discrimination under binarization.

We begin with the first-order sensitivity of the global loss. A first-order Taylor expansion of $\ell$ at $\boldsymbol{Z}$ evaluated at $\hat{\boldsymbol{Z}}$ yields

$$\ell(\hat{\boldsymbol{Z}}) - \ell(\boldsymbol{Z}) \approx \sum_{i,j} \frac{\partial \ell}{\partial \boldsymbol{Z}_{ij}} (\hat{\boldsymbol{Z}}_{ij} - \boldsymbol{Z}_{ij}) = \langle \nabla_{\boldsymbol{Z}} \ell, \ \hat{\boldsymbol{Z}} - \boldsymbol{Z} \rangle, \tag{7}$$

where the magnitude of the gradient entry $\frac{\partial \ell}{\partial \boldsymbol{Z}_{ij}}$ quantifies the global sensitivity of each output component. This motivates replacing the mean-squared reconstruction objective with a gradient-weighted residual:

$$\mathcal{L}_{\text{guided}} = \big\| \frac{\partial \ell}{\partial \boldsymbol{Z}} \odot (\boldsymbol{Z} - \hat{\boldsymbol{Z}}) \big\|_F^2, \tag{8}$$

which upweights errors in directions more sensitive to the global loss and aligns optimization with the global objective. Leveraging the linear relationship between layer outputs and weights ($\boldsymbol{Z} = \boldsymbol{X}\boldsymbol{W}, \hat{\boldsymbol{Z}} = \boldsymbol{X}\hat{\boldsymbol{W}}$), the output error can be converted into weight error: $\boldsymbol{Z} - \hat{\boldsymbol{Z}} = \boldsymbol{X}(\boldsymbol{W} - \hat{\boldsymbol{W}})$. Accordingly, the objective function can be decomposed along the output channel $j$ as:

$$\mathcal{L}_{\text{guided},j} = (\boldsymbol{W}_{:j} - \hat{\boldsymbol{W}}_{:j})^\top (\boldsymbol{D}_j \boldsymbol{X})^\top (\boldsymbol{D}_j \boldsymbol{X}) (\boldsymbol{W}_{:j} - \hat{\boldsymbol{W}}_{:j}), \tag{9}$$

where $\boldsymbol{D}_j = \text{Diag}\big(\frac{\partial \ell}{\partial \boldsymbol{Z}_{:j}}\big)$ is the diagonal matrix of global gradients for output channel $j$. This naturally leads to the global Hessian:

$$H_{\text{global},j} = (\boldsymbol{D}_j \boldsymbol{X})^\top (\boldsymbol{D}_j \boldsymbol{X}) = \sum_i \left( \frac{\partial \ell}{\partial \boldsymbol{Z}_{ij}} \right)^2 \boldsymbol{X}_{i:}^\top \boldsymbol{X}_{i:}, \tag{10}$$

which preserves the input covariance geometry while emphasizing directions most sensitive to the global loss. We then redefine saliency via the global Hessian:

$$s_{ij} = \frac{w_{ij}^2}{\big[(\boldsymbol{H}_{\text{global},j}^{-1})_{ii}\big]^2}. \tag{11}$$

This task-aligned criterion prioritizes weights most critical to downstream performance. Consequently, GLAS elevates saliency from a local, reconstruction-oriented proxy to a measure explicitly informed by the global objective, yielding a more discriminative basis for binarization.

### 4.3 NULL-SPACE GUIDED EXPERT-SHIFT SUPPRESSION (NGES)

In MoE-based LLMs, quantization perturbations not only distort expert outputs but also propagate to the gating distribution, inducing expert-shift (i.e., misrouting tokens to suboptimal experts). Unlike mere numerical errors, expert-shift undermines the division of labor and amplifies quantization noise, thereby posing a major challenge for binarization. As shown in Table 1, perplexity rises sharply once expert-shift occurs, even with full-precision weights. Suppressing expert shift substantially alleviates degradation, underscoring the need for effective control.

Table 1: Impact of Weight Binarization and Its Induced Expert-Shift on Wikitext-2 Perplexity of OLMoE-1B-7B and DeepSeek-V2-Lite (ARB-LLM as Binarization Method).

| Model | Quantized | Expert-Shift | PPL↓ |
|---|---|---|---|
| OLMoE-1B-7B | x | x | 6.65 |
| | x | ✓ | 10.83(+4.18) |
| | ✓ | x | 13.12(+6.47) |
| | ✓ | ✓ | 15.49(+8.84) |
| DeepSeek-V2-Lite | x | x | 6.31 |
| | x | ✓ | 12.21(+5.90) |
| | ✓ | x | 15.33(+9.02) |
| | ✓ | ✓ | 18.77(+12.46) |

Motivated by this observation, we propose Null-Space Guided Expert-Shift Suppression (NGES), which constrains binarization errors to routing-insensitive directions in the input space. Since gating scores depend on the interaction between activations $X$ and weights $W$, we define routing-insensitive directions as the null space of $X$ (equivalently, the eigenspace of $X^\top X$ associated with near-zero eigenvalues). Its SVD, $\text{SVD}(XX^\top) = U\Sigma V^\top$, yields an orthogonal basis $U$, where the submatrix $U_1$ associated with near-zero singular values defines the null-space projector $P = U_1 U_1^\top$. Projecting binarization errors into this space gives

$$\left\| (W - \hat{W})PX \right\|_2^2 \approx 0. \tag{12}$$

However, explicitly storing $P$ is memory-intensive. To address this, we introduce implicit row-wise and column-wise constraints on the binarized weight matrix $\hat{W}$, enabling null-space optimization without storing large projection matrices. Specifically, for each $\hat{W}$, we construct two modulation vectors—a row vector $\beta^r \in \mathbb{R}^m$ and a column vector $\beta^c \in \mathbb{R}^n$—and fuse them with the standard binarization scaling factors $\alpha^r \in \mathbb{R}^m$ and $\alpha^c \in \mathbb{R}^n$ via the Hadamard product, leading to

$$\hat{W} = (\beta^r \cdot \alpha^r) \cdot \text{Sign}(W) \cdot (\alpha^c \cdot \beta^c), \tag{13}$$

where $\alpha^r$ and $\alpha^c$ are the row and column scaling factors defined in the binarization process (Li et al., 2024), with their elements formally defined as $\alpha_j^r = \frac{1}{m}\sum_{j=1}^m |W_{:j}|, \alpha_j^c = \frac{1}{n}\sum_{j=1}^n |\frac{W_{j:}}{\alpha_j^r}|$ for an $m \times n$ weight matrix $W$.

To ensure that these vectors approximate the effect of $P$, we minimize the discrepancy between the null-space projection and its implicit approximation. However, optimizing this term alone can distort the binarized weight distribution or overfit the null-space constraint. We therefore add a small regularizer that keeps $\beta^r$ and $\beta^c$ close to the all-ones vector. The complete regularized objective is

$$\beta_*^r, \beta_*^c = \arg\min_{\beta_r, \beta_c} \left( \left\| (W - \hat{W})P - (W - \beta^r \hat{W}\beta^c) \right\|_F^2 + (\lambda((\beta_r - I)^2 + (\beta_c - I)^2)) \right), \tag{14}$$

where the first term mitigates expert-shift by limiting the routing-sensitive impact of binarization perturbations, while the regularization coefficient $\lambda$ (empirically set to 0.2) controls the trade-off between constraint strength and weight stability. Instead of costly backpropagation, we cast the objective as a least-squares problem with a closed-form solution. Because the quadratic form is strictly positive definite for $\lambda > 0$, the objective is strongly convex and thus admits a unique minimizer. To simplify optimization, we adopt an alternating strategy: initialize $\beta^r = 1$ and $\beta^c = 1$, then iteratively optimize one while holding the other fixed until convergence (derivations in Appendix Sec. A.1.3). This decomposition reduces computational overhead while retaining the global

optimality guaranteed by convexity. Finally, setting the gradients to zero yields closed-form updates:

$$\boldsymbol{\beta}_*^r = \frac{(\hat{\boldsymbol{W}}\boldsymbol{\beta}_*^c)^\top \Delta + \lambda}{(\hat{\boldsymbol{W}}\boldsymbol{\beta}_*^c)^\top (\hat{\boldsymbol{W}}\boldsymbol{\beta}_*^c) + \lambda}, \quad \boldsymbol{\beta}_*^c = \frac{(\boldsymbol{\beta}_*^r \hat{\boldsymbol{W}})^\top \Delta + \lambda}{(\boldsymbol{\beta}_*^r \hat{\boldsymbol{W}})^\top (\boldsymbol{\beta}_*^r \hat{\boldsymbol{W}}) + \lambda}, \tag{15}$$

where $\Delta = \boldsymbol{W} - (\boldsymbol{W} - \hat{\boldsymbol{W}})\boldsymbol{P}$. After convergence, $\boldsymbol{\beta}_*^r$ and $\boldsymbol{\beta}_*^c$ act directly on $\hat{\boldsymbol{W}}$. While retaining sparsity and low-bit storage of the binarized weights, they achieve quantization error suppression comparable to an explicit null-space projection. This design effectively confines binarization errors to the routing-insensitive null space, thereby suppressing expert-shift.

## 5 EXPERIMENTS

### 5.1 EXPERIMENT SETUP

**Models and Datasets.** We conducted binarization on six open-source MoE-based LLMs with diverse architectures and scales: **OLMoE** (Muennighoff et al., 2025), **Deepseek-V2-Lite** (DeepSeek-AI et al., 2024), **Qwen1.5-MoE-A2.7B** (Team, 2024), **Qwen3-30B-A3B** (Yang

Table 2: Architectural Specifications of Evaluated MoE-based LLMs. Memory values are measured under FP16 precision.

| Model Variant | Memory (FP16, GB) | Experts | TopK |
|---|---|---|---|
| OLMoE-1B-7B-0125 | 12.9 | 64 | 8 |
| Qwen1.5-MoE-A2.7B | 26.7 | 60+4 | 4 |
| DeepSeek-V2-Lite | 29.3 | 64+2 | 6 |
| Qwen3-30B-A3B | 59.1 | 128 | 8 |
| Qwen3-Next-80B-A3B | 160.7 | 512+1 | 10 |
| GPT-OSS-20B | 36.9 | 32 | 4 |

et al., 2025), **Qwen3-Next-80B-A3B-Instruct** (Yang et al., 2025), and **GPT-OSS-20B** (OpenAI et al., 2025). Detailed specifications are summarized in Table 2. To evaluate BiMoE, we measure perplexity on WikiText2 (Merity et al., 2016) and accuracy on seven zero-shot datasets: Arc-Challenge and Arc-Easy (Clark et al., 2018), HellaSwag (Zellers et al., 2019), LAMBADA-openai and LAMBADA-standard (Paperno et al., 2016), PIQA (Bisk et al., 2019), and WinoGrande (Sakaguchi et al., 2019).

**Baseline Methods.** We compare BiMoE against binary LLM baselines: **BiLLM** (Huang et al., 2024) and **ARB-LLM** (Li et al., 2024), as well as the widely adopted PTQ method **GPTQ** (Frantar et al., 2023). We also include vector quantization methods **GPTVQ** (van Baalen et al., 2025) and **NoWag** (Liu et al., 2025), which target extremely low-bit scenarios. Higher-bit quantization methods (4-bit, 8-bit) are excluded as they are not directly comparable to binarization.

**Implementation Details.** Experiments are conducted on a single NVIDIA RTX A6000 GPU using PyTorch (Paszke et al., 2019) and HuggingFace (Wolf et al., 2020), with sequence length 4096. For NGES, we run 15 iterations to ensure convergence of projection vectors, using regularization coefficient $\lambda = 0.2$ (empirically tuned; see Appendix Sec. A.4.6). The binarization procedure follows ARB-LLM, with calibration on 128 random WikiText2 samples. In CEJD, the shared high-precision component $\boldsymbol{U}\Sigma$ is quantized to 8 bits to balance accuracy and efficiency. Being a PTQ framework, BiMoE requires no fine-tuning and completes quantization in a single pass.

### 5.2 MAIN RESULTS

We conducted a systematic evaluation of binarization across diverse MoE-based LLMs and datasets, spanning multiple model scales. As shown in Table 3, BiMoE consistently outperforms all competitive baselines on eight benchmarks, often by large margins. In particular, scalar (GPTQ) and vector (GPTVQ/NoWag) quantizers fail under 1-bit settings, with perplexity diverging or accuracy collapsing toward zero. Moreover, methods originally designed for dense LLMs, such as BiLLM and ARB-LLM, retain partial functionality yet are fundamentally mismatched to MoE-based LLMs, leading to substantially degraded performance.

In contrast, BiMoE delivers consistently superior results. On Qwen3-30B-A3B, for example, it reduces perplexity by 52.2% and substantially improves average zero-shot accuracy by 43.4%, while increasing effective weight precision by only 0.23 bits. Taken together, these results underscore BiMoE's robustness and efficiency, establishing it as a practical solution for MoE-based LLM binarization. Detailed dialogue examples are provided in Appendix Sec. A.5.

Table 3: We evaluate the perplexity on WikiText2 and following datasets: Arc-Challenge (AC), Arc-Easy (AE), HellaSwag (HS), LAMBADA-openai (LO), LAMBADA-standard (LS), PIQA (PQ), and WinoGrande (WG). #Bits indicates the average weight bitwidth of activated parameters during inference. Note that the global average bitwidth of the entire model is around **1.11 bits**.

| Model | Method | #Bits(W) | AE↑ | AC↑ | HS↑ | LO↑ | LS↑ | PQ↑ | WG↑ | Avg.[7]↑ | Wiki PPL↓ |
|---|---|---|---|---|---|---|---|---|---|---|---|
| | Baseline | 16 | 76.77 | 49.23 | 78.21 | 70.83 | 65.57 | 79.76 | 68.82 | 69.88 | 6.65 |
| | GPTQ | 1 | 26.85 | 25.68 | 25.91 | 0 | 0 | 49.51 | 50.51 | 25.49 | 65280.50 |
| | GPTVQ | 1.125 | 26.05 | 24.91 | 25.85 | 0 | 0 | 51.69 | 49.80 | 25.47 | 1777.93 |
| OLMoE | NoWag | 1.02 | 26.98 | 25.17 | 26.25 | 0.04 | 0.02 | 52.12 | 51.85 | 26.06 | 38331.15 |
| | BiLLM | 1.11 | 44.19 | 24.91 | 37.40 | 18.53 | 14.13 | 59.58 | 51.46 | 35.74 | 20.32 |
| | ARB-LLM | 1.11 | 46.51 | 26.96 | 43.45 | 32.31 | 24.98 | 63.33 | 57.62 | 42.16 | 15.49 |
| | **BiMoE** | 1.46 | **61.03** | **33.36** | **51.79** | **47.12** | **29.85** | **67.46** | **58.64** | **49.89** | **14.37** |
| | Baseline | 16 | 69.15 | 44.45 | 77.23 | 71.40 | 64.49 | 80.36 | 68.75 | 67.98 | 7.22 |
| | GPTQ | 1 | 25.88 | 25.00 | 26.07 | 0 | 0 | 52.29 | 50.28 | 25.65 | 887514.69 |
| | GPTVQ | 1.125 | 24.16 | 26.96 | 25.83 | 0 | 0 | 51.09 | 50.36 | 25.49 | NaN |
| Qwen1.5-MoE | NoWag | 1.02 | 26.18 | 26.11 | 25.75 | 0 | 0 | 51.09 | 47.28 | 25.20 | 64366.32 |
| | BiLLM | 1.11 | 37.63 | 23.55 | 34.76 | 6.62 | 10.73 | 58.32 | 50.99 | 31.80 | 26.99 |
| | ARB-LLM | 1.11 | 43.39 | 25.43 | 40.16 | 25.89 | 21.09 | 63.71 | 53.99 | 39.09 | 22.21 |
| | **BiMoE** | 1.35 | **54.08** | **29.35** | **48.67** | **50.18** | **38.40** | **70.08** | **59.75** | **50.07** | **14.86** |
| | Baseline | 16 | 74.03 | 46.16 | 73.55 | 68.37 | 61.36 | 79.27 | 67.64 | 67.20 | 6.31 |
| | GPTQ | 1 | 24.58 | 26.02 | 26.47 | 0 | 0 | 50.87 | 50.51 | 25.49 | 56325976 |
| | GPTVQ | 1.125 | 24.71 | 24.40 | 26.40 | 0 | 0 | 49.13 | 48.38 | 24.72 | 4805.14 |
| DeepSeekV2-Lite | NoWag | 1.02 | 26.51 | 24.53 | 26.88 | 0 | 0 | 49.86 | 47.96 | 25.11 | 20021.20 |
| | BiLLM | 1.11 | 39.44 | 23.81 | 36.39 | 16.61 | 19.10 | 60.12 | 52.96 | 35.49 | 22.43 |
| | ARB-LLM | 1.11 | 44.82 | 26.28 | 38.97 | 26.74 | 27.75 | 61.59 | 54.06 | 40.03 | 18.77 |
| | **BiMoE** | 1.47 | **45.41** | **25.34** | **42.98** | **35.98** | **28.62** | **64.91** | **55.25** | **42.64** | **15.83** |
| | Baseline | 16 | 78.96 | 55.97 | 77.64 | 64.86 | 63.26 | 80.47 | 69.53 | 70.10 | 8.70 |
| | GPTQ | 1 | 24.24 | 26.88 | 25.95 | 0 | 0 | 50.27 | 50.20 | 25.36 | 20167.24 |
| | GPTVQ | 1.125 | 25.08 | 22.70 | 25.04 | 0 | 0 | 49.51 | 49.57 | 24.56 | NaN |
| Qwen3-MoE | NoWag | 1.03 | 26.22 | 24.23 | 26.14 | 0 | 0 | 51.03 | 50.43 | 25.44 | 778096.02 |
| | BiLLM | 1.11 | 32.45 | 23.55 | 35.35 | 17.19 | 14.56 | 55.17 | 52.80 | 33.01 | 52.78 |
| | ARB-LLM | 1.11 | 44.15 | 27.30 | 44.63 | 24.72 | 20.05 | 63.93 | 56.20 | 40.14 | 26.76 |
| | **BiMoE** | 1.34 | **66.04** | **43.17** | **58.54** | **52.01** | **45.55** | **71.82** | **65.90** | **57.58** | **12.80** |

## 5.3 ABLATION STUDIES

BiMoE integrates three core components (CEJD, GLAS, and NGES), each aimed at reducing quantization error for MoE-based LLMs. Our ablation study proceeds in three parts: (i) evaluating the individual contribution of each module; (ii) validating NGES both by accuracy gains and by its mitigation of expert shift after quantization; and (iii) analyzing the effective runtime bitwidth of active weights and the accuracy impact of varying the $U\Sigma$ precision in CEJD. Details of the calibration data settings are provided in Appendix Sec. A.4.5.

**Modular Sensitivity Study.** We analyze individual and joint effects of CEJD, GLAS, and NGES. Table 4 shows: (i) each module individually improves zero-shot accuracy, with CEJD contributing the largest gain; and (ii) combining

Table 4: Impact of different components in BiMoE on GPT-OSS-20B and Qwen3-Next-80B-A3B-Instruct.

| Modules | | | GPT-OSS-20B (Avg[4] 67.03) | | | | | Qwen3-Next-80B-A3B (Avg[4] 77.48) | | | | |
|---|---|---|---|---|---|---|---|---|---|---|---|---|
| CEJD | GLAS | NGES | AE↑ | AC↑ | PQ↑ | WG↑ | Avg[4]↑ | AE↑ | AC↑ | PQ↑ | WG↑ | Avg[4]↑ |
| | | | 34.19 | 23.23 | 56.89 | 51.41 | 41.43 | 50.77 | 47.16 | 70.12 | 55.89 | 55.99 |
| ✓ | | | 46.93 | 27.82 | 62.68 | 54.14 | 47.89 | 66.15 | 51.33 | 75.12 | 63.83 | 64.11 |
| | ✓ | | 38.71 | 24.86 | 59.11 | 52.08 | 43.69 | 59.33 | 49.21 | 73.39 | 60.43 | 60.59 |
| | | ✓ | 40.12 | 25.07 | 60.33 | 52.49 | 44.50 | 61.72 | 50.27 | 74.06 | 61.13 | 61.80 |
| ✓ | ✓ | | 53.16 | 31.74 | **66.43** | 56.04 | 51.84 | 74.61 | 52.79 | 76.69 | 67.28 | 67.84 |
| ✓ | ✓ | ✓ | **55.47** | **32.17** | 65.89 | **58.80** | **53.08** | **80.35** | **54.61** | **79.54** | **71.35** | **71.46** |

all three yields the best results, boosting average accuracy on Qwen3-Next-80B-A3B-Instruct by 15.47% over baseline, demonstrating necessity and synergy. Beyond accuracy, we evaluate NGES's suppression of expert shift in OLMoE via routing consistency $S_\ell$ (mean cosine similarity between quantized and full-precision routing probabilities). As shown in Fig. 5 (inset, OLMoE), BiLLM and ARB-LLM fall below $S_\ell = 0.8$ in deeper layers, whereas BiMoE+NGES maintains $S_\ell \geq 0.9$ (drop $\leq 0.1$), indicating stable routing. Additional analyses appear in Appendix Sec. A.4.4. Moreover, Appendix Sec. A.4.7 shows that GLAS and NGES integrate seamlessly into BiLLM and ARB-LLM and—without retraining—consistently improve MoE-based LLMs quantization performance.

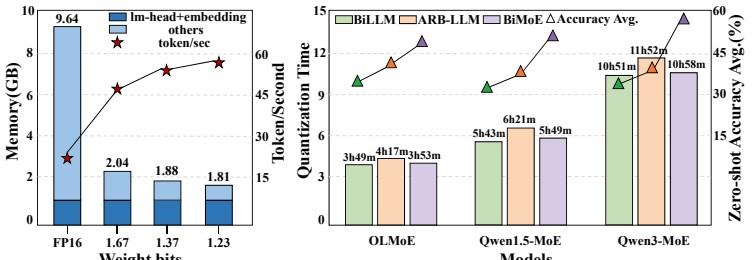

Figure 5: Comparison of expert activation similarity between original FP16 and binary models.

Table 5: Comparison of weight bitwidth and performance for OLMoE/Qwen1.5-MoE under different bit settings of shared $U\Sigma$.

| Model | $U\Sigma$-bitwidth | #Bits(W) | Wiki PPL↓ | Avg[7]↑ |
|-------|--------|----------|-----------|--------|
| OLMoE | / | 16 | 6.65 | 69.88 |
| | 16 | 1.97 | 14.67 | 49.83 |
| | 8 | 1.46 | **14.37** | **49.89** |
| | 4 | 1.23 | 16.29 | 47.43 |
| Qwen1.5-MoE | / | 16 | 7.22 | 67.98 |
| | 16 | 1.67 | 15.23 | 48.83 |
| | 8 | 1.35 | **14.86** | **50.07** |
| | 4 | 1.23 | 17.81 | 44.49 |

**Bit Setting Analysis of Shared $U\Sigma$.** Table 5 reports the effect of different bitwidths for the shared $U\Sigma$ representation in OLMoE and Qwen1.5-MoE. Interestingly, the 8-bit setting not only matches but even surpasses the 16-bit baseline: perplexity decreases (14.67 → 14.37 on OLMoE; 15.23 → 14.86 on Qwen1.5-MoE), and average zero-shot accuracy improves as well (49.83 → 49.89; 48.83 → 50.07). By contrast, the 4-bit setting leads to a sharp performance drop. These results indicate that 8-bit provides the best efficiency–accuracy trade-off and is therefore adopted as the default.

## 5.4 EFFICIENCY ANALYSIS OF BiMoE

**Speedup and Memory Savings.** We benchmarked BiMoE's system efficiency on a single NVIDIA RTX A6000 GPU, comparing throughput (tokens/s) and activated parameter memory usage between Qwen1.5-MoE and its binarized counterpart, measured across average weight bitwidths (determined by the precision of shared $U\Sigma$). With batch size 4, prefill 1024 tokens, and decoding 256 tokens, Fig. 6 (left) shows at 1.37 bits (corresponding to the 8-bit $U\Sigma$ setting),

Figure 6: Efficiency Analysis. **Left**: Throughput (Tokens/Sec) and memory consumption of the original FP16 model and BiMoE under different average weight bitwidths. **Right**: Comparison of quantization time and average zero-shot accuracy across three methods (BiLLM, ARB-LLM, and BiMoE) on different MoE-based LLMs.

throughput increases from 22.05 tokens/s (FP16) to 51.38 tokens/s (2.33× speedup), while activated parameter memory falls from 9.64 GB (FP16) to 1.88 GB, with expert-layer memory reduced by over 90%. These results demonstrate BiMoE's substantial efficiency gains.

**Quantization Time Comparison.** As a fine-tuning-free PTQ framework, BiMoE performs per-layer SVD and null-space iterative updates offline, incurring no runtime overhead. Fig. 6 (right) shows that BiMoE delivers higher quantization throughput than ARB-LLM, whose column-group bitmap requires a costly percentile-based search. Compared with BiLLM, which lacks such iterative updates, BiMoE is only about six minutes slower on average across three models. Yet BiMoE achieves about 20% higher average zero-shot accuracy than BiLLM and over 10% higher than the slower ARB-LLM. These results underscore BiMoE's efficiency–accuracy advantages.

## 6 CONCLUSION

We present BiMoE, a binarization framework tailored to MoE-based LLMs. By jointly addressing expert redundancy, saliency misalignment, and expert-shift, BiMoE strikes a strong balance between efficiency and accuracy in this challenging setting. Our experiments show that BiMoE surpasses state-of-the-art binary PTQ methods and achieves superior inference efficiency, making it a practical solution for the deployment of large-scale MoE-based LLMs. We release our open-source framework to facilitate research on binarization and to broaden the applicability of efficient MoE-based LLMs. Furthermore, we discuss limitations of our method in Appendix Sec. A.7.

ETHICS STATEMENT

This work focuses on algorithmic techniques for compressing Mixture-of-Experts (MoE) large language models—specifically CEJD, GLAS, and NGES within the BiMoE framework—and does not involve human subjects, new data collection, or the processing of personally identifiable or sensitive information. We evaluate on widely used public corpora and third-party model checkpoints under their original licenses, and we only release aggregate metrics, code, and (when permitted) model configuration files; we do not redistribute underlying datasets or proprietary weights.

REPRODUCIBILITY STATEMENT

We provide comprehensive descriptions of our algorithms, hyperparameter settings, and evaluation protocols in both the main text and the appendix. All models are systematically benchmarked on publicly available datasets, including WikiText-2, ARC, PIQA, HellaSwag, LAMBADA, Wino-Grande, and other widely adopted language benchmarks. To further support the open-source community, we have released all experimental code. These resources are intended to ensure that the reported findings are fully transparent, verifiable, and reproducible.

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

APPENDIX OVERVIEW

# A   APPENDIX

## A.1   DETAILED PROOFS

### A.1.1   DETAILED THEORETICAL DERIVATION OF CEJD

The key idea of CEJD is that MoE-based LLMs experts, while parameterized independently, share functional similarity. From a manifold learning and representation alignment perspective, each expert's weight matrix lies on a different submanifold of the parameter space, but these submanifolds exhibit a common latent structure due to shared semantics across experts. Our goal is to recover this shared structure explicitly and disentangle expert-specific variations.

Consider $n$ experts within the same MoE-based LLMs layer, each with weight matrix $W^{(i)} \in \mathbb{R}^{O \times I}$, where $I$ and $O$ denote input and output dimensions respectively. We horizontally concatenate them:

$$W_{\text{concat}} = \left[ W^{(1)}, W^{(2)}, \cdots, W^{(n)} \right] \in \mathbb{R}^{O \times nI}. \tag{16}$$

Applying singular value decomposition (SVD) yields

$$W_{\text{concat}} = U\Sigma V^{\top}, \tag{17}$$

where $U \in \mathbb{R}^{O \times r}$ contains orthogonal basis vectors spanning the shared output feature subspace, $\Sigma \in \mathbb{R}^{r \times r}$ is diagonal with singular values quantifying importance of each basis dimension, and $V^{\top} \in \mathbb{R}^{r \times nI}$ encodes expert-specific projections.

Partition $V^{\top}$ into $n$ blocks:

$$V^{\top} = \left[ V^{(1)\top}, V^{(2)\top}, \cdots, V^{(n)\top} \right], \tag{18}$$

so that each expert weight can be reconstructed as

$$W^{(i)} \approx U\Sigma V^{(i)T}. \tag{19}$$

**Optimization View.** This decomposition can be equivalently formulated as minimizing the reconstruction error across all experts on real data samples $x \in \mathbb{R}^I$:

$$\min_{U, \Sigma, \{V^{(i)}\}} \sum_{i=1}^{n} \mathbb{E}_{x \sim \mathcal{D}} \left\| W^{(i)} x - U \Sigma V^{(i)T} x \right\|_2^2, \tag{20}$$

subject to $U^\top U = \boldsymbol{I}$ (orthogonality constraint). This reveals CEJD as a form of constrained low-rank approximation: experts are first projected into a common subspace $U$, aligned by $\Sigma$, and only the lightweight projections $V^{(i)}$ differ across experts.

**Why preserve $U\Sigma$ in high precision?** The shared backbone $U\Sigma$ captures the semantic manifold common to all experts, ensuring their outputs remain compatible after binarization. Since the majority of variance is absorbed by $U\Sigma$, the residual discretization error from binarized $V^{(i)}$ is confined to a stable coordinate system, preventing feature collapse. The storage overhead of keeping one shared $U\Sigma$ per layer is negligible compared with $n$ full-precision experts, making this strategy highly efficient.

In summary, CEJD transforms the problem of binarizing heterogeneous experts into binarizing lightweight projections within a unified subspace, preserving both expressivity and cross-expert consistency.

### A.1.2 DETAILED THEORETICAL DERIVATION OF GLAS

As discussed in Section 4.2, traditional layer-wise saliency metrics for MoE-based LLMs quantization suffer from *local bias*: they construct Hessian matrices solely from input activations of the current layer, ignoring how output perturbations affect the *global task loss* (e.g., perplexity in language modeling). This mismatch can lead to either over-protection or under-protection of critical weights during binarization, thereby amplifying accuracy degradation. Global Loss-Aligned Saliency (GLAS) resolves this by incorporating gradient information of the global loss into the Hessian-based saliency metric, ensuring quantization decisions align with task-level sensitivity. Below we provide a step-by-step theoretical derivation.

**Preliminaries.** For a single MoE-based LLMs expert (the derivation naturally extends to all experts under BiMoE's cross-expert consistency), we define:

- Input activations: $X \in \mathbb{R}^{b \times d}$, where $b$ is batch size, $d$ is hidden dimension;

- Full-precision weights: $W \in \mathbb{R}^{d \times d_{\text{out}}}$; quantized weights: $\hat{W}$, with error $\Delta W = W - \hat{W}$;

- Full-precision output: $Z = XW \in \mathbb{R}^{b \times d_{\text{out}}}$; quantized output: $\hat{Z} = X\hat{W}$, with error $\Delta Z = Z - \hat{Z}$;

- Global task loss: $\ell(\hat{Z})$, e.g., cross-entropy related to perplexity.

Standard notation applies: $\odot$ for Hadamard product, $\| \cdot \|_F$ for Frobenius norm, and $(\cdot)^\top$ for transpose.

**Global sensitivity to output perturbations.** In the PTQ setting of BiMoE, quantization errors $\Delta Z$ are small enough to justify a *first-order Taylor expansion* of the global loss around full-precision outputs:

$$\ell(\hat{Z}) \approx \ell(Z) + \sum_{i=1}^{b} \sum_{j=1}^{d_{\text{out}}} \frac{\partial \ell}{\partial Z_{ij}} \cdot (Z_{ij} - \hat{Z}_{ij}), \tag{21}$$

ignoring higher-order terms. Thus, the loss change is approximately

$$\ell(\hat{Z}) - \ell(Z) \approx \sum_{i=1}^{b} \sum_{j=1}^{d_{\text{out}}} \frac{\partial \ell}{\partial Z_{ij}} \cdot \Delta Z_{ij}. \tag{22}$$

This shows that perturbations on different outputs have unequal impact: the coefficient $\frac{\partial \ell}{\partial Z_{ij}}$ quantifies the *global sensitivity* of $Z_{ij}$. Traditional local metrics (e.g., MSE) treat all $\Delta Z_{ij}$ equally, which GLAS corrects.

**Gradient-guided loss objective.** To prioritize errors with higher sensitivity, GLAS defines a gradient-weighted objective:

$$\mathcal{L}_{\text{guided}} = \left\| \frac{\partial \ell}{\partial Z} \odot \Delta Z \right\|_F^2 = \sum_{i=1}^{b} \sum_{j=1}^{d_{\text{out}}} \left( \frac{\partial \ell}{\partial Z_{ij}} \cdot \Delta Z_{ij} \right)^2. \tag{23}$$

Compared with the unweighted local MSE objective, $\mathcal{L}_{\text{guided}}$ directly emphasizes output errors that most affect global loss.

**From output error to weight error.** Since binarization acts on weights, we substitute

$$\Delta Z = X \Delta W, \tag{24}$$

to rewrite Eq. equation 23. For output channel $j$, define

$$\mathcal{L}_{\text{guided},j} = \left\| D_j X \Delta W_{:j} \right\|_2^2,$$

where $D_j = \text{Diag}\left( \frac{\partial \ell}{\partial Z_{1j}}, \ldots, \frac{\partial \ell}{\partial Z_{bj}} \right)$.

**Global Hessian derivation.** Expanding,

$$\mathcal{L}_{\text{guided},j} = (D_j X \Delta W_{:j})^\top (D_j X \Delta W_{:j}) = \Delta W_{:j}^\top \left( (D_j X)^\top (D_j X) \right) \Delta W_{:j}. \tag{25}$$

Thus, the global Hessian is

$$H_{\text{global},j} = (D_j X)^\top (D_j X) = \sum_{i=1}^{b} \left( \frac{\partial \ell}{\partial Z_{ij}} \right)^2 X_{i:}^\top X_{i:}. \tag{26}$$

Unlike the local Hessian $H_{\text{local}} = 2X^\top X$, which weights all samples equally, $H_{\text{global},j}$ explicitly reweights samples by their global sensitivity, aligning curvature estimation with the downstream task.

**Saliency metric.** Finally, for a weight $W_{ij}$, its saliency is defined by combining magnitude and sensitivity:

$$s_{ij} = \frac{W_{ij}^2}{\left( [H_{\text{global},j}^{-1}]_{ii} \right)^2}. \tag{27}$$

Here, larger $W_{ij}^2$ indicates stronger contribution to outputs, while smaller diagonal entries of $H_{\text{global},j}^{-1}$ indicate higher vulnerability to perturbations. Together, $s_{ij}$ provides a global loss-aligned measure for guiding binarization.

In summary, GLAS replaces purely local sensitivity metrics with a globally consistent Hessian-weighted framework, ensuring that BiMoE preserves task-critical weights during binarization.

### A.1.3 DETAILED THEORETICAL DERIVATION OF THE CLOSED-FORM SOLUTION FOR NGES.

To derive the closed-form solutions for the row projection vector $\beta^r$ and column projection vector $\beta^c$ under the alternating optimization strategy, we start from the objective function defined in the main text (Eq. 14):

$$\beta_*^r, \beta_*^c = \arg \min_{\beta_r, \beta_c} \left\| (W - \hat{W}) \boldsymbol{P} - (W - \beta^r \hat{W} \beta^c) \right\|_F^2 + \lambda \left( (\beta_r - I)^2 + (\beta_c - I)^2 \right) \tag{28}$$

where $\hat{W}$ denotes the binarized weight matrix, $\boldsymbol{P}$ is the null-space projection matrix, and $\Delta = W - (W - \hat{W})\boldsymbol{P}$ (serving as the target reference for optimization). We adopt an alternating optimization strategy: fix one vector and optimize the other iteratively until convergence, which decomposes the bivariate optimization into tractable univariate problems.

**Step 1: Fix $\beta^c$ and Optimize $\beta^r$**

Let $\beta^c = \tilde{\beta}^c$ (initialized as the identity matrix $I$ or updated from the previous iteration). The objective function simplifies to a univariate optimization over $\beta^r$:

$$\mathcal{L}(\beta^r) = \left\| \beta^r(\hat{W}\tilde{\beta}^c) - \Delta \right\|_F^2 + \lambda(\beta^r - I)^2 \tag{29}$$

To find the minimum, we first expand the Frobenius norm term using the trace operator (since $\|\boldsymbol{A}\|_F^2 = \text{Tr}(\boldsymbol{A}^\top \boldsymbol{A})$ for any matrix $\boldsymbol{A}$):

$$\left\| \beta^r(\hat{W}\tilde{\beta}^c) - \Delta \right\|_F^2 = \text{Tr}\left[ \left( \beta^r(\hat{W}\tilde{\beta}^c) - \Delta \right)^\top \left( \beta^r(\hat{W}\tilde{\beta}^c) - \Delta \right) \right] \tag{30}$$

Taking the derivative of $\mathcal{L}(\beta^r)$ with respect to $\beta^r$ and setting it to zero (necessary condition for minima):

$$\frac{\partial \mathcal{L}}{\partial \beta^r} = 2(\hat{W}\tilde{\beta}^c)^\top \left( \beta^r(\hat{W}\tilde{\beta}^c) - \Delta \right) + 2\lambda(\beta^r - I) = 0 \tag{31}$$

Rearranging terms to isolate $\beta^r$:

$$\beta^r \left[ (\hat{W}\tilde{\beta}^c)^\top (\hat{W}\tilde{\beta}^c) + \lambda I \right] = (\hat{W}\tilde{\beta}^c)^\top \Delta + \lambda I \tag{32}$$

Solving for $\beta^r$ yields the closed-form solution:

$$\beta_*^r = \frac{(\hat{W}\tilde{\beta}^c)^\top \Delta + \lambda I}{(\hat{W}\tilde{\beta}^c)^\top (\hat{W}\tilde{\beta}^c) + \lambda I} \tag{33}$$

**Step 2: Fix $\beta^r$ and Optimize $\beta^c$**

With $\beta^r$ fixed to $\beta_*^r$ (the solution from Step 1), we now optimize $\beta^c$. The objective function becomes:

$$\mathcal{L}(\beta^c) = \left\| (\beta_*^r\hat{W})\beta^c - \Delta \right\|_F^2 + \lambda(\beta^c - I)^2 \tag{34}$$

Expanding the Frobenius norm term similarly:

$$\left\| (\beta_*^r\hat{W})\beta^c - \Delta \right\|_F^2 = \text{Tr}\left[ \left( (\beta_*^r\hat{W})\beta^c - \Delta \right)^\top \left( (\beta_*^r\hat{W})\beta^c - \Delta \right) \right] \tag{35}$$

Taking the derivative with respect to $\beta^c$ and setting it to zero:

$$\frac{\partial \mathcal{L}}{\partial \beta^c} = 2(\beta_*^r\hat{W})^\top \left( (\beta_*^r\hat{W})\beta^c - \Delta \right) + 2\lambda(\beta^c - I) = 0 \tag{36}$$

Rearranging terms to isolate $\beta^c$:

$$\beta^c \left[ (\beta_*^r\hat{W})^\top (\beta_*^r\hat{W}) + \lambda I \right] = (\beta_*^r\hat{W})^\top \Delta + \lambda I \tag{37}$$

The closed-form solution for $\beta^c$ is thus:

$$\beta_*^c = \frac{(\beta_*^r\hat{W})^\top \Delta + \lambda I}{(\beta_*^r\hat{W})^\top (\beta_*^r\hat{W}) + \lambda I} \tag{38}$$

**Iterative Alternation Procedure**

The optimization proceeds iteratively as follows:

- **Initialization**: Set $\beta^r = I$ and $\beta^c = I$ (aligning with the regularization goal of minimal deviation from the identity matrix).
- **Iteration**: Update $\beta^r$ using Eq. 33 with the current $\beta^c$, then update $\beta^c$ using Eq. 38 with the newly obtained $\beta^r$.
- **Convergence**: Terminate when the element-wise changes in $\beta^r$ and $\beta^c$ fall below a predefined threshold (e.g., $10^{-5}$).

This alternating strategy leverages the strong convexity of the quadratic objective to ensure unique solutions at each step and monotonic convergence to a global optimum. Compared to joint backpropagation, it reduces computational complexity from $O((n + m)^2)$ to $O(n + m)$ per iteration (where $n, m$ are the dimensions of $\beta^r, \beta^c$), making it scalable for large MoE-based LLMs.

## A.2 PARAMETER COMPRESSION ANALYSIS

**Notation.** Let $N$ be the number of experts in a layer. Each expert weight is a matrix $W^{(e)} \in \mathbb{R}^{O \times I}$, and we concatenate all experts column-wise into $\bar{W} = [W^{(1)}, \ldots, W^{(N)}] \in \mathbb{R}^{O \times (NI)}$. Let $r = \text{rank}(\bar{W}) = \min(O, NI)$ be the retained rank after SVD. We denote by $B_{\text{fp}}$ the baseline bitwidth for full-precision storage (e.g., 16 or 32), $B_h$ the bitwidth for the shared high-precision component ($U\Sigma$), and $B_b$ the bitwidth for the expert-specific projections ($V$). At inference, only top-$k$ experts are activated per token.

### A.2.1 STATIC PARAMETER STORAGE

**Baseline (no decomposition).** Storing all expert parameters directly requires

$$S_{\text{base}}^{\#} = N \times O \times I \quad \text{(parameter count)}, \qquad S_{\text{base}}^{\text{bits}} = N \times O \times I \times B_{\text{fp}} \quad \text{(bits)} \tag{39}$$

**After CEJD (parameter count).** Applying cross-expert joint SVD and retaining rank $r$ yields a shared $U\Sigma \in \mathbb{R}^{O \times r}$ and expert-specific blocks $V^{(e)} \in \mathbb{R}^{r \times I}$. The total parameter count becomes

$$S_{\text{CEJD}}^{\#} = O \times r + N \times r \times I \tag{40}$$

A convenient and exact compression factor (CF) against the baseline parameter count is

$$\text{CF}_{\text{static}}^{\#} = \frac{S_{\text{CEJD}}^{\#}}{S_{\text{base}}^{\#}} = \frac{O \times r + N \times r \times I}{N \times O \times I} = \underbrace{\frac{r}{N I}}_{\text{shared term}} + \underbrace{\frac{r}{O}}_{\text{expert term}} \tag{41}$$

**When does parameter *count* reduce?** In principle, the count after CEJD is $S_{\text{CEJD}}^{\#} = Or + NrI$, compared with the baseline $S_{\text{base}}^{\#} = NOI$. If $r$ is truncated below $O$, one can satisfy the reduction condition in Eq. 41. However, in our setting we do not truncate aggressively: we always retain the full $r = \min(O, NI)$. Since in typical MoE-based LLMs $NI > O$, this gives $r = O$, and thus $S_{\text{CEJD}}^{\#} = O^2 + NOI = S_{\text{base}}^{\#} + O^2$ That is, the parameter count strictly increases by an additive $O^2$ term. Yet, because $O^2 \ll NOI$ in practical configurations, this increase is negligible (usually well below 1% of the baseline). Therefore, CEJD can be regarded as *parameter-count preserving* up to a vanishingly small overhead, while enabling substantial bit-level and activation-time compression benefits.

**After CEJD (bits with quantization).** With high-bit quantization for the shared part and low-bit quantization for expert-specific parts, the bit-level storage becomes

$$S_{\text{CEJD}}^{\text{bits}} = O r B_h + N r I B_b, \qquad \text{CF}_{\text{static}}^{\text{bits}} = \frac{S_{\text{CEJD}}^{\text{bits}}}{S_{\text{base}}^{\text{bits}}} = \frac{r}{N I} \frac{B_h}{B_{\text{fp}}} + \frac{r}{O} \frac{B_b}{B_{\text{fp}}} \tag{42}$$

where $B_{\text{fp}}$ is the baseline full precision (e.g., FP16/FP32), $B_h$ is the precision of the shared $U\Sigma$, and $B_b$ is the precision of expert-specific $V$. Even when the parameter *count* is nearly unchanged, setting $B_h \ll B_{\text{fp}}$ and especially $B_b \ll B_{\text{fp}}$ (e.g., $B_h{=}8$, $B_b{=}1$, $B_{\text{fp}}{=}16/32$) yields a strong *bit*-level compression.

Table 6: Storage formulas before and after CEJD. $B_{\text{fp}}$ is the baseline precision (e.g., FP16), $B_h$ the precision of the shared $U\Sigma$, and $B_b$ the precision of expert-specific $V$. Since $r = \min(O, NI)$ and typically $NI \gg O$, we take $r = O$ in practice.

| Quantity | Baseline (no CEJD) | After CEJD (with $r = O$) |
|---|---|---|
| Static params (count) | $N\,O\,I$ | $O^2 + N\,O\,I$ |
| Static storage (bits) | $N\,O\,I\,B_{\text{fp}}$ | $O^2\,B_h + N\,O\,I\,B_b$ |
| Static CF (bits) | — | $\dfrac{O}{N\,I}\dfrac{B_h}{B_{\text{fp}}} + \dfrac{B_b}{B_{\text{fp}}}$ |
| Activation storage (bits) | $k\,O\,I\,B_{\text{fp}}$ | $O^2\,B_h + k\,O\,I\,B_b$ |
| Activation CF (bits) | — | $\dfrac{O}{k\,I}\dfrac{B_h}{B_{\text{fp}}} + \dfrac{B_b}{B_{\text{fp}}}$ |
| Count-level overhead | — | $+O^2$ (negligible vs. $NOI$) |

### A.2.2 ACTIVATION-TIME (ON-DEVICE) PARAMETER MEMORY

At inference, only top-$k$ experts are activated. Without CEJD, the activation-time memory (in bits) is

$$S_{\text{base,act}}^{\text{bits}} \;=\; k\,O\,I\,B_{\text{fp}} \tag{43}$$

With CEJD (and quantization), the shared $U\Sigma$ is resident once per layer, and only $k$ expert-specific blocks are needed:

$$S_{\text{CEJD,act}}^{\text{bits}} \;=\; O\,r\,B_h \;+\; k\,r\,I\,B_b, \qquad \text{CF}_{\text{act}}^{\text{bits}} \;=\; \frac{S_{\text{CEJD,act}}^{\text{bits}}}{S_{\text{base,act}}^{\text{bits}}} \;=\; \frac{r}{k\,I}\frac{B_h}{B_{\text{fp}}} \;+\; \frac{r}{O}\frac{B_b}{B_{\text{fp}}} \tag{44}$$

Compared to Eq. 44, the shared term is divided by $k$ because only $k$ experts are active; thus activation-time savings are usually even more pronounced.

### A.2.3 EFFECTIVE AVERAGE BITWIDTH

For reporting, it is sometimes convenient to express CEJD's memory footprint as an *effective average bitwidth*:

$$\bar{B}_{\text{static}} \;\triangleq\; \frac{S_{\text{CEJD}}^{\text{bits}}}{S_{\text{base}}^{\#}} \;=\; \left(\frac{r}{N\,I}\right) B_h \;+\; \left(\frac{r}{O}\right) B_b \tag{45}$$

$$\bar{B}_{\text{act}} \;\triangleq\; \frac{S_{\text{CEJD,act}}^{\text{bits}}}{S_{\text{base,act}}^{\#}} \;=\; \left(\frac{r}{k\,I}\right) B_h \;+\; \left(\frac{r}{O}\right) B_b \tag{46}$$

These measures allow apples-to-apples comparisons across different $N, O, I, k$. In the main experiments, we report the *activation-time* average bitwidth, since it directly reflects runtime memory usage; the *static* average bitwidth remains essentially unchanged.

**Takeaways.** As show in Table 6, (i) CEJD does not materially reduce the raw parameter *count*: with $r = O$ it adds only an $O^2$ term, negligible relative to $NOI$. (ii) The main efficiency stems from bit-level compression and activation-time compression, enabled by assigning much smaller bitwidths to $B_h$ and especially $B_b$. (iii) Because the shared overhead scales inversely with $NI$ (static) and $kI$ (runtime), larger expert pools and small $k$ values further amplify the advantages.

## A.3 PSEUDOCODE OF BIMOE

### A.3.1 OVERALL OF BIMOE

Algorithm 1 summarizes the pipeline. CEJD extracts and preserves a shared high-precision backbone while isolating lightweight expert-specific projections for binarization. GLAS computes saliency scores that align with global task loss and therefore guide adaptive (mixed-order) binarization. NGES constrains the remaining binarization perturbations into routing-insensitive directions via compact row/column scaling vectors, suppressing expert-shift without extra storage.

---

**Algorithm 1:** BiMoE: Detailed functions process

---

**Input:** Pretrained MoE-based LLMs $\mathcal{M}$ with $L$ layers, calibration dataset $\mathcal{D}$

**Output:** Binarized MoE-based LLMs $\hat{\mathcal{M}}$

**1 for** *each layer* $l = 1, \ldots, L$ **do**

    // Stage 1: Cross-Expert Joint Decomposition (CEJD)

**2**    Concatenate expert weights $\{W_l^{(i)}\}_{i=1}^N$ horizontally;

**3**    Perform SVD:

$$\left[W_l^{(1)}, \cdots, W_l^{(N)}\right] = (U_l \Sigma_l) \, [V_l^{(1)}, \cdots, V_l^{(N)}]^\top$$

    Keep $U_l \Sigma_l$ in high precision (e.g., 8-bit);

**4**    Mark expert-specific projections $V_l^{(i)}$ for binarization;

    // Stage 2: Global Loss-Aligned Saliency (GLAS)

**5**    Collect activations $X_l$ on $\mathcal{D}$;

**6**    Compute global gradients $\nabla_{Z_l} \ell = \partial \ell / \partial Z_l$;

**7**    Build global Hessians $H_{\text{global},j}$ and derive saliency scores $s_{ij}$ (see Algorithm 2);

**8**    Apply saliency-guided mixed-order binarization (Algorithm 3) to obtain $\hat{V}_l^{(i)}$;

    // Stage 3: Null-Space Guided Expert-Shift Suppression (NGES)

**9**    Estimate routing null-space projector $P_l$ from $X_l X_l^\top$ (SVD);

**10**    Initialize projection vectors $\boldsymbol{\beta}^r \leftarrow \mathbf{1}_d, \; \boldsymbol{\beta}^c \leftarrow \mathbf{1}_M$;

**11**    Alternate closed-form updates for $\boldsymbol{\beta}^r, \boldsymbol{\beta}^c$ until convergence (Algorithm 4);

**12**    Fuse scaling:

$$\hat{V}_l^{(i)} \leftarrow (\boldsymbol{\alpha}^r \odot \boldsymbol{\beta}^r) \, \cdot \, \text{Sign}(V_l^{(i)}) \, \cdot \, (\boldsymbol{\alpha}^c \odot \boldsymbol{\beta}^c)$$

    where $\boldsymbol{\alpha}^r, \boldsymbol{\alpha}^c$ are row/col scaling factors from Stage 2.

**13** Assemble final binarized MoE-based LLMs: $\hat{\mathcal{M}} = \{U_l \Sigma_l, \hat{V}_l^{(i)}\}_{l,i}$;

**14 return** $\hat{\mathcal{M}}$

---

### A.3.2 GLOBAL LOSS-ALIGNED SALIENCY (GLAS) CALCULATION

As the core of Stage 2 in BiMoE, GLAS fuses global loss sensitivity into Hessian-based saliency estimation so that importance reflects task-level impact rather than only local reconstruction sensitivity. The procedure reuses calibration activations $X$ and computes per-output-channel global Hessians that weight input covariance by squared output gradients.

Algorithm 2 yields per-weight saliency measures that better reflect global loss sensitivity. In practice we aggregate column-wise saliency $s_j = \frac{1}{n} \sum_i s_{ij}$ and threshold by a percentile (e.g., $\tau$=95th) to select columns for second-order refinement.

### A.3.3 SALIENCY-GUIDED MIXED-ORDER BINARIZATION

Inspired by decomposition-based binarization ideas (e.g., ARB-LLM), we apply a *saliency-aware mixed-order* scheme (Algorithm 3): low-saliency columns are approximated with a lightweight first-order binarization, while high-saliency columns receive an additional second-order residual correction. All steps operate on expert-specific projection matrices $V \in \mathbb{R}^{n \times m}$ (rows = output dim in the reduced basis, columns = projection channels).

**First-order (matrix) binarization.** Let $B = \text{sign}(V)$. We approximate

$$V \approx \widehat{V}^{(1)} = \text{diag}(\boldsymbol{\alpha}^r) \, B \, \text{diag}(\boldsymbol{\alpha}^c), \tag{47}$$

where $\boldsymbol{\alpha}^r \in \mathbb{R}^n$ and $\boldsymbol{\alpha}^c \in \mathbb{R}^m$. Minimizing the Frobenius reconstruction error $\|V - \widehat{V}^{(1)}\|_F^2$ yields alternating closed-form updates (per ARB-style derivation):

$$(\alpha^r)_i \leftarrow \frac{\sum_{k=1}^m V_{ik} \, (\alpha^c)_k \, B_{ik}}{\sum_{k=1}^m (\alpha_k^c)^2 \, B_{ik}^2 + \varepsilon}, \qquad (\alpha^c)_k \leftarrow \frac{\sum_{i=1}^n V_{ik} \, (\alpha^r)_i \, B_{ik}}{\sum_{i=1}^n (\alpha_i^r)^2 \, B_{ik}^2 + \varepsilon}, \tag{48}$$

where $\varepsilon$ prevents division by zero. Iterate a small number of times (typically $T \leq 10$).

---

**Algorithm 2:** Global Loss-Aligned Saliency (GLAS) Calculation (Stage 2 of BiMoE)

---

**Input:** Expert-specific projections $V = \{V^{(1)}, ..., V^{(M)}\}$ (from CEJD),
Calibration activations $X \in \mathbb{R}^{b \times d}$,
Global loss function $\ell$ (e.g., cross-entropy for language modelling),
Shared backbone $U\Sigma$ (kept in high precision)
**Output:** Saliency scores $s = \{s_{ij}\}$ for all weights in $V^{(i)}$

**1 for** *each expert* $i = 1, \ldots, M$ **do**

**2**     Compute full-precision output: $Z = X \cdot (U\Sigma) \cdot V^{(i)}$;

**3**     Evaluate calibration loss $\ell_{\text{calib}} = \ell(Z)$ and backpropagate to obtain $\nabla_Z \ell \in \mathbb{R}^{b \times d_{\text{out}}}$;

**4**     **for** *each output channel* $j = 1, \ldots, d_{out}$ **do**

**5**        Form diagonal gradient matrix $D_j = \text{Diag}(\nabla_{Z_{:j}} \ell)$;

**6**        Compute global Hessian:

$$H_{\text{global},j} = (D_j X)^\top (D_j X) = \sum_{k=1}^{b} \left(\tfrac{\partial \ell}{\partial Z_{kj}}\right)^2 X_{k:}^\top X_{k:}.$$

       Compute a numerically stable pseudo-inverse $H_{\text{global},j}^{-1}$ (add small damping $\epsilon I$ if needed);

**7**        **for** *each row index* $p$ *in the projection* $V^{(i)}$ *corresponding to channel* $j$ **do**

**8**           Compute saliency:

$$s_{pj} = \frac{V_{pj}^{(i)2}}{\left([H_{\text{global},j}^{-1}]_{pp} + \varepsilon\right)^2},$$

          where $\varepsilon$ is a tiny constant for numerical stability.

**9** Aggregate all scores $s = \{s_{ij}\}$ and return.

---

**Second-order (residual) correction.** For high-saliency columns we compute the residual $\boldsymbol{R} = V - \widehat{V}^{(1)}$, form $B^{(2)} = \text{sign}(\boldsymbol{R})$, and similarly approximate $\boldsymbol{R}$ by $\widehat{V}^{(2)} = \text{diag}(\boldsymbol{\alpha}_2^r) \, B^{(2)} \, \text{diag}(\boldsymbol{\alpha}_2^c)$, optimizing $\boldsymbol{\alpha}_2^r, \boldsymbol{\alpha}_2^c$ with the same alternating updates but restricting sums to the selected high-saliency columns (or equivalently zeroing low-saliency columns in $\boldsymbol{R}$). The final reconstruction is $\widehat{V} = \widehat{V}^{(1)} + \widehat{V}^{(2)}$ for high-saliency columns and $\widehat{V} = \widehat{V}^{(1)}$ otherwise.

**Implementation notes.** (1) Using matrix-level first-order updates is efficient and numerically stable; second-order correction is restricted to a small set of columns (e.g., 5% most salient), so overhead is modest. (2) Use a small damping $\varepsilon$ (e.g., $10^{-8}$) in denominators and guard iteration counts to avoid overfitting to calibration data.

### A.3.4    NULL-SPACE GUIDED EXPERT-SHIFT SUPPRESSION (NGES) EXECUTION

NGES targets gate (router) weights to suppress routing perturbation (expert-shift). Below we give an implementable (Algorithm 4) alternating update for the compact row/column projection vectors $\boldsymbol{\beta}^r \in \mathbb{R}^{d_{\text{gate}}}$ and $\boldsymbol{\beta}^c \in \mathbb{R}^M$. These vectors are fused into the existing row/col scales (no extra matrix storage) via elementwise product: final gate binarized weights become

$$\widehat{G}_{\text{NGES}} = \text{diag}(\boldsymbol{\alpha}^r \odot \boldsymbol{\beta}^r) \, \text{Sign}(G) \, \text{diag}(\boldsymbol{\alpha}^c \odot \boldsymbol{\beta}^c). \tag{49}$$

Derivation (sketch): with $\widehat{G}$ denoting the Stage-2 binarized gate weights and $\Delta \equiv G - (G - \widehat{G})P$ the target that approximates null-space-constrained fidelity, the regularized least-squares objective

$$\min_{\boldsymbol{\beta}^r, \boldsymbol{\beta}^c} \|\Delta - \text{diag}(\boldsymbol{\beta}^r) \, \widehat{G} \, \text{diag}(\boldsymbol{\beta}^c)\|_F^2 + \lambda\left(\|\boldsymbol{\beta}^r - \mathbf{1}\|_2^2 + \|\boldsymbol{\beta}^c - \mathbf{1}\|_2^2\right) \tag{50}$$

decouples to closed-form per-row / per-column updates when alternating and treating the other vector fixed. The per-element updates used in practice are:

---

**Algorithm 3:** Saliency-Guided Mixed-Order Binarization (Stage 2 of BiMoE)

---

**Input:** Expert projection $V \in \mathbb{R}^{n \times m}$, saliency scores $s \in \mathbb{R}^m$, threshold $\tau$, max iter $T$, tol $\epsilon$

**Output:** Binarized projection $\hat{V}$

1 Partition columns: $\mathcal{C}_{\text{high}} = \{j \mid s_j > \tau\}$, $\mathcal{C}_{\text{low}} = \{j \mid s_j \leq \tau\}$;

   // (A) First-order approximation for full matrix

2 $B \leftarrow \text{sign}(V)$;

3 Initialize $\boldsymbol{\alpha}^r \leftarrow \mathbf{1}_n$, $\boldsymbol{\alpha}^c \leftarrow \mathbf{1}_m$;

4 **for** $t = 1$ **to** $T$ **do**

5     Update each row-scale:
$$(\alpha^r)_i \leftarrow \frac{\sum_{k=1}^m V_{ik} \, (\alpha^c)_k \, B_{ik}}{\sum_{k=1}^m (\alpha_k^c)^2 \, B_{ik}^2 + \varepsilon}$$

       Update each col-scale:
$$(\alpha^c)_k \leftarrow \frac{\sum_{i=1}^n V_{ik} \, (\alpha^r)_i \, B_{ik}}{\sum_{i=1}^n (\alpha_i^r)^2 \, B_{ik}^2 + \varepsilon}$$

    **if** *maximum relative change in* $\boldsymbol{\alpha}^r, \boldsymbol{\alpha}^c < \epsilon$ **then**

6        break

7 $\widehat{V}^{(1)} \leftarrow \text{diag}(\boldsymbol{\alpha}^r) \, B \, \text{diag}(\boldsymbol{\alpha}^c)$;

   // (B) Residual correction on high-saliency columns only

8 $\boldsymbol{R} \leftarrow V - \widehat{V}^{(1)}$;

9 Zero out low-saliency columns in $\boldsymbol{R}$: for $j \notin \mathcal{C}_{\text{high}}$, set $\boldsymbol{R}_{:,j} \leftarrow 0$;

10 $B^{(2)} \leftarrow \text{sign}(\boldsymbol{R})$;

11 Initialize $\boldsymbol{\alpha}_2^r \leftarrow \mathbf{1}_n$, $\boldsymbol{\alpha}_2^c \leftarrow \mathbf{1}_m$;

12 **for** $t = 1$ **to** $T$ **do**

13     Update row/col scales using $\boldsymbol{R}$ and $B^{(2)}$ (same formulas as above, but sums only over columns in $\mathcal{C}_{\text{high}}$);

14     **if** *maximum relative change* $< \epsilon$ **then**

15        break

16 $\widehat{V}^{(2)} \leftarrow \text{diag}(\boldsymbol{\alpha}_2^r) \, B^{(2)} \, \text{diag}(\boldsymbol{\alpha}_2^c)$;

   // (C) Final reconstruction

17 **for** *each column* $j$ **do**

18     **if** $j \in \mathcal{C}_{high}$ **then**

19        $\hat{V}_{:,j} \leftarrow \widehat{V}_{:,j}^{(1)} + \widehat{V}_{:,j}^{(2)}$

20     **else**

21        $\hat{V}_{:,j} \leftarrow \widehat{V}_{:,j}^{(1)}$

22 **return** $\hat{V}$

---

$$(\beta^r)_i \leftarrow \frac{\sum_{k=1}^M \widehat{G}_{ik} \, (\beta^c)_k \, \Delta_{ik} \, + \, \lambda}{\sum_{k=1}^M \left( \widehat{G}_{ik} \, (\beta^c)_k \right)^2 \, + \, \lambda \, + \, \varepsilon}, \tag{51}$$

$$(\beta^c)_k \leftarrow \frac{\sum_{i=1}^{d_{\text{gate}}} (\beta^r)_i \, \widehat{G}_{ik} \, \Delta_{ik} \, + \, \lambda}{\sum_{i=1}^{d_{\text{gate}}} \left( (\beta^r)_i \, \widehat{G}_{ik} \right)^2 \, + \, \lambda \, + \, \varepsilon}. \tag{52}$$

These updates are elementwise and numerically stable; they also have a clear ridge regularization interpretation (the $\lambda$ term encourages $\boldsymbol{\beta}^r, \boldsymbol{\beta}^c$ to remain close to 1).

**Remarks.** (1) In contrast to explicitly storing a dense projector $P$, NGES achieves similar routing-protection effects with two compact vectors; this respects the "no extra storage" design. (2) The per-element closed-form updates are inexpensive (loops are trivially parallelizable) and converge

---

**Algorithm 4:** Null-Space Guided Expert-Shift Suppression (NGES) Execution (Gate-weight targeted)

---

**Input:** Full-precision gate weights $G \in \mathbb{R}^{d_{\text{gate}} \times M}$, Stage-2 binarized gate weights $\widehat{G}$, null-space projector $P$ (from SVD of $X_{\text{gate}} X_{\text{gate}}^{\top}$), regularization $\lambda$ (default 0.1), max iter $T$ (default 15), tol $\epsilon$, small $\varepsilon$ for stability

**Output:** NGES-corrected binarized gate weights $\widehat{G}_{\text{NGES}}$

   // Step 0:  Precompute target

1  $\Delta \leftarrow G - (G - \widehat{G}) \cdot P$;

   // Step 1:  Initialize row/col projection vectors (vectors, not matrices)

2  $\boldsymbol{\beta}^r \leftarrow \mathbf{1}_{d_{\text{gate}}}, \quad \boldsymbol{\beta}^c \leftarrow \mathbf{1}_M$;

3  **for** $t = 1$ **to** $T$ **do**

     // Update row-wise scalars

4    **for** $i = 1$ **to** $d_{gate}$ **do**

5      $\text{numerator}_r \leftarrow \sum_{k=1}^{M} \widehat{G}_{ik} (\beta^c)_k \Delta_{ik} + \lambda$;

6      $\text{denominator}_r \leftarrow \sum_{k=1}^{M} \left( \widehat{G}_{ik} (\beta^c)_k \right)^2 + \lambda + \varepsilon$;

7      $(\beta^r)_i \leftarrow \dfrac{\text{numerator}_r}{\text{denominator}_r}$;

     // Update column-wise scalars

8    **for** $k = 1$ **to** $M$ **do**

9      $\text{numerator}_c \leftarrow \sum_{i=1}^{d_{\text{gate}}} (\beta^r)_i \widehat{G}_{ik} \Delta_{ik} + \lambda$;

10     $\text{denominator}_c \leftarrow \sum_{i=1}^{d_{\text{gate}}} \left( (\beta^r)_i \widehat{G}_{ik} \right)^2 + \lambda + \varepsilon$;

11     $(\beta^c)_k \leftarrow \dfrac{\text{numerator}_c}{\text{denominator}_c}$;

     // Convergence check (max element-wise change)

12    $\text{err}_r \leftarrow \|\boldsymbol{\beta}^r_{\text{new}} - \boldsymbol{\beta}^r\|_{\infty}, \quad \text{err}_c \leftarrow \|\boldsymbol{\beta}^c_{\text{new}} - \boldsymbol{\beta}^c\|_{\infty}$;

13    **if** $err_r < \epsilon$ **and** $err_c < \epsilon$ **then**

14     break

   // Step 4:  Fuse the projection vectors into final binarized gate weights

15  Retrieve row/col scales from Stage 2: $\boldsymbol{\alpha}^r, \boldsymbol{\alpha}^c$;

16

$$\widehat{G}_{\text{NGES}} \leftarrow \text{diag}(\boldsymbol{\alpha}^r \odot \boldsymbol{\beta}^r) \, \text{Sign}(G) \, \text{diag}(\boldsymbol{\alpha}^c \odot \boldsymbol{\beta}^c).$$

**return** $\widehat{G}_{\text{NGES}}$

---

quickly (empirically $T \leq 15$ is sufficient for common MoE-based LLMs gate sizes). (3) The small constant $\varepsilon$ and ridge $\lambda$ prevent pathological updates when some rows/columns have near-zero energy.

## A.4   MORE EXPERIMENTAL RESULTS

### A.4.1   MORE RESULTS

To fully validate the effectiveness and practical advantages of BiMoE, we expand our experimental evaluation beyond the original 1-bit quantization baselines to include a more comprehensive set of competitors, aiming to provide a rigorous and holistic performance comparison(Table 7). Specifically, our expanded evaluation encompasses two categories of state-of-the-art baselines: 1) practical low-bit post-training quantization (PTQ) methods, including 3-bit and 2-bit configurations of AWQ and GPTQ, which are widely adopted in real-world deployment scenarios; and 2) MoE-specific mixed-precision quantization approaches, such as MoEQuant, QuantMoE-Bench, and Mx-MoE, which are tailored for the unique expert-parallel structure of MoE models.

The results of the expanded experiments reveal several important insights. First, when compared to general low-bit PTQ methods, BiMoE demonstrates remarkable performance superiority: it sig-

Table 7: We evaluate the following datasets: Arc-Challenge (AC), Arc-Easy (AE), HellaSwag (HS), LAMBADA-openai (LO), LAMBADA-standard (LS), PIQA (PQ), and WinoGrande (WG). #Bits indicates the average weight bitwidth of activated parameters during inference. Note that the global average bitwidth of the entire model is around **1.11 bits**.

| Model | Method | #Bits(W) | AE↑ | AC↑ | HS↑ | LO↑ | LS↑ | PQ↑ | WG↑ | Avg.[7]↑ |
|---|---|---|---|---|---|---|---|---|---|---|
| Qwen1.5-MoE | Baseline | 16 | 76.77 | 49.23 | 78.21 | 70.83 | 65.57 | 79.76 | 68.82 | 69.88 |
| | AWQ | 3 | 30.18 | 26.37 | 30.41 | 3.29 | 0.18 | 50.33 | 50.12 | 27.27 |
| | GPTQ | 3 | 52.77 | 29.18 | 49.61 | 46.18 | 35.47 | 71.29 | 59.99 | 49.21 |
| | AWQ | 2 | 26.05 | 28.07 | 25.43 | 0 | 0 | 50.92 | 50.91 | 25.91 |
| | GPTQ | 2 | 24.45 | 26.45 | 25.97 | 0 | 0 | 51.41 | 50.43 | 25.53 |
| | MoEQuant | 2 | 34.54 | 35.09 | 35.55 | 12.03 | 8.05 | 59.08 | 58.21 | 34.64 |
| | NoWag | 2.08 | **58.16** | **34.64** | **63.35** | 28.90 | 22.92 | **71.82** | 55.09 | 47.84 |
| | QuantMoe-Bench | 2.3 | 40.13 | 26.30 | 48.72 | 20.12 | 16.77 | 59.47 | 52.40 | 37.74 |
| | MxMoE | 2.25 | 53.28 | 31.66 | 62.80 | **56.43** | **51.00** | 71.33 | **61.25** | **55.39** |
| | **BiMoE** | **1.35** | 54.08 | 29.35 | 48.67 | 50.18 | 38.40 | 70.08 | 59.75 | 50.07 |
| Mixtral | Baseline | 16 | 69.15 | 44.45 | 77.23 | 71.40 | 64.49 | 80.36 | 68.75 | 67.98 |
| | AWQ | 3 | 32.25 | 28.51 | 34.46 | 5.21 | 3.93 | 54.59 | 54.98 | 30.56 |
| | GPTQ | 3 | 70.33 | 41.38 | 66.18 | 62.39 | 54.19 | 75.90 | 66.28 | 62.38 |
| | AWQ | 2 | 25.35 | 28.41 | 25.81 | 0 | 0 | 49.78 | 50.08 | 25.63 |
| | GPTQ | 2 | 26.22 | 29.10 | 26.88 | 0 | 0 | 50.16 | 50.33 | 26.10 |
| | MoEQuant | 2 | 49.85 | 38.93 | 40.12 | 22.98 | 18.94 | 60.38 | 49.91 | 40.16 |
| | NoWag | 2.08 | 72.73 | 44.62 | 70.08 | 50.46 | 36.70 | 74.42 | 65.51 | 59.22 |
| | QuantMoE-Bench | 2.3 | 56.87 | 39.90 | 52.80 | 34.74 | 30.11 | 61.59 | 57.85 | 47.69 |
| | MxMoE | 2.25 | **72.77** | **48.98** | **77.44** | 68.68 | **62.18** | 76.28 | **68.90** | **67.89** |
| | **BiMoE** | **1.60** | 69.57 | 42.57 | 65.41 | **71.05** | 58.24 | **76.42** | 68.09 | 64.48 |

nificantly outperforms all 2-bit AWQ and GPTQ baselines across all evaluation metrics, and even achieves slightly better results than 3-bit GPTQ—while consuming less than half the memory footprint of these low-bit baselines. Second, against MoE-specific quantization methods, BiMoE maintains a clear advantage: it achieves over 40% higher average accuracy compared to MoEQuant and QuantMoE-Bench, consistently surpassing their 2-bit configurations on every benchmark task. Third, regarding MxMoE—a mixed-precision method optimized for hardware compatibility—it is worth noting that MxMoE does not consider quantization cost efficiency: it requires exhaustive search over all possible quantization schemes for each expert and each linear layer, leading to GPU-hour costs that are several orders of magnitude higher than BiMoE. In contrast, BiMoE retains over 91% of MxMoE's performance while offering substantial improvements in efficiency: it eliminates the need for time-consuming scheme search and produces a binary model that is only 60% the size of the MxMoE-quantized model.

Collectively, these expanded experimental results confirm that BiMoE not only establishes state-of-the-art performance among binary quantization methods but also outperforms or closely matches strong 2–3 bit baselines—while providing unparalleled advantages in memory usage, quantization efficiency, and deployment cost. This validates the practical value of BiMoE for resource-constrained environments where both performance and efficiency are critical requirements.

### A.4.2 DETAILED RESULTS FOR CHALLENGING TASKS

In this section, to better demonstrate the capability of models binarized using our method, we include more challenging benchmarks such as the multi-domain knowledge understanding task MMLU, the mathematical reasoning tasks MathQA and GSM8K, and the code generation tasks MBPP and HumanEval, which collectively provide a more comprehensive evaluation of our method's effectiveness across diverse domains. As shown in Table 8, results on the five tasks demonstrate that BiMoE consistently outperforms other low-bit quantization methods. Notably, even MoEQuant, which is

Table 8: Results of AWQ and GPTQ with 3/2-bit weight quantization, MoEQuant and NoWag with 2-bit quantization, and BiLLM, ARB-LLM, and BiMoE with 1-bit quantization on five tasks for OLMoE and Qwen1.5-MoE.

| Model | Method | #Bits(W) | MMLU↑ | MathQA↑ | GSM8K↑ | MBPP↑ | HumanEval↑ | Avg.[5]↑ |
|---|---|---|---|---|---|---|---|---|
| OLMoE | Baseline | 16 | 53.40 | 28.41 | 52.84 | 21.80 | 10.98 | 33.45 |
| | AWQ | 3 | 8.30 | 6.31 | 4.77 | 0 | 0 | 3.88 |
| | GPTQ | 3 | 43.08 | 19.88 | 42.03 | 12.00 | 7.28 | 24.85 |
| | AWQ | 2 | 5.21 | 1.09 | 3.21 | 0 | 0 | 1.90 |
| | GPTQ | 2 | 14.78 | 7.28 | 8.30 | 0 | 0 | 6.07 |
| | MoEQuant | 2 | 20.15 | 10.41 | 17.88 | 0 | 0 | 9.69 |
| | NoWag | 2.04 | 24.46 | 12.20 | 21.14 | 1.02 | 0.02 | 11.77 |
| | BiLLM | 1.11 | 28.01 | 15.18 | 27.21 | 3.41 | 0.42 | 14.85 |
| | ARB-LLM | 1.11 | 30.17 | 16.32 | 30.02 | 6.32 | 2.01 | 16.97 |
| | **BiMoE** | **1.46** | **42.33** | **20.02** | **41.88** | **12.12** | **6.33** | **24.54** |
| Qwen1.5-MoE | Baseline | 16 | 60.87 | 35.34 | 61.33 | 38.20 | 34.76 | 46.10 |
| | AWQ | 3 | 28.30 | 16.35 | 14.79 | 6.31 | 9.73 | 15.10 |
| | GPTQ | 3 | 49.11 | 29.75 | 52.01 | 30.18 | 27.05 | 37.48 |
| | AWQ | 2 | 15.22 | 11.05 | 4.21 | 0 | 0 | 12.10 |
| | GPTQ | 2 | 26.94 | 19.33 | 15.42 | 9.24 | 13.88 | 16.96 |
| | MoEQuant | 2 | 34.75 | 22.42 | 23.21 | 14.32 | 18.12 | 22.56 |
| | NoWag | 2.04 | 38.44 | 23.73 | 35.54 | 16.76 | 20.23 | 26.94 |
| | BiLLM | 1.11 | 39.01 | 25.15 | 38.29 | 19.41 | 22.12 | 28.80 |
| | ARB-LLM | 1.11 | 40.19 | 26.77 | 40.32 | 21.88 | 23.86 | 30.60 |
| | **BiMoE** | **1.35** | **48.72** | **30.25** | **52.33** | **29.27** | **27.12** | **37.54** |

specifically designed for MoE-based LLMs, collapses severely under 2-bit quantization. In contrast, BiMoE outperforms both the current state-of-the-art binary methods and the 2-bit baselines across all benchmarks. Our method achieves performance comparable to 3-bit GPTQ, while using less than half of its memory footprint. Whereas other low-bit quantization methods substantially degrade the model's reasoning capability, integrating BiMoE effectively preserves such ability in generative tasks—showing less than a 10-point accuracy drop compared with the full-precision model. This is particularly important for complex reasoning benchmarks such as HumanEval, highlighting the practical value and robustness of BiMoE.

### A.4.3 EXPERIMENTS OF INSTRUCTION-TUNED MODELS

Instruction fine-tuning can significantly improve the application capabilities of the model and has become a necessary process for deployment of LLMs in different scenarios. The quantization of instruction-tuned models is often more challenging than that of base models. We perform benchmark tests on Qwen-MoE-14B-Chat and DeepSeekMoE-16B-Chat, covering three tasks. For Qwen-MoE-14B-Chat, BiMoE consistently preserves around 80% of the full-precision performance, effectively recovering most of the model's original reasoning capability. As shown in Table 9, previous methods suffer from severe accuracy degradation on instruction-tuned models for code generation and mathematical reasoning tasks. For instance, when quantized to 2-bit, both GPTQ and MoEQuant cause Qwen-MoE-14B-Chat to completely collapse on HumanEval. In contrast, BiMoE exhibits only a modest drop—less than 28%—demonstrating its superior robustness under extreme compression.

### A.4.4 ABLATION STUDY ON ROUTING SIMILARITY

To systematically evaluate the effectiveness of our proposed **NGES** in suppressing post-quantization *expert-shift*, we conduct a layer-wise routing similarity analysis on Qwen1.5-MoE-A2.7B.

Table 9: Results of GPTQ, MoEQuant, and NoWag with 2-bit weight quantization, and BiLLM, ARB-LLM, and BiMoE with 1-bit weight quantization on five tasks using Qwen and DeepSeek MoE instruction-tuned models.

| Model | Method | #Bits(W) | MMLU↑ | GSM8K↑ | HumanEval↑ | Avg.[3]↑ |
|---|---|---|---|---|---|---|
| | Baseline | 16 | 59.00 | 30.71 | 21.34 | 37.02 |
| | GPTQ | 2 | 27.30 | 3.11 | 0 | 10.14 |
| | MoEQuant | 2 | 32.77 | 6.33 | 0 | 13.03 |
| Qwen-MoE-14B-Chat | NoWag | 2.04 | 36.99 | 6.46 | 3.13 | 15.53 |
| | BiLLM | 1.11 | 38.72 | 10.76 | 9.02 | 19.50 |
| | ARB-LLM | 1.11 | 39.11 | 13.33 | 10.20 | 20.88 |
| | **BiMoE** | 1.35 | **49.22** | **22.18** | **15.75** | **29.08** |
| | Baseline | 16 | 48.90 | 54.28 | 24.39 | 42.52 |
| | GPTQ | 2 | 15.49 | 1.90 | 0 | 5.80 |
| | MoEQuant | 2 | 20.88 | 8.22 | 2.31 | 10.47 |
| DeepSeekMoE-16B-Chat | NoWag | 2.04 | 23.01 | 16.77 | 6.83 | 15.54 |
| | BiLLM | 1.11 | 30.18 | 30.87 | 17.20 | 26.08 |
| | ARB-LLM | 1.11 | 33.21 | 35.11 | 19.31 | 29.21 |
| | **BiMoE** | 1.46 | **40.06** | **45.00** | **25.66** | **36.91** |

**Metric.** Given the router logits $\mathbf{z} \in \mathbb{R}^E$ for $E$ experts, we first apply the softmax transformation to obtain the routing probability vector

$$\mathbf{p} = \mathrm{softmax}(\mathbf{z}) \in \Delta^{E-1}, \tag{53}$$

where $\Delta^{E-1}$ denotes the $(E-1)$-dimensional probability simplex. For each token $t$, let $\mathbf{p}_{\mathrm{FP}}^{(t)}$ and $\mathbf{p}_{\mathrm{Q}}^{(t)}$ denote the routing probability vectors of the full-precision (FP) and quantized (Q) models, respectively. We define the token-level routing similarity as the cosine similarity:

$$s^{(t)} = \frac{\left\langle \mathbf{p}_{\mathrm{FP}}^{(t)}, \mathbf{p}_{\mathrm{Q}}^{(t)} \right\rangle}{\left\| \mathbf{p}_{\mathrm{FP}}^{(t)} \right\|_2 \left\| \mathbf{p}_{\mathrm{Q}}^{(t)} \right\|_2}. \tag{54}$$

Finally, the layer-wise similarity score is obtained by averaging across all tokens in the evaluation set:

$$S_\ell = \frac{1}{T} \sum_{t=1}^{T} s^{(t)}, \tag{55}$$

where $T$ is the total number of tokens. Higher $S_\ell$ indicates stronger consistency between the routing behaviors of quantized and FP models.

**Results and analysis.** We use the WikiText-2 dataset as the evaluation corpus. For each layer $\ell$, we compare the similarity scores $S_\ell$ of three quantization baselines (BiLLM, ARB-LLM, and our BiMoE with NGES) against the original FP model. As shown in Figure 7, BiLLM and ARB-LLM suffer from progressive degradation with depth, leading to noticeable distortions in routing behavior. In contrast, BiMoE consistently achieves higher routing similarity across all layers, on average outperforming prior methods by $\sim 0.05$. Moreover, even in the worst-performing layer, the similarity drop relative to FP is limited to $\sim 0.05$, highlighting the robustness of NGES in preserving routing stability.

These results clearly demonstrate that NGES is critical for mitigating expert-shift under quantization. By ensuring stable routing distributions, BiMoE not only preserves expert utilization patterns but also underpins the downstream zero-shot improvements reported in the main paper.

### A.4.5 ABLATION STUDY ON CALIBRATION DATA

We further investigated the impact of calibration data on BiMoE. Specifically, when binarizing Qwen3-30B-A3B on WikiText-2, we fixed all quantization hyperparameters and varied the cali-

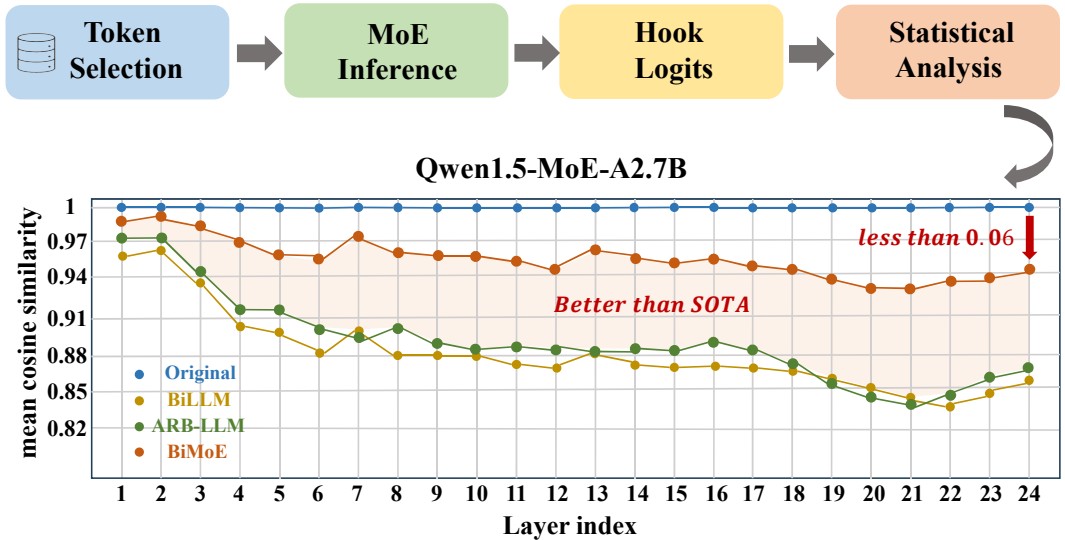

Figure 7: Layer-wise Comparison of Expert Activation Pattern Similarity between Full-Precision and Binary Models on Qwen1.5-MoE-A2.7B.

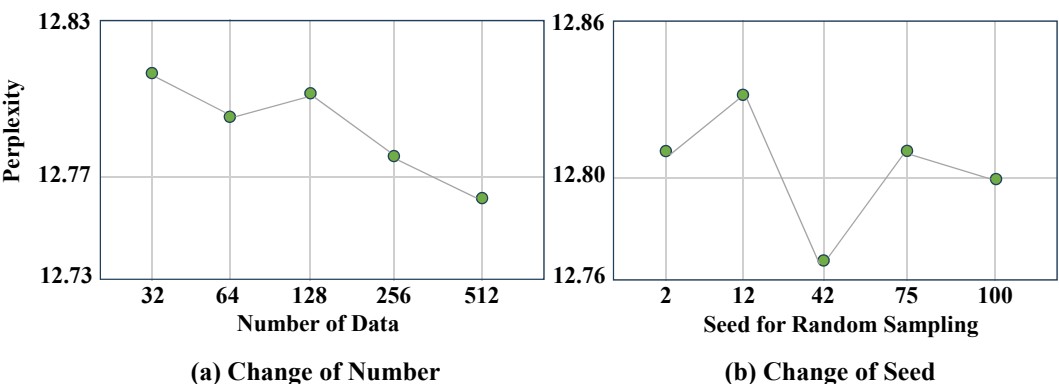

(a) Change of Number

(b) Change of Seed

Figure 8: Perplexity of Qwen3-30B-A3B using calibration data sampled with different number or seeds from WikiText2.

bration set by (i) subsampling different numbers of calibration samples and (ii) resampling with multiple random seeds that control both sample selection and token order. For each setting, we re-quantized once and evaluated perplexity on a held-out split. Across all sizes and seeds, the resulting perplexity fluctuates by $<1\%$ relative to the mean. As shown in Figure 8, the curves remain nearly flat as calibration size increases, and seed-wise traces largely overlap, indicating that even our smallest tested calibration subsets perform on par with larger ones. These observations confirm that BiMoE is highly robust to calibration data selection.

### A.4.6 ABLATION STUDY ON REGULARIZATION COEFFICIENT

To achieve the best performance under our framework, we carefully study the choice of the regularization coefficient $\lambda$ in the closed-form solution of NGES projection vectors. As shown in Table 10, we adopt a coordinate descent strategy to search for the optimal hyperparameter within the range $[0.1, 0.9]$. After extensive experiments across different MoE-based LLMs and baselines, we empirically find that $\lambda = 0.2$ yields consistently strong performance in most cases. We also explored making this hyperparameter learnable during optimization, but the results were inferior to either coordinate descent or grid search. We acknowledge that such manual hyperparameter tuning remains a limitation of the current method.

Table 10: Selection of the NGES regularization coefficient $\lambda$, evaluated on OLMoE-1B-7B-0125 quantized by BiMoE. Results are reported as perplexity on WikiText2, C4, and PTB. Red, green, and blue numbers indicate the $first-$, $second-$, and $third-best$ results on each calibration dataset, respectively. Overall, $\lambda = 0.2$ achieves two firsts and one second, making it the optimal choice.

| $\lambda$ | Wikitext2 | | | | C4 | | | | PTB | | | |
|---|---|---|---|---|---|---|---|---|---|---|---|---|
| | Wikitext2 | C4 | PTB | Avg. | Wikitext2 | C4 | PTB | Avg. | Wikitext2 | C4 | PTB | Avg. |
| 0.1 | 14.63 | 26.86 | 34.40 | 25.30 | 19.58 | 24.09 | 34.70 | 26.12 | 19.26 | 28.66 | 28.68 | 25.53 |
| **0.2** | 14.37 | 25.91 | 33.53 | 24.60 | 18.48 | 23.01 | 33.59 | 25.03 | 18.57 | 27.58 | 28.86 | 24.67 |
| 0.3 | 14.61 | 26.14 | 34.69 | 25.15 | 18.69 | 23.60 | 32.90 | 25.06 | 18.51 | 29.61 | 29.33 | 25.82 |
| 0.4 | 14.67 | 26.95 | 36.75 | 26.12 | 18.73 | 24.05 | 34.03 | 25.60 | 18.18 | 26.37 | 28.68 | 24.41 |
| 0.5 | 14.56 | 26.43 | 34.49 | 25.16 | 19.55 | 23.89 | 36.32 | 26.59 | 19.01 | 27.69 | 28.66 | 25.12 |
| 0.6 | 14.62 | 26.88 | 35.08 | 25.53 | 19.33 | 23.61 | 33.32 | 25.42 | 18.40 | 27.23 | 28.74 | 24.82 |
| 0.7 | 14.75 | 27.21 | 37.27 | 26.41 | 19.34 | 23.70 | 32.94 | 25.33 | 19.30 | 28.09 | 29.25 | 25.55 |
| 0.8 | 14.52 | 26.50 | 35.47 | 25.50 | 19.55 | 24.27 | 33.43 | 25.75 | 18.42 | 27.60 | 28.82 | 24.95 |
| 0.9 | 14.71 | 27.15 | 36.88 | 26.25 | 19.47 | 23.97 | 35.86 | 26.43 | 19.18 | 28.43 | 28.79 | 25.47 |

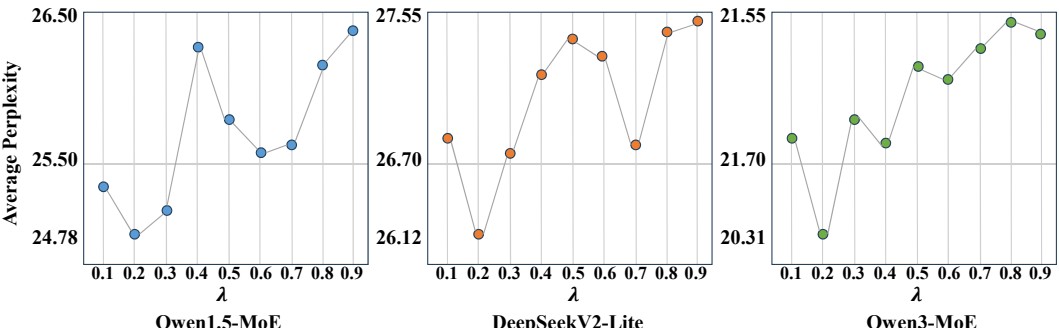

Figure 9: Average perplexity under different values of the regularization coefficient $\lambda$ on WikiText2, C4, and PTB datasets for three MoE-based LLMs: Qwen1.5-MoE, DeepSeekV2-Lite, and Qwen3-MoE. Across all architectures, $\lambda = 0.2$ achieves the lowest average perplexity, confirming its strong generalization capability.

Furthermore, to demonstrate the generalization capability of this manually selected hyperparameter across different architectures, we present in Figure 9 the average perplexity on WikiText2, C4, and PTB under varying $\lambda$ values for Qwen1.5-MoE, DeepSeekV2-Lite, and Qwen3-MoE. The results show that $\lambda = 0.2$ consistently achieves the best performance across all model architectures.

### A.4.7 PLUG-AND-PLAY ABLATIONS OF GLAS AND NGES

To assess the plug-and-play nature and generality of our two components, GLAS (global loss-aligned saliency) and NGES (MoE-aware gating/expert stabilization), we integrate them into two representative PTQ backbones—BiLLM and ARB-LLM—without altering any other quantization settings. We evaluate perplexity (lower is better) on three corpora (WikiText-2, C4, PTB) across three MoE model families (OLMoE-1B-7B, Qwen1.5-MoE-A2.7B, DeepSeekV2-Lite). For each backbone, we report three configurations: the original baseline, +GLAS, and +GLAS+NGES. Calibration data, tokenization, and inference hyperparameters are held fixed; no fine-tuning is performed.

The results in Fig. 10 (BiLLM series) and Fig. 11 (ARB-LLM series) show a clear and consistent trend. First, adding GLAS alone yields uniform perplexity drops on all three datasets and all three model families, indicating that loss-aligned saliency better preserves task-critical weights under PTQ. Second, further appending NGES typically brings additional gains, especially on C4 and PTB, where MoE routing distortions are more detrimental; NGES mitigates these distortions

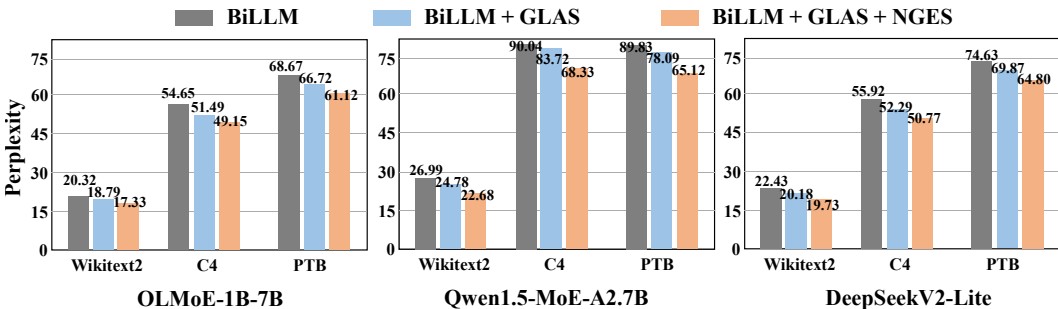

Figure 10: Plug-and-play ablation on the BiLLM backbone. Perplexity on WikiText-2, C4, PTB for OLMoE-1B-7B, Qwen1.5-MoE-A2.7B, and DeepSeekV2-Lite. Bars denote BiLLM (gray), BiLLM + GLAS (blue), and BiLLM + GLAS + NGES (orange). Adding GLAS consistently lowers perplexity, and stacking NGES brings further reductions in most settings, evidencing good compatibility and complementarity.

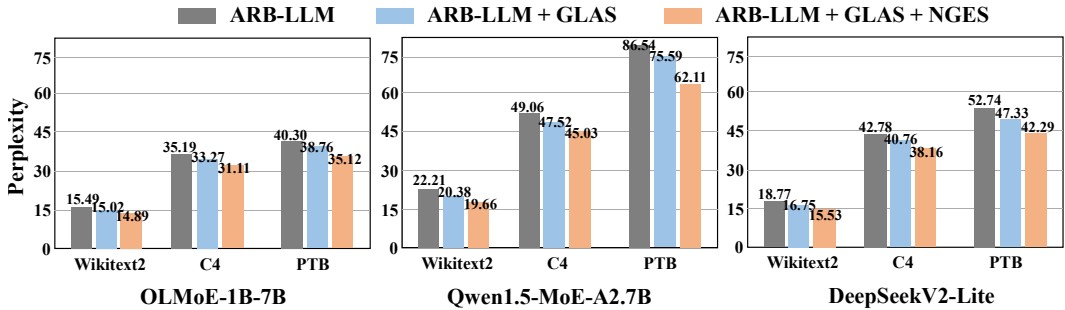

Figure 11: Plug-and-play ablation on the ARB-LLM backbone. Same evaluation as Fig. 9, replacing the backbone with ARB-LLM. ARB-LLM + GLAS (blue) generally outperforms the baseline (gray), and ARB-LLM + GLAS + NGES (orange) achieves additional decreases in most model–dataset pairs, confirming seamless integration and robust gains on a second, independent PTQ pipeline.

by stabilizing expert/gate behavior under quantization. The improvement pattern is largely monotonic—baseline → +GLAS → +GLAS+NGES—across backbones and corpora, with only minor fluctuations on a few grid points that do not change the overall conclusion.

These ablations demonstrate that GLAS and NGES are drop-in, training-free, and backbone-agnostic: they integrate seamlessly into existing PTQ pipelines and systematically reduce perplexity for MoE LLMs. Beyond confirming compatibility with BiLLM and ARB-LLM, the consistency across WikiText-2, C4, and PTB suggests that our components capture model- and data-independent failure modes of MoE quantization, providing complementary benefits to standard PTQ design.

## A.5 DIALOG EXAMPLES

To further illustrate the qualitative impact of different binarization schemes, Table 11 presents representative generations from four MoE-based LLMs—OLMoE-1B-7B, DeepSeek-V2-Lite, Qwen1.5-MoE-A2.7B, and Qwen3-30B-A3B—in their full-precision form as well as after compression by BiLLM, ARB-LLM, and our proposed BiMoE. As can be observed, when BiLLM or ARB-LLM is applied to MoE-based LLMs, the resulting outputs almost completely lose coherence: sentences often degenerate into meaningless repetitions, truncated fragments, or irrelevant statements, indicating a collapse of the generative ability. In stark contrast, BiMoE maintains sentence fluency and semantic relevance. The compressed models not only produce complete and coherent responses but also preserve the essential informational content, achieving a generation quality that is close to that of their full-precision counterparts. This qualitative evidence corroborates the quantitative results, highlighting that BiMoE effectively prevents the catastrophic degradation observed in prior binary PTQ approaches and enables practical deployment of MoE-based LLMs under extreme compression.

To complement the qualitative analysis, we further conduct a systematic quantitative evaluation of generative quality to directly assess whether the outputs of quantized models "make sense" in practical scenarios. Specifically, we use GPT-5 to construct a benchmark consisting of 500 everyday questions covering common knowledge, daily communication, and practical problem-solving. All quantized models (including baselines and our BiMoE) are required to generate responses with a maximum length of 1024 tokens, and GPT-5 then rates each response on a 1–10 scale based on four key metrics: linguistic fluency, semantic coherence, logical consistency, and factual correctness. Table 12 summarizes the evaluation results for OLMoE and Qwen1.5-MoE across different quantization methods.

The results demonstrate that BiMoE achieves remarkable generative quality preservation: it retains over 90% of the full-precision model's score (8.1 for OLMoE and 8.6 for Qwen1.5-MoE), significantly outperforming all other binary and low-bit baselines. In contrast, most competing methods suffer severe performance degradation or even complete generative collapse: 3-bit AWQ only achieves scores of 2.3 (OLMoE) and 2.6 (Qwen1.5-MoE), while 2-bit methods (including AWQ, GPTQ, and MoEQuant) all score below 1.5, indicating their outputs are largely incoherent or meaningless. Even other 1-bit/near-1-bit methods (e.g., NoWag, BiLLM, ARB-LLM) only reach scores between 4.3 and 5.5, which are far lower than BiMoE. Notably, 3-bit GPTQ performs relatively better among low-bit baselines (7.5 for OLMoE and 7.9 for Qwen1.5-MoE) but still lags behind BiMoE, while BiMoE maintains comparable generative quality with a much lower bit-width (1.46 bits) and smaller memory footprint.

This quantitative evaluation further validates that BiMoE effectively mitigates the catastrophic degradation of generative ability observed in prior binary quantization approaches. Even under extreme 1-bit compression, BiMoE ensures that MoE-based LLMs can still produce fluent, coherent, and factually accurate responses, which is critical for practical deployment scenarios where both compression efficiency and generative quality are indispensable.

## A.6 Use of Large Language Models

In preparing this manuscript, we employed large language models (LLMs) as writing assistants. Specifically, LLMs were used for grammar correction, wording improvement, and stylistic polishing of the text. In certain cases, LLMs were also leveraged to rephrase or restructure preliminary drafts of specific sections to improve clarity and readability. Importantly, all conceptual contributions, technical methods, experimental designs, and analyses were conceived and developed by the authors without reliance on LLMs. The final responsibility for the accuracy and integrity of the content rests entirely with the authors.

## A.7 Limitations

While BiMoE provides clear benefits in accuracy preservation and efficiency, certain limitations remain. The framework introduces additional steps such as decomposition and saliency estimation, which may increase the complexity of implementation compared with uniform quantization. In addition, the quality of saliency-guided binarization depends on the calibration dataset; if there is a distribution shift between calibration and deployment, some degradation may occur. Finally, although our null-space correction alleviates routing instability, the MoE-based LLMs is inherently sensitive to quantization noise, and rare cases of expert misallocation may still arise. These limitations open potential directions for further refinement, including more lightweight saliency estimation and adaptive calibration strategies.

Table 11: We select OLMoE, Deepseek-V2-Lite, Qwen1.5-MoE, and Qwen3-MoE to demonstrate their language completion and question-answering capabilities, with BiLLM and ARB-LLM used as comparison models. The **input** is marked in bold and the normal texts are the generated sentences and we color the text to indicate whether the responses are reasonable or inappropriate.

| OLMoE-1B-7B-0125 | |
|---|---|
| Original | **What are the capitals of the United States, the United Kingdom, and France respectively?** Washington D.C., London, Paris. |
| BiLLM | **What are the capitals of the United States, the United Kingdom, and France respectively?** The capitals of the United States, the United Kingdom, and France. |
| ARB-LLM | **What are the capitals of the United States, the United Kingdom, and France respectively?** The capitals of these three countries. |
| **BiMoE** | **What are the capitals of the United States, the United Kingdom, and France respectively?** The capital of the United States is Washington DC. The capital of the United Kingdom is London. The capital of France is Paris. |
| Qwen1.5-MoE-A2.7B | |
| Original | **I have a friend who lives in Beijing and I would like to visit him. How would I go there from New York?** You can fly from New York to Beijing, but you could also take a train or drive if you wanted to. |
| BiLLM | **I have a friend who lives in Beijing and I would like to visit him. How would I go there from New York?** What do I want to do is go there? He doesn't want to do is stay here. Why did you do was doing is going to do is staying here. Why did you do was done is doing is did. Did you do did was done did do did do did did did . . . |
| ARB-LLM | **I have a friend who lives in Beijing and I would like to visit him. How would I go there from New York?** A. 10 hours. B. 96 hours C. 23 hours D. 17 hours. Answer B. |
| **BiMoE** | **I have a friend who lives in Beijing and I would like to visit him. How would I go there from New York?** The answer is that you need to find a way to travel between two cities. The first city is called "Beijing" and the second city is called "New York". You can get on a plane and fly to Beijing from New York. |
| DeepSeekV2-Lite | |
| Original | **Who were the first three presidents of the United States?** The first three U.S. presidents were George Washington, John Adams and Thomas Jefferson. |
| BiLLM | **Who were the first three presidents of the United States?** The...Ss,s,s,s,s,s,s,s,s,s,s,s,s,s,s,o,o,o,o,o,o,o,o,o,o,o,o,o,o,o,o,o,o,o,o.).t,s,s,s,s,o,o . . . |
| ARB-LLM | **Who were the first three presidents of the United States?** 50 Years Old And Now Have Been A Serious Moment. |
| **BiMoE** | **Who were the first three presidents of the United States?** The first President of the United States is George Washington. The second President of the United States is John Adams. The third President of the United States is Thomas Jefferson. |
| Qwen3-30B-A3B | |
| Original | **How should I adjust if I didn't perform well in my final exams? Please give me a short answer.** If you didn't do well in your exams, reflect on what went wrong, seek help, and revise strategically. Prepare by starting early, using active learning, and managing your time. Improve memory through practice, good habits, and mnemonics. Effective techniques include spaced repetition, active recall, and interleaving. Stay focused and keep going—you can improve! |
| BiLLM | **How should I adjust if I didn't perform well in my final exams? Please give me a short answer.** Let me explain the situation. So, I want to explain the situation. Let me explain the situation. So, I want to explain the situation . . . |
| ARB-LLM | **How should I adjust if I didn't perform well in my final exams? Please give me a short answer.** (The answer is 10-20 seconds) [please] [give me] [the answer] Please [please] . . . |
| **BiMoE** | **How should I adjust if I didn't perform well in my final exams? Please give me a short answer.** Assess your study methods, identify areas needing improvement, and seek guidance from teachers. |

Table 12: Quantitative evaluation of generative quality (1–10 scale) assessed by GPT-5. Higher scores indicate better linguistic fluency, semantic coherence, logical consistency, and factual correctness.

| Model | Method | #bits(W) | Score↑ |
|---|---|---|---|
| OLMoE | Baseline | 16 | **9.2** |
| | AWQ | 3 | 2.3 |
| | GPTQ | 3 | 7.5 |
| | AWQ | 2 | 1.1 |
| | GPTQ | 2 | 1.0 |
| | MoEQuant | 2 | 1.2 |
| | NoWag | 2.04 | 4.3 |
| | BiLLM | 1.11 | 4.5 |
| | ARB-LLM | 1.11 | 5.3 |
| | **BiMoE** | **1.46** | **8.1** |
| Qwen1.5-MoE | Baseline | 16 | **9.5** |
| | AWQ | 3 | 2.6 |
| | GPTQ | 3 | 7.9 |
| | AWQ | 2 | 1.2 |
| | GPTQ | 2 | 1.5 |
| | MoEQuant | 2 | 1.3 |
| | NoWag | 2.04 | 4.3 |
| | BiLLM | 1.11 | 4.9 |
| | ARB-LLM | 1.11 | 5.5 |
| | **BiMoE** | **1.46** | **8.6** |

