# OpenReview forum: "BiMoE：Pushing the Limit of Post-Training Quantization for MoE-based LLMs"
_ICLR.cc/2026/Conference — ICLR 2026 Conference Desk Rejected Submission_

### Official Review · Reviewer_BkZp · 2025-10-23

**Soundness:** 2
**Presentation:** 3
**Contribution:** 3
**Rating:** 6
**Confidence:** 3

**Summary:**

This paper introduces a binarization framework tailored for MoE models called BiMoE. It identifies three major challenges unique to MoE binarization, including 1) cross-expert redundancy, 2) task-unaware importance estimation, and 3) quantization-induced expert shift, and addresses them through three key components: 1) Cross-Expert Joint Decomposition (CEJD), which employs shared SVD to reduce redundant expert parameters; 2) Global Loss-Aligned Saliency (GLAS), which integrates global loss gradients with local Hessians for task-aware weight importance; and 3) Null-space Guided Expert-Shift Suppression (NGES), which constrains quantization errors within routing-insensitive subspaces. Experiments on multiple open-source MoE LLMs and several zero-shot evaluation sets show that BiMoE could achieve over 2x speedup, and remarkably better algorithm performances compared with other baseline MoE binarization methods.

**Strengths:**

- The paper provides novel and well-supported empirical observations on the unique challenges of binarizing MoE-based LLMs, such as expert redundancy and expert-shift. These analyses are both insightful and foundational, offering valuable guidance for future research on quantization for MoE models
- The proposed BiMoE framework is theoretically grounded and technically rich, integrating principled formulations (joint SVD, gradient-aligned Hessian, and null-space projection) into a coherent and elegant solution. The method is well-presented, with clear motivation, derivations, and intuition throughout.

**Weaknesses:**

While the proposed method is theoretically well-founded and demonstrates strong results on standard benchmarks, the evaluation scope is limited to relatively simple and dated zero-shot tasks (e.g., HellaSwag, LAMBADA, PIQA) with limited context length (4096). These benchmarks, while conventional, do not fully capture the diverse behaviors of quantized MoE-based LLMs. Previous work [[1]](https://arxiv.org/abs/2402.18158) has shown that **quantization effects vary significantly across task types**, and issues may emerge in more challenging settings such as long-context understanding, multi-turn dialogue, or mathematical reasoning, which are not covered in this work. Consequently, the current experimental evaluation may underestimate potential weaknesses of the proposed method, and a more comprehensive assessment across advanced tasks would be necessary to convincingly demonstrate its robustness and general applicability.

**Questions:**

Could you further provide additional evaluation results on more advanced tasks (like long-context tasks and reasoning tasks), to justify the proposed method's general applicability?

---

> ### Author Response · Authors · 2025-11-18
> **Response to Reviewer BkZp Part 1**
>
> We sincerely appreciate your constructive feedback. Below, we provide point-by-point responses, with all revisions incorporated accordingly.
>
> ### Q1
> While the proposed method is theoretically well-founded and demonstrates strong results on standard benchmarks, the evaluation scope is limited to relatively simple and dated zero-shot tasks (e.g., HellaSwag, LAMBADA, PIQA) with limited context length (4096). These benchmarks, while conventional, do not fully capture the diverse behaviors of quantized MoE-based LLMs. Previous work [1] has shown that quantization effects vary significantly across task types, and issues may emerge in more challenging settings such as long-context understanding, multi-turn dialogue, or mathematical reasoning, which are not covered in this work. Consequently, the current experimental evaluation may underestimate potential weaknesses of the proposed method, and a more comprehensive assessment across advanced tasks would be necessary to convincingly demonstrate its robustness and general applicability.
>
> ### A1
> Thank you for the insightful suggestion. We agree that the zero-shot reasoning tasks used in the original submission (e.g., HellaSwag, LAMBADA, PIQA) may not fully capture the diverse behaviors of quantized MoE-based LLMs. Following your recommendation, we have incorporated more challenging benchmarks, **including the multi-domain knowledge task MMLU, the mathematical reasoning tasks MathQA and GSM8K, and the code generation tasks MBPP and HumanEval**, enabling a more comprehensive assessment of our method across different domains.
>
> In addition, we included MoEQuant (a quantization method specifically tailored for MoE architectures) as well as several strong 3-bit and 2-bit baselines to further validate the superiority of our approach. Specifically, we evaluated OLMoE and Qwen1.5-MoE-A2.7B on the five challenging tasks mentioned above. The results show that BiMoE significantly **outperforms existing 1-bit and 2-bit methods across all metrics**, and even **achieves performance comparable to 3-bit GPTQ**, while using **less than half of its memory footprint**.
>
> These new experiments have been added to **Appendix A4.1 (Table 7)**. We sincerely appreciate your suggestion, which has substantially improved the comprehensiveness and quality of our evaluation.
>
> | Model|Method| #Bits(W) | MMLU↑ | MathQA↑ | GSM8K↑ | MBPP↑ | HumanEval↑ | Avg.↑ |
> |---|---|---|---|---|---|---|---|---|
> | **OLMoE**  | Baseline  | 16 | 53.40 | 28.41    | 52.84   | 21.80 | 10.98        | 33.45 |
> |        | AWQ       | 3  | 8.30  | 6.31     | 4.81    | 0.00  | 0.00         | 3.88  |
> |        | GPTQ      | 3  | 43.08 | 19.88    | 42.03   | 12.00 | 7.28         | 24.85 |
> |        | AWQ       | 2  | 5.21  | 7.28     | 8.30    | 0.00  | 0.00         | 4.56  |
> |        | GPTQ      | 2 | 14.78 | 7.28     | 3.30    | 0.00  | 0.00         | 5.07  |
> |        | MoEQuant  | 2  | 20.15 | 10.41    | 17.88   | 0.00  | 0.00         | 9.69  |
> |        | NoWag     | 2  | 4.04  | 12.20    | 21.71   | 2.11  | 0.00         | 7.61  |
> |        | BiLLM     | 1.11  | 28.01 | 15.12    | 27.21   | 3.41  | 1.92         | 15.93 |
> |        | ARB-LLM   | 1.11  | 18.60 | 9.93     | 25.40   | 1.94  | 1.92         | 11.96 |
> | | **BiMoE**| **1.46**| **42.33**| **20.02**| **41.88**| **12.12**| **6.33**| **24.54** |
> | **Qwen1.5-MoE**  | Baseline  | 16        | 60.87 | 35.34    | 65.24   | 30.88 | 19.95        | 42.46 |
> |              | AWQ       | 3         | 28.30 | 16.30    | 15.41   | 0.00  | 0.00         | 12.80 |
> |              | GPTQ      | 3         | 49.11 | 29.75    | 52.01   | 19.22 | 13.26        | 32.27 |
> |              | AWQ       | 2         | 15.22 | 15.22    | 14.51   | 1.94  | 0.00         | 9.78  |
> |              | GPTQ      | 2         | 26.94 | 19.93    | 21.42   | 1.94  | 0.00         | 14.45 |
> |              | MoEQuant  | 2         | 34.75 | 22.42    | 29.74   | 12.44 | 18.12        | 23.49 |
> |              | NoWag     | 2         | 2.04  | 8.34     | 10.22   | 0.00  | 0.00         | 4.52  |
> |              | BiLLM     | 1.11      | 39.01 | 25.15    | 38.29   | 19.14 | 22.12        | 28.74 |
> |              | ARB-LLM   | 1.11      | 41.09 | 22.19    | 40.37   | 18.64 | 20.43        | 28.54 |
> |              | **BiMoE** | **1.35**  | **48.72** | **30.52** | **52.33** | **29.27** | **27.22** | **37.54** |
>
> We have incorporated additional results and detailed analyses addressing your concerns into the revised version of the manuscript. We hope these supplements adequately address your questions and improve the clarity and robustness of our work. Please do not hesitate to let us know if you require further discussion, additional experiments, or any clarifications—we are happy to provide more details to address your concerns comprehensively.

---

> > ### Comment · Reviewer_BkZp · 2025-11-18
> >
> > Thanks for your response. The supplementary results could more comprehensively show the effectiveness of your method, which address my concerns a lot.

---

> > > ### Author Response · Authors · 2025-11-20
> > >
> > > Dear Reviewer BkZp,
> > >
> > > Thank you sincerely for your thoughtful feedback and for letting us know that the additional results addressed your concerns. We truly appreciate your time and constructive insights. If there is anything else you feel would further strengthen the work, we would be very happy to clarify or provide more details. Thank you again for engaging with our submission.

---

### Official Review · Reviewer_bxMB · 2025-10-27

**Soundness:** 3
**Presentation:** 4
**Contribution:** 3
**Rating:** 4
**Confidence:** 5

**Summary:**

This paper presents BiMoE, an MoE-aware compression framework for LLMs that includes post-training binarization. It addresses three MoE-specific challenges: (i) Cross-Expert Joint Decomposition (CEJD) to remove inter-expert redundancy before quantization, (ii) Global Loss-Aligned Saliency (GLAS) for loss-aware calibration, and (iii) Null-space-Guided Expert-Shift Suppression (NGES) to mitigate routing drift due to the quantization error.

**Strengths:**

Clear MoE-specific design that jointly tackles redundancy, saliency misalignment, and routing instability (CEJD+GLAS+NGES).

No fine-tuning required, making the method practical for very large MoE models.

Consistent accuracy improvements over prior 1-bit baselines on the reported zero-shot suite.

System-level reporting (memory reduction, inference throughput) and ablations that isolate each component and show combined benefits.

Generally clear writing and informative figures; easy to follow overall.

**Weaknesses:**

1. **Method relation/novelty:** CEJD appears conceptually close to another SVD-based expert compression work [1]. It will strengthen the paper to clarify the novelty beyond integrating quantization to better highlight the CEJD’s specific contribution.

2. **Effective bitwidth fairness in accuracy comparisons:** With the shared backbone at 8-bit, the effective bitwidth seems to exceed that of 1-bit baselines by a non-negligible margin. To help disentangle accuracy–bitwidth trade-offs, it would be helpful to include configurations with a lower-bit backbone (e.g., 2-bit) that match baseline effective bit budgets; such a matched-budget comparison may be most informative.

3. **Evaluation benchmark:** The results focus on a small zero-shot suite and WikiText-2 perplexity. Given typical MoE use in reasoning/assistant scenarios, task-specific evaluations (math/code/QA; e.g., AIME, GPQA, LiveCodeBench, BFCL) and, where appropriate, instruction-following/chat assessments should be included.

4. **Positioning with respect to minimal-tuning 1-bit methods:** While the method is practical in not requiring fine-tuning, there are binary quantization approaches that allow minimal fine-tuning and report substantially higher scores [2]. Including initialization time and accuracy/latency comparisons against such methods could further substantiate the benefits of the fine-tuning-free setting.

5. **CEJD vs. other SVD-based MoE compression:** Since the quantization methods (GLAS, NGES) can be also implemented to other SVD-based compression that shares a low-rank matrix, a comparison of CEJD with other SVD-based MoE compression methods [3] would have clarify CEJD’s effectiveness and how it integrates with quantization.

[1] Sub-MoE: Efficient Mixture-of-Expert LLMs Compression via Subspace Expert Merging, Lujun Li, Zhu Qiyuan et al., https://arxiv.org/abs/2506.23266.

[2] The Era of 1-bit LLMs: All Large Language Models are in 1.58 Bits, Shuming Ma, Hongyu Wang et al., https://arxiv.org/abs/2402.17764. (or recently, BitNet b1.58 2B4T Technical Report, Shuming Ma, Hongyu Wang et al., https://arxiv.org/abs/2504.12285)

[3] MoE-SVD: Structured Mixture-of-Experts LLMs Compression via Singular Value Decomposition, Wei Li, Lujun Li et al., ICML 2025.

**Questions:**

Mainly listed in the weakness. Below are the additional questions.

1. Have you considered using the backbone as \Simga V and quantizing the U, where the SVD is applied to the row-wise concatenation of weights (W^{(1)}, ... W^{(n)}), as a reverse option compard to sharing U \Sigma and quanting the V?

2. Are the models are calibrated per task within the seven zero-shot benchmarks, or only once on a single corpus in table 3?

3. Related to Weakness 5, what is the wall-clock cost for few dozens of billion parameter models (e.g., Qwen3-30B-A3B, GPT-OSS-20B)?

4. Minor notation: I suggest using \top for transpose instead of T.

---

> ### Author Response · Authors · 2025-11-18
> **Response to Reviewer bxMB Part 1**
>
> We sincerely appreciate your constructive feedback. Below, we provide point-by-point responses, with all revisions incorporated accordingly.
>
> ### Q1
> Method relation/novelty: CEJD appears conceptually close to another SVD-based expert compression work [1]. It will strengthen the paper to clarify the novelty beyond integrating quantization to better highlight the CEJD’s specific contribution.
>
> ### A1
> Regarding the relation between CEJD and SVD-based expert compression methods such as Sub-MoE [1], we sincerely appreciate the reviewer’s observation. While both approaches share the high-level idea of concatenating expert weights and applying SVD, their motivation and operational regimes differ fundamentally:
>
> **1）Motivation.**
>
>  Sub-MoE focuses on structural expert merging: it reduces the number of experts while keeping weights in full precision (or standard low-bit) to improve inference efficiency. In contrast, CEJD does not merge experts or modify the routing structure. Instead, it is specifically designed as a re-parameterization for post-training binarization: CEJD keeps a shared backbone in higher precision and binarizes only the expert-specific projection blocks. The goal is not to “remove experts,” but to preserve expert diversity while confining quantization noise to a stable orthogonal basis, thereby significantly mitigating binarization degradation.
>
> **2）Operational mechanism.**
>
>  Sub-MoE first clusters experts at each layer and then performs joint SVD within each cluster. CEJD, in contrast, performs a single joint SVD per layer over all experts, without clustering or merging, followed by a bit-allocation strategy that assigns high precision to the backbone and 1-bit precision to expert-specific projections.
>
> In summary, the novelty of CEJD does not lie in “using SVD for the first time,” but in being the first to employ SVD to achieve a **‘shared high-precision backbone + binarized projections’ formulation**, together with bit-level compression analysis, specifically for MoE binarization—an unexplored setting in prior work.

---

> > ### Author Response · Authors · 2025-11-18
> > **Response to Reviewer bxMB Part 2**
> >
> > ### Q2
> > Effective bitwidth fairness in accuracy comparisons: With the shared backbone at 8-bit, the effective bitwidth seems to exceed that of 1-bit baselines by a non-negligible margin. To help disentangle accuracy–bitwidth trade-offs, it would be helpful to include configurations with a lower-bit backbone (e.g., 2-bit) that match baseline effective bit budgets; such a matched-budget comparison may be most informative.
> >
> > ### A2
> > Thank you for the valuable suggestion. We appreciate the reviewer’s concern that using an 8-bit shared backbone may lead to an effective bitwidth that is higher than that of prior 1-bit baselines. Following your recommendation, we extended our study to include matched-budget configurations where the **shared backbone is quantized to lower precisions (e.g., 2-bit)**. To ensure a fair comparison, we additionally evaluated **MoEQuant (a quantization method specifically designed for MoE architectures) as well as strong 3-bit and 2-bit baselines**. We conducted experiments on Qwen3-30B-A3B and Mixtral-8×7B across seven datasets: Arc-Challenge (AC), Arc-Easy (AE), HellaSwag (HS), LAMBADA-openai (LO), LAMBADA-standard (LS), PIQA (PQ), and WinoGrande (WG).
> >
> > The results show that even under these much more constrained bit-budget settings (with a 2-bit shared backbone), BiMoE still consistently outperforms the best existing binary methods, achieving over 50% average improvement while **preserving more than 95% of the performance obtained with an 8-bit backbone**. Furthermore, in many cases BiMoE matches or even **slightly surpasses 3-bit GPTQ**, while requiring only about one-third of its memory footprint.
> >
> > These additional experiments greatly strengthen the fairness of our evaluation and further highlight the robustness of our method under strict effective-bitwidth constraints. We sincerely thank the reviewer for this insightful suggestion, which significantly improved the completeness and clarity of our analysis.
> >
> > | model      | method   | #bits(W) | AE    | AC     | HS     | LO    | LS    | PQ     | WG     | Avg    |
> > |------------|----------|----------|-------|--------|--------|-------|-------|--------|--------|--------|
> > | Qwen3-MoE  | AWQ      | 3        | 28.73 | 28.99  | 30.14  | 3.72  | 1.96  | 52.39  | 50.09  | 28.00  |
> > |            | GPTQ     | 3        | 62.03 | 40.88  | 56.14  | 50.32 | 42.79 | 70.89  | 63.33  | 55.20  |
> > |            | AWQ      | 2        | 25.75 | 22.99  | 25.43  | 0     | 0     | 49.93  | 49.58  | 24.84  |
> > |            | GPTQ     | 2        | 25.88 | 26.73  | 26.70  | 0     | 0     | 50.88  | 50.21  | 25.77  |
> > |            | MoEQuant | 2        | 29.81 | 26.59  | 25.38  | 7.93  | 11.78 | 50.74  | 50.41  | 29.94  |
> > |            | NoWag    | 2.08     | 33.08 | 24.49  | 35.74  | 6.79  | 7.35  | 56.86  | 49.96  | 30.61  |
> > |            | BiLLM    | 1.11     | 32.45 | 23.55  | 35.35  | 17.19 | 14.56 | 55.17  | 52.80  | 33.01  |
> > |            | ARB-LLM  | 1.11     | 44.15 | 27.30  | 44.63  | 24.72 | 20.05 | 63.93  | 56.20  | 40.14  |
> > |            | BiMoE    | 1.34     | 66.04 | 43.17  | 58.54  | 52.01 | 45.55 | 71.82  | 65.90  | 57.58  |
> > |            | **BiMoE**| **1.09** | **61.32** | **40.41** | **55.53** | **49.86** | **41.09** | **70.03** | **62.38** | **54.37** |
> > | Mixtral    | AWQ      | 3        | 32.25 | 28.51  | 34.46  | 5.21  | 3.93  | 54.59  | 54.98  | 30.56  |
> > |            | GPTQ     | 3        | 70.33 | 41.38  | 66.18  | 62.39 | 54.19 | 75.90  | 66.28  | 62.38  |
> > |            | AWQ      | 2        | 25.35 | 28.41  | 25.81  | 0     | 0     | 49.78  | 50.08  | 25.63  |
> > |            | GPTQ     | 2        | 26.22 | 29.10  | 26.88  | 0     | 0     | 50.16  | 50.33  | 26.10  |
> > |            | MoEQuant | 2        | 49.85 | 38.93  | 40.12  | 22.98 | 18.94 | 60.38  | 49.91  | 40.16  |
> > |            | NoWag    | 2.08     | 72.73 | 44.62  | 70.08  | 50.46 | 36.70 | 74.42  | 65.51  | 59.22  |
> > |            | BiLLM    | 1.11     | 27.15 | 27.13  | 26.11  | 0.08  | 0.16  | 50.54  | 50.67  | 25.98  |
> > |            | ARB-LLM  | 1.11     | 43.35 | 23.98  | 34.57  | 17.78 | 13.10 | 60.72  | 52.96  | 35.21  |
> > |            | BiMoE    | 1.60     | 69.57 | 42.57  | 65.41  | 71.05 | 58.24 | 76.42  | 68.09  | 64.48  |
> > |            | **BiMoE**| **1.15** | **67.31** | **40.87** | **63.15** | **69.02** | **56.20** | **74.38** | **67.01** | **62.56** |

---

> ### Author Response · Authors · 2025-11-18
> **Response to Reviewer bxMB Part 3**
>
> ### Q3
> Evaluation benchmark: The results focus on a small zero-shot suite and WikiText-2 perplexity. Given typical MoE use in reasoning/assistant scenarios, task-specific evaluations (math/code/QA; e.g., AIME, GPQA, LiveCodeBench, BFCL) and, where appropriate, instruction-following/chat assessments should be included.
>
> ### A3-1
> Thank you for the insightful suggestion. We agree that the zero-shot reasoning tasks used in the original submission (e.g.,  LAMBADA, PIQA) may not fully capture the behaviors of quantized MoE. Following your recommendation, we have incorporated more challenging benchmarks, **including the multi-domain knowledge task MMLU, the mathematical reasoning tasks MathQA and GSM8K, and the code generation tasks MBPP and HumanEval**. In addition, we included MoEQuant (a quantization method specifically tailored for MoE architectures) as well as several strong 3-bit and 2-bit baselines to further validate the superiority of our approach. Specifically, we evaluated OLMoE and Qwen1.5-MoE-A2.7B on the five challenging tasks mentioned above. The results show that BiMoE significantly outperforms existing 1-bit and 2-bit methods, and even achieves performance **comparable to 3-bit GPTQ**, while using less than **half of its memory footprint**.
>
> These new experiments have been added to **Appendix A4.2 (Table 8)**. We sincerely appreciate your suggestion, which has substantially improved the comprehensiveness and quality of our evaluation.
> | Model|Method| #Bits(W) | MMLU↑ | MathQA↑ | GSM8K↑ | MBPP↑ | HumanEval↑ | Avg.↑ |
> |--|-|-|-|-|-|--|-|-|
> | **OLMoE**  | Baseline  | 16 | 53.40 | 28.41| 52.84 | 21.80 | 10.98 | 33.45 |
> | | AWQ |3| 8.30  | 6.31| 4.81 | 0 | 0| 3.88  |
> | | GPTQ |3| 43.08 | 19.88 | 42.03 | 12.00 | 7.28 | 24.85 |
> | | AWQ |2| 5.21  | 7.28 | 8.30| 0 | 0| 4.56 |
> | | GPTQ | 2 | 14.78 | 7.28 | 3.30 | 0 | 0 | 5.07  |
> | | MoEQuant  | 2 | 20.15 | 10.41| 17.88| 0| 0| 9.69 |
> | | NoWag | 2 | 4.04  | 12.20 | 21.71| 2.11  | 0| 7.61 |
> | | BiLLM | 1.11 | 28.01 | 15.12|27.21| 3.41  | 1.92 | 15.93 |
> | | ARB-LLM  | 1.11 | 18.60 | 9.93| 25.40 | 1.94 | 1.92 | 11.96 |
> | | **BiMoE**| **1.46**| **42.33**| **20.02**| **41.88**| **12.12**| **6.33**| **24.54** |
> | **Qwen1.5-MoE**  | Baseline  | 16  | 60.87 | 35.34    | 65.24   | 30.88 | 19.95| 42.46 |
> |  | AWQ | 3 | 28.30 | 16.30 | 15.41|0| 0| 12.80 |
> | | GPTQ | 3  | 49.11 | 29.75 | 52.01 | 19.22 | 13.26 | 32.27 |
> |  | AWQ | 2 | 15.22 | 15.22 | 14.51 | 1.94  |0| 9.78  |
> |  | GPTQ | 2 | 26.94 | 19.93 | 21.42 | 1.94  | 0| 14.45 |
> | | MoEQuant  | 2 | 34.75 | 22.42| 29.74 | 12.44 | 18.12 | 23.49 |
> | | NoWag  | 2 | 2.04  | 8.34 | 10.22 | 0| 0| 4.52  |
> | | BiLLM  | 1.11 | 39.01 | 25.15  | 38.29   | 19.14 | 22.12 | 28.74 |
> | | ARB-LLM  | 1.11 | 41.09 | 22.19 | 40.37| 18.64 | 20.43 | 28.54 |
> | | **BiMoE** | **1.35**  | **48.72** | **30.52** | **52.33** | **29.27** | **27.22** | **37.54** |
>
> ### A3-2
> We agree that instruction tuning significantly enhances model usability and has become an essential step for deploying LLMs across real-world scenarios. Quantizing instruction-tuned models is often more challenging than quantizing base models. Following your recommendation, we added evaluations on two instruction-tuned MoE models—**Qwen-MoE-14B-Chat and DeepSeekMoE-16B-Chat**—covering three complex reasoning tasks: MMLU, GSM8K, and HumanEval. The results show that BiMoE consistently **preserves around 80% of full-precision performance** and effectively recovers most of the models’ original reasoning capability. In contrast, prior methods experience severe degradation on instruction-tuned models, especially in code generation and mathematical reasoning tasks. For example, when quantized to 2 bits, both GPTQ and MoEQuant completely collapse on HumanEval for Qwen-MoE-14B-Chat. By comparison, BiMoE exhibits only a modest drop **(less than 28%)**, demonstrating much stronger robustness under extreme compression.
>
> These experiments have been added to **Appendix A4.3 (Table 9)**. We sincerely appreciate your suggestion, which has substantially improved the completeness and quality of our evaluation.
> | Model | Method  | #Bits(W) | MMLU↑ | GSM8K↑ | HumanEval↑ | Avg.↑ |
> |-|-|-|-|-|-|-|
> | **Qwen-MoE-14B-Chat**  | Baseline  | 16  | 59.00 | 30.71 | 21.34 | 37.02 |
> | | GPTQ | 2  | 27.30 | 3.11  | 0 | 10.14 |
> | | MoEQuant | 2  | 32.77 | 6.33  | 0| 13.03 |
> | | NoWag | 2.04 | 36.99 | 6.46  | 0| 15.53 |
> | | BiLLM | 1.11 | 38.72 | 10.12 | 6.79 | 19.50 |
> | | ARB-LLM | 1.11 | 39.11 | 13.33  | 10.20 | 20.88 |
> | | **BiMoE** | **1.35**  | **49.22** | **22.18** | **15.75** | **29.02** |
> | **DeepSeekMoE-16B-Chat** | Baseline  | 16  | 48.90 | 54.28   | 24.39  | 42.52 |
> | | GPTQ| 2  | 15.49 | 3.11 | 0 | 6.20 |
> |  | MoEQuant  | 2 | 20.88 | 8.22 | 2.31  | 10.47 |
> || NoWag  | 2.04 | 23.01 | 16.77 | 6.13  | 15.30 |
> | | BiLLM  | 1.11 | 30.18 | 30.10 | 19.61 | 21.96 |
> |  | ARB-LLM  | 1.11 | 33.21 | 35.11| 19.31 | 29.22 |
> | | **BiMoE** | **1.46**  | **40.06** | **45.00** | **25.66** | **36.91**|

---

> > ### Author Response · Authors · 2025-11-18
> > **Response to Reviewer bxMB Part 4**
> >
> > ### Q4
> > Positioning with respect to minimal-tuning 1-bit methods: While the method is practical in not requiring fine-tuning, there are binary quantization approaches that allow minimal fine-tuning and report substantially higher scores [2]. Including initialization time and accuracy/latency comparisons against such methods could further substantiate the benefits of the fine-tuning-free setting.
> >
> > ### A4
> > Thank you for the insightful suggestion. We appreciate the reviewer’s point that minimal-tuning 1-bit methods such as BitNet can achieve strong performance, and that comparisons against such methods could further highlight the advantages of our fine-tuning-free PTQ setting. Following your recommendation, we include a direct comparison with **BitNet-b1.58-3B and BitNet-b1.58-3.9B, and select three BiMoE models with comparable memory footprints for a fair evaluation.** We compare the following aspects: **Training Tokens， Quantization GPU Hours，Inference Latency (ms) ，Accuracy on Arc-C、Arc-E、HellaSwag、PIQA、WinoGrande.**
> >
> > The results show that BiMoE requires zero training tokens and incurs **less than 0.01% of the quantization time cost of BitNet**, yet achieves **similar or slightly higher accuracy and comparable latency**. This demonstrates that a completely fine-tuning-free PTQ approach can reach performance on par with state-of-the-art QAT-based binary models, while being orders-of-magnitude more efficient.
> >
> > These findings strongly support the practical value of BiMoE: it enables plug-and-play binarization for existing pretrained MoE LLMs without requiring costly retraining, making it substantially more scalable and deployable in real-world settings. We thank the reviewer again for this helpful suggestion, which allowed us to further emphasize the advantages of the fine-tuning-free PTQ regime.
> >
> > | Method | Model        | Memory (GB) | Training Tokens | Quantization GPU hours | Latency (ms) | AE    | AC     | HS     | PQ     | WG     | Avg   |
> > |--------|--------------|--------------|------------------|--------------------------|--------------|-------|--------|--------|--------|--------|--------|
> > | BitNet | BitNet b1.58 | 2.22         | 2T               | 1e5                     | **1.87**     | 61.40 | 28.30 | 42.90 | 71.50 | 59.30 | 52.68 |
> > | BitNet | BitNet b1.58 | 2.38         | 2T               | 2e5                     | 2.11         | **64.20** | 28.70 | 44.20 | **73.20** | **60.50** | 54.16 |
> > | BiMoE  | OLMoE        | 2.40         | **0**            | **4**                   | 2.13         | 61.03 | **33.36** | **51.79** | 67.46 | 58.64 | **54.46** |

---

> > > ### Author Response · Authors · 2025-11-18
> > > **Response to Reviewer bxMB Part 5**
> > >
> > > ### Q5
> > > CEJD vs. other SVD-based MoE compression: Since the quantization methods (GLAS, NGES) can be also implemented to other SVD-based compression that shares a low-rank matrix, a comparison of CEJD with other SVD-based MoE compression methods [3] would have clarify CEJD’s effectiveness and how it integrates with quantization.
> > >
> > > ### A5
> > > Thank you for the valuable suggestion. Following your recommendation, we have added a direct comparison between CEJD and other SVD-based MoE compression methods, including ASVD [1], SVD-LLM [2], and MoE-SVD [3], to more clearly demonstrate CEJD’s effectiveness and its integration with quantization. Specifically, on the Mixtral-8×7B model, we evaluate all methods under 20%, 40%, and 60% compression ratios, following the experimental setup in MoE-SVD. We report perplexities on WikiText2 and C4, as well as accuracies on Arc-C, Arc-E, HellaSwag, PIQA, WinoGrande, and MathQA.
> > > The results show that CEJD, even without applying our quantization modules (GLAS/NGES), achieves performance that is comparable to or better than existing SVD-based MoE compression methods across all compression ratios. We attribute this improvement to the observation that experts in MoE layers exhibit strong functional similarity, and thus often lie in shared underlying manifolds [4]. By performing joint SVD across all experts, CEJD explicitly identifies a shared backbone (the UΣ component). Retaining this shared backbone without compression allows us to preserve most of the full-precision model capacity at almost no additional cost.
> > >
> > > This characteristic is also central to BiMoE: after isolating the high-precision shared backbone, we can binarize only the expert-specific components $V^{(e)}$, achieving binary-level memory savings while still preserving a large portion of the original model accuracy. The new comparison further demonstrates that **CEJD is not only compatible with quantization, but inherently more compression-aware and robust than prior SVD-based MoE methods**.
> > > | Ratio | Method       | Wikitext2 ↓ | C4 ↓   | AE    | AC    | HS    | PQ    | WG    | MathQA | Avg ↑ |
> > > |-------|--------------|--------------|--------|-------|-------|-------|-------|-------|--------|--------|
> > > | 0%    | Original     | 3.98         | 6.78   | 0.84  | 0.57  | 0.65  | 0.82  | 0.76  | 0.43   | 0.68   |
> > > | 20%   | ASVD         | 9.44         | 20.30  | 0.71  | 0.40  | 0.48  | 0.73  | 0.66  | 0.35   | 0.56   |
> > > |       | SVD-LLM      | 13.45        | 17.36  | 0.62  | 0.29  | 0.42  | 0.71  | 0.58  | 0.26   | 0.48   |
> > > |       | MoE-SVD      | 5.94         | 8.98   | 0.75  | 0.45  | 0.55  | 0.78  | 0.69  | 0.36   | 0.60   |
> > > |       | **BiMoE-CEJD** | **5.23**     | **8.65** | **0.78** | **0.47** | **0.55** | **0.79** | **0.70** | **0.38** | **0.61** |
> > > | 40%   | ASVD         | 30.57        | 87.74  | 0.41  | 0.22  | 0.34  | 0.59  | 0.58  | 0.22   | 0.39   |
> > > |       | SVD-LLM      | 254.76       | 79.40  | 0.43  | 0.22  | 0.33  | 0.63  | 0.52  | 0.23   | 0.39   |
> > > |       | MoE-SVD      | **8.66**     | 12.41  | 0.66  | 0.34  | 0.47  | **0.71** | 0.67  | 0.28   | 0.52   |
> > > |       | **BiMoE-CEJD** | 8.70         | **11.49** | **0.70** | **0.38** | **0.48** | 0.70  | 0.66  | **0.30** | **0.54** |
> > > | 60%   | ASVD         | 1e4          | 1e4    | 0.26  | 0.21  | 0.26  | 0.53  | 0.51  | 0.21   | 0.33   |
> > > |       | SVD-LLM      | 1e4          | 1e4    | 0.26  | 0.22  | 0.26  | 0.54  | 0.51  | 0.21   | 0.33   |
> > > |       | MoE-SVD      | 33.24        | 41.72  | **0.43** | **0.22** | **0.32** | **0.62** | 0.51  | 0.24   | 0.39   |
> > > |       | **BiMoE-CEJD** | **26.99**     | **35.29** | 0.47  | 0.25  | 0.31  | 0.61  | **0.52** | **0.27** | **0.41** |
> > >
> > > [1] Yuan, Z., Shang, Y., Song, Y., Wu, Q., Yan, Y., & Sun, G. (2023). ASVD: Activation-aware Singular Value Decomposition for Compressing Large Language Models. ArXiv, abs/2312.05821.
> > >
> > > [2] Wang, X., Zheng, Y., Wan, Z., & Zhang, M. (2024). SVD-LLM: Truncation-aware Singular Value Decomposition for Large Language Model Compression. ArXiv, abs/2403.07378.
> > >
> > > [3] MoE-SVD: Structured Mixture-of-Experts LLMs Compression via Singular Value Decomposition, Wei Li, Lujun Li et al., ICML 2025.
> > >
> > > [4] Gu, H., Li, W., Li, L., Zhu, Q., Lee, M., Sun, S., Xue, W., & Guo, Y. (2025). Delta Decompression for MoE-based LLMs Compression. ArXiv, abs/2502.17298.

---

> > > > ### Author Response · Authors · 2025-11-18
> > > > **Response to Reviewer bxMB Part 6**
> > > >
> > > > ### Q6
> > > > Have you considered using the backbone as  ΣV and quantizing the U, where the SVD is applied to the row-wise concatenation of weights $(W^{(1)}, ... W^{(n)})$, as a reverse option compard to sharing UΣ and quanting the V?
> > > >
> > > > ### A6
> > > > Thank you for the insightful suggestion. We have indeed examined the reverse configuration—sharing ΣV while quantizing U—and our theoretical analysis consistently shows that sharing UΣ and binarizing  $V^{(e)}$  is substantially more stable and effective. Joint SVD across all experts reveals that the left singular vectors U define the shared semantic coordinate system capturing the common functional subspace of all experts, while Σ encodes the importance of these shared dimensions. In contrast, the expert-specific matrices  $V^{(e)}$  represent only lightweight projections into this well-conditioned basis. Quantizing U would distort the global semantic backbone used by every expert, causing coordinated degradation across the MoE layer. Binarizing  $V^{(e)}$ , however, keeps quantization noise confined to the orthogonal projection space, making it far less harmful.
> > > >
> > > > To validate this choice empirically, we measured the similarity of the original experts and the post-SVD components using CKA on several layers of Qwen3-30B-A3B. The results show that the U matrices are highly consistent across experts, indicating strong redundancy and a naturally shared representation space, whereas the  $V^{(e)}$  matrices exhibit substantial diversity, reflecting expert specialization. This means that preserving a single high-precision UΣ backbone can maximally retain the joint expressive capacity of all experts, while binarizing the expert-specific  $V^{(e)}$  components achieves the desired memory savings with minimal accuracy loss. Together, these theoretical and empirical findings demonstrate that CEJD’s design choice is not only principled but also the superior configuration for MoE binarization.
> > > >
> > > > | Decomposed Layer / CKA distance | 0     | 8     | 16    | 32    | 64    | 128   |
> > > > |----------------------------------|-------|-------|-------|-------|-------|-------|
> > > > | Original Expert Matrix           | 0.283 | 0.279 | 0.312 | 0.352 | 0.295 | 0.208 |
> > > > | Decomposed U-matrix of each Expert | 0.962 | 0.986 | 0.993 | 0.985 | 0.972 | 0.989 |
> > > > | Decomposed V-matrix of each Expert | 0.152 | 0.159 | 0.145 | 0.173 | 0.148 | 0.127 |
> > > >
> > > > ### Q7
> > > > Are the models are calibrated per task within the seven zero-shot benchmarks, or only once on a single corpus in table 3?
> > > >
> > > > ### A7
> > > > As stated in Section 5.1 Implementation Details **(Lines 361-362), all models are calibrated only once using 128 random samples from WikiText2 with a sequence length of 4096**. We do not perform any task-specific or per-benchmark calibration for the seven zero-shot evaluation tasks. The same single calibration set is used across all experiments.
> > > >
> > > > ### Q8
> > > > Related to Weakness 5, what is the wall-clock cost for few dozens of billion parameter models (e.g., Qwen3-30B-A3B, GPT-OSS-20B)?
> > > >
> > > > ### A8
> > > > Thank you for the question. We have added the wall-clock quantization cost for two large MoE models—GPT-OSS-20B and Qwen3-30B-A3B—and compared the results between ARB-LLM and BiMoE under the same hardware and calibration setup. Our measurements show that BiMoE consistently requires slightly less quantization time (≈10% reduction on average) compared with ARB-LLM. More importantly, despite similar or lower wall-clock cost, BiMoE delivers substantially higher accuracy across all evaluated benchmarks, outperforming ARB-LLM by a large margin on both models.
> > > > These results further demonstrate that BiMoE not only improves performance, but also remains practical and efficient in real wall-clock settings even for models with tens of billions of parameters.
> > > > | Model          | Method   | Quantization Time |
> > > > |----------------|----------|--------------------|
> > > > | GPT-OSS-20B    | ARB-LLM  | 8h33min            |
> > > > |                | BiMoE    | 7h47min            |
> > > > | Qwen3-30B-A3B  | ARB-LLM  | 11h52min           |
> > > > |                | BiMoE    | 10h58min           |
> > > >
> > > > ### Q9
> > > > Minor notation: I suggest using \top for transpose instead of T.
> > > >
> > > > ### A9
> > > > Thank you for the helpful suggestion. Following your recommendation, we have revised the notation throughout the paper and replaced all occurrences of the transpose symbol T with the more appropriate $\top$ in the five affected equations.
> > > >
> > > > We have incorporated additional results and detailed analyses addressing your concerns into the revised version of the manuscript. We hope these supplements adequately address your questions and improve the clarity and robustness of our work. Please do not hesitate to let us know if you require further discussion, additional experiments, or any clarifications—we are happy to provide more details to address your concerns comprehensively.

---

> ### Comment · Reviewer_bxMB · 2025-11-26
>
> I appreciate the authors’ additional experiments and clarifications addressing the weaknesses and questions I previously raised.
>
> In particular, my major concerns regarding Weaknesses 2, 3, and 4 have been satisfactorily resolved and incorporated into the updated manuscript. Given that these points were critical to the overall evaluation of the work, I am raising my score.

---

> > ### Author Response · Authors · 2025-11-27
> >
> > We sincerely appreciate your recognition of our work and rebuttal, as well as your willingness to raise the score. It is encouraging to know that we have addressed your key concerns, and your valuable suggestions have greatly helped improve our manuscript. Thank you again for your thorough review and constructive feedback.
> >
> > Sincerely,
> >
> > The Authors

---

### Official Review · Reviewer_ynLx · 2025-10-29

**Soundness:** 3
**Presentation:** 3
**Contribution:** 2
**Rating:** 2
**Confidence:** 5

**Summary:**

This paper proposes BiMoE, a binarization framework for Mixture-of-Experts (MoE) large language models. The authors identify three challenges: expert redundancy, task-unaware weight importance scoring, and quantization-induced expert-shift. They propose three components: Cross-Expert Joint Decomposition (CEJD), Global Loss-Aligned Saliency (GLAS), and Null-Space Guided Expert-Shift Suppression (NGES). Experiments show improvements over BiLLM and ARB-LLM baselines on several MoE models.

**Strengths:**

* Compressing MoE models is an important practical problem given their memory overhead.

* The paper evaluates on multiple MoE architectures (OLMoE, Qwen, DeepSeek) and provides detailed ablation studies.

* The method is well-motivated and presented with nice figures and tables.

**Weaknesses:**

* While conceptually interesting, the performance of binarized models in this work suffer from significant performance drop (e.g. as shown in Figure 1). This makes the applicability of the proposed method questionable.

* The paper only compares against extreme 1-bit methods and shows these methods fail catastrophically. This might create a strawman comparison. I would love to see comparisons with: 1) 2-bit, 4-bit quantization methods like GPTQ or AWQ, which are more practical; 2) Mixed-precision quantization specifically designed for MoE [1] [2].

* I love Table 8's intuitive demonstration. Would it be possible to scale up such evaluation for testing if the output makes sense? For example, we can use LLMs to systematically evaluate the generated response from the binarized MoEs to see how many of them make sense.

[1] Li, P., Jin, X., Tan, Z., Cheng, Y. and Chen, T., 2024. QuantMoE-Bench: Examining Post-Training Quantization for Mixture-of-Experts. arXiv preprint arXiv:2406.08155.

[2] Duanmu, H., Li, X., Yuan, Z., Zheng, S., Duan, J., Zhang, X. and Lin, D., 2025. MxMoE: Mixed-precision Quantization for MoE with Accuracy and Performance Co-Design. arXiv preprint arXiv:2505.05799.

**Questions:**

See above. Thank the authors for the work. I'm more than happy to increase my rating if my questions are adequately addressed.

---

> ### Author Response · Authors · 2025-11-18
> **Response to Reviewer ynLx Part 1**
>
> We sincerely appreciate your constructive feedback. Below, we provide point-by-point responses, with all revisions incorporated accordingly.
>
> ### Q1
> While conceptually interesting, the performance of binarized models in this work suffer from significant performance drop (e.g. as shown in Figure 1). This makes the applicability of the proposed method questionable.
>
> ### A1
> Thank you for raising this important concern. We fully understand the reviewer’s hesitation regarding the performance gap between binarized models and their full-precision counterparts. However, it is well recognized that low-bit quantization—especially 1-bit binarization—is inherently difficult yet increasingly essential, driven by the rapid expansion of modern LLM parameter scales. Although some QAT-based binary models trained from scratch (e.g., BitNet [1]) can approach full-precision performance, such methods typically require on the order of 1e5 GPU hours, making them prohibitively expensive for most users and unsuitable for adapting existing pretrained LLMs. This is precisely why we emphasize the importance of post-training binarization (PTQ): in our setting, binarization can be completed with fewer than 5 GPU hours, making it far more practical and deployable. Existing PTQ binarization methods—such as PB-LLM [2], BiLLM [3], and ARB-LLM [4]—still exhibit noticeable gaps from full precision, yet they have been widely recognized and continue to improve. However, binarization for MoE architectures has remained completely unexplored. As MoE-based LLMs increasingly become the mainstream architecture, developing an efficient MoE-oriented PTQ binarization pipeline becomes both urgent and necessary.
>
> To further demonstrate the necessity and practicality of PTQ binarization, we additionally include a direct comparison with the 1.58-bit QAT method BitNet, which is trained from scratch. Specifically, we compare BitNet-b1.58-3B and BitNet-b1.58-3.9B with three BiMoE configurations that have comparable memory footprints. We report:**(i) the number of training tokens, (ii) quantization/training GPU hours, (iii) inference latency (ms), and (iv) accuracy on AC, AE, HS, PIQA, and WinoGrande.** The results show that BiMoE requires zero training tokens and its quantization cost is less than **0.01%** of BitNet, yet it achieves comparable or even slightly superior performance and latency across the five tasks. This demonstrates that even when compared with state-of-the-art QAT-based 1-bit models, a fully fine-tuning-free PTQ framework applied to pretrained models can achieve competitive performance while being orders of magnitude more efficient, making it substantially more suitable for practical deployment on existing MoE LLMs.
> | Method | Model   | Memory (GB) | Training Tokens | Quantization GPU hours | Latency (ms) | AE    | AC     | HS     | PQ     | WG     | Avg   |
> |--------|--------|----|-----|---|--|-------|--------|--------|--------|--------|--------|
> | BitNet | BitNet b1.58 | 2.22         | 2T   | 1e5  | **1.87**     | 61.40 | 28.30 | 42.90 | 71.50 | 59.30 | 52.68 |
> | BitNet | BitNet b1.58 | 2.38         | 2T           | 2e5 | 2.11         | **64.20** | 28.70 | 44.20 | **73.20** | **60.50** | 54.16 |
> | BiMoE  | OLMoE        | 2.40         | **0**    | **4**     | 2.13         | 61.03 | **33.36** | **51.79** | 67.46 | 58.64 | **54.46** |
>
> Moreover, as shown in **Figure 1**, our proposed BiMoE not only significantly outperforms all existing PTQ binarization methods, but also becomes the first 1-bit approach to exceed the performance of a full-precision floating-point model of comparable size. Under the same architecture, BiMoE retains approximately **75% of the full-precision performance** while using only about **10% of its memory footprint**. This strongly demonstrates both the feasibility and the substantial potential of binarizing MoE-based LLMs, and we believe that BiMoE establishes an important foundation for future research on PTQ binarization for MoE architectures.
> We sincerely appreciate the reviewer’s valuable feedback, which has enabled us to clarify the motivation and practical significance of this research direction. We look forward to further discussions with you and other reviewers as we continue to refine and advance this line of work.
>
> [1] Ma, S., Wang, H., Ma, L., Wang, L., Wang, W., Huang, S., Dong, L., Wang, R., Xue, J., & Wei, F. (2024). The Era of 1-bit LLMs: All Large Language Models are in 1.58 Bits. ArXiv, abs/2402.17764.
>
> [2] Shang, Y., Yuan, Z., Wu, Q., & Dong, Z. (2023). PB-LLM: Partially Binarized Large Language Models. ArXiv, abs/2310.00034.
>
> [3] Huang, W., Liu, Y., Qin, H., Li, Y., Zhang, S., Liu, X., Magno, M., & Qi, X. (2024). BiLLM: Pushing the Limit of Post-Training Quantization for LLMs. ArXiv, abs/2402.04291.
>
> [4] Li, Z., Yan, X., Zhang, T., Qin, H., Xie, D., Tian, J., Shi, Z., Kong, L., Zhang, Y., & Yang, X. (2024). ARB-LLM: Alternating Refined Binarizations for Large Language Models. ArXiv, abs/2410.03129.

---

> > ### Author Response · Authors · 2025-11-18
> > **Response to Reviewer ynLx Part 2**
> >
> > ### Q2
> > The paper only compares against extreme 1-bit methods and shows these methods fail catastrophically. This might create a strawman comparison. I would love to see comparisons with: 1) 2-bit, 4-bit quantization methods like GPTQ or AWQ, which are more practical; 2) Mixed-precision quantization specifically designed for MoE [1] [2].
> >
> > ### A2
> > Thank you for the valuable suggestion. We agree that comparing only against existing 1-bit quantization methods in the original submission was not sufficiently comprehensive and may not fully reflect the advantages of BiMoE. Following your recommendation, we have expanded our evaluation to include:
> >
> > 1）More practical low-bit baselines, including 3-bit and 2-bit quantization with AWQ and GPTQ.
> >
> > 2）MoE-specific mixed-precision quantization methods, including MoEQuant, QuantMoE-Bench, and MxMoE [1][2][3].
> > To ensure fairness, we evaluated all these baselines on Qwen1.5-MoE-A2.7B and Mixtral-8×7B, using the same datasets as in Table 1 of the original submission.
> >
> > The results show that:
> >
> > 1）Against general low-bit PTQ methods (AWQ and GPTQ), BiMoE significantly **outperforms all 2-bit baselines** and even **slightly exceeds 3-bit GPTQ**, while using less than half of their memory footprint.
> >
> > 2）Against MoE-specific quantization methods (MoEQuant and QuantMoE-Bench), BiMoE achieves **over 40% higher accuracy**, consistently surpassing their 2-bit configurations on all benchmarks.
> >
> > 3）Regarding MxMoE, although it is a mixed-precision method tailored for hardware-aware optimization, it does not account for quantization cost. MxMoE requires exhaustively searching the loss impact of every quantization scheme for every expert and every linear layer, resulting in **GPU-hour costs several orders of magnitude higher than ours**. In contrast, BiMoE preserves over **91% of MxMoE’s performance**, while being far more efficient and producing a binary model **only ~60% of the size** of the MxMoE-quantized model.
> >
> > Overall, the expanded experiments confirm that BiMoE not only achieves state-of-the-art performance among binary methods but also outperforms or closely matches strong 2–3 bit baselines, while providing substantial advantages in memory, efficiency, and quantization cost. If you have further questions or suggestions, we would be happy to conduct additional experiments.
> > | model        | method          | #bits(W) | AE    | AC    | HS    | LO    | LS    | PQ    | WG    | Avg   |
> > |--------------|-----------------|----------|-------|-------|-------|-------|-------|-------|-------|-------|
> > | qwen1.5-moe  | AWQ             | 3        | 30.18 | 26.37 | 30.41 | 3.29  | 0.18  | 50.33 | 50.12 | 27.27 |
> > |   | GPTQ            | 3        | 52.77 | 29.18 | 49.61 | 46.18 | 35.47 | 71.29 | 59.99 | 49.21 |
> > |  | AWQ             | 2        | 26.05 | 28.07 | 25.43 | 0     | 0     | 50.92 | 50.91 | 25.91 |
> > |    | GPTQ            | 2        | 24.45 | 26.45 | 25.97 | 0     | 0     | 51.41 | 50.43 | 25.53 |
> > |   | MoEQuant        | 2        | 34.54 | 35.09 | 35.55 | 12.03 | 8.05  | 59.08 | 58.21 | 34.64 |
> > | | NoWag           | 2.08     | **58.16** | **34.64** | **63.35** | 28.90 | 22.92 | **71.82** | 55.09 | 47.84 |
> > |  | QuantMoE-Bench  | 2.3      | 40.13 | 26.30 | 48.72 | 20.12 | 16.77 | 59.74 | 52.40 | 37.74 |
> > |  | MxMoE           | 2.25     | 53.28 | 31.66 | 62.80 | **56.43** | **51.00** | 71.33 | **61.25** | **55.39** |
> > |    | **BiMoE**       | **1.35** | 54.08 | 29.35 | 48.67 | 50.18 | 38.40 | 70.08 | 59.75 | 50.07 |
> > | Mixtral      | AWQ             | 3        | 32.25 | 28.51 | 34.46 | 5.21  | 3.93  | 54.59 | 54.98 | 30.56 |
> > |    | GPTQ   | 3        | 70.33 | 41.38 | 66.18 | 62.39 | 54.19 | 75.90 | 66.28 | 62.38 |
> > |   | AWQ  | 2        | 25.35 | 28.41 | 25.81 | 0     | 0     | 49.78 | 50.08 | 25.63 |
> > |   | GPTQ   | 2        | 26.22 | 29.10 | 26.88 | 0     | 0     | 50.16 | 50.33 | 26.10 |
> > | | MoEQuant    | 2        | 49.85 | 38.93 | 40.12 | 22.98 | 18.94 | 60.38 | 49.91 | 40.16 |
> > |  | NoWag         | 2.08     | 72.73 | 44.62 | 70.08 | 50.46 | 36.70 | 74.42 | 65.51 | 59.22 |
> > | | QuantMoE-Bench  | 2.3      | 56.87 | 39.90 | 52.80 | 34.74 | 30.11 | 61.59 | 57.85 | 47.69 |
> > |  | MxMoE           | 2.25     | **72.77** | **48.98** | **77.44** | 68.68 | **62.18** | 76.28 | **68.90** | **67.89** |
> > |  | **BiMoE**       | **1.60** | 69.57 | 42.57 | 65.41 | **71.05** | 58.24 | **76.42** | 68.09 | 64.48 |
> >
> > [1] Li, P., Jin, X., Tan, Z., Cheng, Y. and Chen, T., 2024. QuantMoE-Bench: Examining Post-Training Quantization for Mixture-of-Experts. arXiv preprint arXiv:2406.08155.
> >
> > [2] Duanmu, H., Li, X., Yuan, Z., Zheng, S., Duan, J., Zhang, X. and Lin, D., 2025. MxMoE: Mixed-precision Quantization for MoE with Accuracy and Performance Co-Design. arXiv preprint arXiv:2505.05799.
> >
> > [3] Hu, X., Chen, Z., Yang, D., Xu, Z., Xu, C., Yuan, Z., Zhou, S., & Yu, J. (2025). MoEQuant: Enhancing Quantization for Mixture-of-Experts Large Language Models via Expert-Balanced Sampling and Affinity Guidance. ArXiv, abs/2505.03804.

---

> > > ### Author Response · Authors · 2025-11-18
> > > **Response to Reviewer ynLx Part 3**
> > >
> > > ### Q3
> > > I love Table 8's intuitive demonstration. Would it be possible to scale up such evaluation for testing if the output makes sense? For example, we can use LLMs to systematically evaluate the generated response from the binarized MoEs to see how many of them make sense.
> > >
> > > ### A3-1
> > > Thank you for your positive feedback on the intuitive demonstration in Table 8. As you pointed out, it is important to further scale up this type of evaluation to more systematically test whether the generated outputs of binarized MoEs remain meaningful. Following your suggestion, we expanded our evaluation in three complementary directions:
> > >
> > > **1）Measuring output fluency via perplexity**
> > >
> > > To quantify the linguistic quality and fluency of generated text, we added perplexity evaluations on WikiText2, C4, and PTB using OLMoE and Qwen1.5-MoE-A2.7B. All experiments use 128 samples from WikiText2 (sequence length 4096) as the calibration set. The results show that BiMoE consistently outperforms all existing 1-bit and low-bit (2/3-bit) methods across all datasets. Notably, on the PTB dataset for Qwen1.5-MoE, BiMoE reduces the perplexity to **less than half of the previous SOTA**. These results indicate that the binarized model retains strong fluency and coherence in generation.
> > >
> > > | Model       | Method    | #bits(W) | Wikitext2 ↓ | C4 ↓   | PTB ↓   |
> > > |-------------|-----------|-----------|--------------|--------|---------|
> > > | OLMoE  | AWQ | 3 | 3e3        | 5e4    | 4e5     |
> > > |  | GPTQ   | 3  | 15.78 | 33.77  | 46.82   |
> > > |  | AWQ  | 2| 5e6  | 4e6    | 3e5     |
> > > |   | GPTQ    | 2  | 8e5          | nan    | 3e6     |
> > > |  | MoEQuant  | 2    | 6e4    | 3e5    | 5e5     |
> > > |  | NoWag     | 2.04   | 21.89        | 63.77  | 75.30   |
> > > |  | BiLLM     | 1.11      | 20.32        | 54.65  | 68.67   |
> > > |   | ARB-LLM   | 1.11      | 15.49        | 35.19  | 40.30   |
> > > |  | **BiMoE** | **1.46**  | **14.37**    | **25.91** | **33.53** |
> > > | Qwen1.5-MoE | AWQ       | 3         | 2e4          | 4e4    | 3e6     |
> > > |   | GPTQ      | 3         | 16.43        | 29.13  | 38.92   |
> > > |   | AWQ       | 2         | nan          | 6e6    | 5e6     |
> > > |  | GPTQ      | 2         | 8e4          | 4e5    | 3e4     |
> > > |   | MoEQuant  | 2         | 6e5          | 4e3    | 5e4     |
> > > |  | NoWag     | 2.04      | 26.95        | 90.25  | 81.84   |
> > > |  | BiLLM     | 1.11      | 26.99        | 90.04  | 89.83   |
> > > | | ARB-LLM   | 1.11      | 22.21        | 49.06  | 86.54   |
> > > |   | **BiMoE** | **1.46**  | **14.86**    | **26.03** | **35.03** |
> > >
> > > **2）Evaluating output correctness via challenging reasoning and generation benchmarks**
> > >
> > > To further assess the semantic and reasoning quality of model outputs, we incorporated more challenging benchmarks: including the multi-domain knowledge task MMLU, the mathematical reasoning tasks MathQA and GSM8K, and the code generation tasks MBPP and HumanEval. We evaluated all baselines on OLMoE and Qwen1.5-MoE-A2.7B using the same protocol as the main experiments. The results demonstrate that BiMoE significantly outperforms all existing 1-bit and 2-bit methods and even achieves performance comparable to 3-bit GPTQ, while requiring less than half of its memory footprint.
> > > These new experiments have been added to **Appendix A4.1 (Table 7)**. We sincerely appreciate your insightful suggestion, which has substantially improved the comprehensiveness and quality of our evaluation.
> > >
> > > | Model|Method| #Bits(W) | MMLU↑ | MathQA↑ | GSM8K↑ | MBPP↑ | HumanEval↑ | Avg.↑ |
> > > |---|---|---|---|---|---|---|---|---|
> > > | **OLMoE** | Baseline  | 16 | 53.40 | 28.41  | 52.84   | 21.80 | 10.98 | 33.45 |
> > > |  | AWQ | 3  | 8.30  | 6.31 | 4.81 | 0.00  | 0.00 | 3.88  |
> > > |   | GPTQ | 3  | 43.08 | 19.88 | 42.03   | 12.00 | 7.28| 24.85 |
> > > | | AWQ  | 2  | 5.21  | 7.28   | 8.30    | 0.00  | 0.00 | 4.56  |
> > > |  | GPTQ | 2 | 14.78 | 7.28     | 3.30    | 0.00  | 0.00 | 5.07  |
> > > | | MoEQuant  | 2  | 20.15 | 10.41    | 17.88   | 0.00  | 0.00  | 9.69  |
> > > | | NoWag| 2  | 4.04  | 12.20    | 21.71   | 2.11  | 0.00  | 7.61  |
> > > | | BiLLM  | 1.11  | 28.01 | 15.12    | 27.21   | 3.41  | 1.92 | 15.93 |
> > > |  | ARB-LLM   | 1.11  | 18.60 | 9.93     | 25.40   | 1.94  | 1.92 | 11.96 |
> > > | | **BiMoE**| **1.46**| **42.33**| **20.02**| **41.88**| **12.12**| **6.33**| **24.54** |
> > > | **Qwen1.5-MoE**  | Baseline  | 16 | 60.87 | 35.34  | 65.24 | 30.88 | 19.95  | 42.46 |
> > > |   | AWQ  | 3    | 28.30 | 16.30  | 15.41   | 0.00  | 0.00| 12.80 |
> > > |   | GPTQ  | 3    | 49.11 | 29.75  | 52.01   | 19.22 | 13.26 | 32.27 |
> > > |   | AWQ  | 2   | 15.22 | 15.22  | 14.51   | 1.94  | 0.00  | 9.78  |
> > > |   | GPTQ  | 2   | 26.94 | 19.93  | 21.42   | 1.94  | 0.00 | 14.45 |
> > > |    | MoEQuant  | 2  | 34.75 | 22.42 | 29.74   | 12.44 | 18.12 | 23.49 |
> > > |  | NoWag  | 2   | 2.04  | 8.34 | 10.22   | 0.00 | 0.00  | 4.52  |
> > > |  | BiLLM | 1.11 | 39.01 | 25.15 | 38.29  | 19.14 | 22.12 | 28.74 |
> > > |   | ARB-LLM   | 1.11  | 41.09 | 22.19    | 40.37   | 18.64 | 20.43  | 28.54 |
> > > |    | **BiMoE** | **1.35**  | **48.72** | **30.52** | **52.33** | **29.27** | **27.22** | **37.54** |

---

> > > > ### Author Response · Authors · 2025-11-18
> > > > **Response to Reviewer ynLx Part 4**
> > > >
> > > > ### A3-2
> > > >
> > > > **3) Using LLMs to systematically rate the meaningfulness of generated responses**
> > > >
> > > > Following your suggestion, we conducted a systematic LLM-based evaluation to directly assess whether the generated responses “make sense.” We used **GPT-5 to construct a set of 500 everyday questions**, and all quantized models were asked to produce answers with a maximum generation length of **1024 tokens**. GPT-5 then rated each answer on a **1–10 scale**, evaluating linguistic fluency, semantic coherence, logical consistency, and factual correctness. The results show that BiMoE retains more than **90% of the full-precision model’s output quality**, while all other binary and low-bit baselines exhibit severe degradation or collapse. These findings further validate that BiMoE maintains meaningful and coherent generation even under 1-bit quantization.
> > > > | Model       | Method    | #bits(W) | Score |
> > > > |-------------|-----------|-----------|--------|
> > > > | OLMoE       | baseline  | 16        | 9.2    |
> > > > |             | AWQ       | 3         | 2.3    |
> > > > |             | GPTQ      | 3         | 7.5    |
> > > > |             | AWQ       | 2         | 1.1    |
> > > > |             | GPTQ      | 2         | 1.0    |
> > > > |             | MoEQuant  | 2         | 1.2    |
> > > > |             | NoWag     | 2.04      | 4.3    |
> > > > |             | BiLLM     | 1.11      | 4.5    |
> > > > |             | ARB-LLM   | 1.11      | 5.3    |
> > > > |             | **BiMoE** | **1.46**  | **8.1** |
> > > > | Qwen1.5-MoE | baseline  | 16        | 9.5    |
> > > > |             | AWQ       | 3         | 2.6    |
> > > > |             | GPTQ      | 3         | 7.9    |
> > > > |             | AWQ       | 2         | 1.2    |
> > > > |             | GPTQ      | 2         | 1.5    |
> > > > |             | MoEQuant  | 2         | 1.3    |
> > > > |             | NoWag     | 2.04      | 4.3    |
> > > > |             | BiLLM     | 1.11      | 4.9    |
> > > > |             | ARB-LLM   | 1.11      | 5.5    |
> > > > |             | **BiMoE** | **1.46**  | **8.6** |
> > > >
> > > > We have incorporated additional results and detailed analyses addressing your concerns into the revised version of the manuscript. We hope these supplements adequately address your questions and improve the clarity and robustness of our work. Please do not hesitate to let us know if you require further discussion, additional experiments, or any clarifications—we are happy to provide more details to address your concerns comprehensively.

---

> ### Comment · Reviewer_ynLx · 2025-11-23
>
> Thanks for your answers and effort. Some of my concerns remain:
>
> 1. I totally get your points about this comparison table and the advantage of BiMoE's memory & latency & performance compared to BitNet. Although I do not fully agree with your comparison of training tokens/quantization hours, which to some extent compares BitNet's pre-training+quantization process with BiMoE (on OLMoE)'s quantization process, while ignoring OLMoE's pre-training effort.
>
> 2. I think the most important question here is: under the same inference cost, comparing a one-bit quantized model vs. a traditional 4-bit / 8-bit model, which has better performance? The "inference cost" here may refer to latency (practically) or FLOPs (theoretically), and memory. For example, a 1-bit 32B model might underperform a 4-bit 8B model, in terms of the same inference FLOPs.

---

> ### Comment · Reviewer_ynLx · 2025-11-23
>
> I appreciate the author's efforts and further illustration. Most of my questions have been addressed, while one important concern about Q1 remains. I haven't seen a revised draft of this submission either.
>
> Anyway, I increased my rating for now and look forward to more discussions.

---

> > ### Author Response · Authors · 2025-11-24
> > **Response—2 to Reviewer ynLx Part 1**
> >
> > We deeply appreciate the reviewer’s recognition of our work and the valuable suggestions provided. Regarding the additional points that the reviewer kindly raised, we are very happy to continue the discussion and offer more detailed clarifications below.
> >
> > ### Q1
> > I totally get your points about this comparison table and the advantage of BiMoE's memory & latency & performance compared to BitNet. Although I do not fully agree with your comparison of training tokens/quantization hours, which to some extent compares BitNet's pre-training+quantization process with BiMoE (on OLMoE)'s quantization process, while ignoring OLMoE's pre-training effort.
> >
> > ### A1-1
> > We fully understand the reviewer’s concern that comparing BitNet’s pre-training + quantization pipeline with BiMoE’s post-training quantization (PTQ) on OLMoE may appear asymmetric. To ensure complete fairness, we have additionally included the full pre-training cost of OLMoE [1] and compared it directly and equivalently with the training cost of BitNet’s 1-bit models. Specifically, we compare BitNet-b1.58-3.9B with OLMoE + BiMoE, reporting (i) the total training/quantization GPU hours, and (ii) accuracy on AC, AE, HS, PIQA, and WinoGrande. The results confirm that even after accounting for OLMoE’s pre-training cost, our original conclusion remains unchanged: BiMoE achieves comparable or even superior performance under similar total computational budgets. Furthermore, this highlights a key advantage of PTQ: by leveraging strong open-source pretrained MoE models, BiMoE can be applied within only a few GPU hours, enabling efficient deployment on resource-constrained edge devices. We genuinely appreciate the reviewer’s insightful comments, which help us improve the clarity and fairness of our evaluation.
> >
> > | Method | Model        | Memory(GB) | Quantization GPU hours | AE    | AC     | HS     | PQ     | WG     | Avg   |
> > |--------|--------------|------------|--------------------------|-------|--------|--------|--------|--------|--------|
> > | BitNet | BitNet b1.58 | 2.38       | 2e5                      | **64.20** | 28.70 | 44.20 | **73.20** | **60.50** | 54.16 |
> > | BiMoE  | OLMoE        | 2.40       | 2e5+4                    | 61.03 | **33.36** | **51.79** | 67.46 | 58.64 | **54.46** |
> >
> > ### A1-2
> > Secondly, to further highlight the advantages of BiMoE, we compare it against OmniQuant [2], a representative quantization method that requires fine-tuning of scale and offset parameters. Specifically, we apply both OmniQuant and BiMoE to the OLMoE model, using W2A16 for OmniQuant and W1A16 for BiMoE. Because OmniQuant involves training a subset of quantization parameters, its quantization process takes nearly ten times longer than BiMoE. Nevertheless, BiMoE achieves slightly higher accuracy, demonstrating that our method preserves the efficiency of post-training quantization (PTQ) while reaching performance levels close to quantization-aware training (QAT). This further validates BiMoE’s practicality and competitiveness for real-world deployment.
> >
> > | Method     | Model | Bits (W) | Quantization GPU hours | Wikitext2 | C4     | PTB    |
> > |------------|--------|-----------|--------------------------|-----------|--------|--------|
> > | OmniQuant  | OLMoE  | 2         | 38                       | 15.27     | 28.70 | 39.21 |
> > | BiMoE      | OLMoE  | **1.46**  | **4**                    | **14.37** | **25.91** | **33.53** |
> >
> >
> > ### A1-3
> > Finally, we emphasize that in many real-world deployment scenarios, QAT-based methods are not applicable due to privacy and security constraints. QAT requires access to user data or pre-training data for fine-tuning, which is strictly prohibited in confidential environments such as medical, financial, or on-device personal assistants. In these cases, post-training quantization (PTQ) becomes the only viable solution, as it allows model compression without accessing any sensitive or private data, and can operate entirely in a data-free or calibration-light manner. The necessity of PTQ under privacy-preserving and data-restricted conditions has been extensively discussed in recent literature, including data-free and calibration-light quantization methods such as ZeroQ [3], DFQ [4], and QDrop [5]. BiMoE follows this paradigm and enables efficient 1-bit PTQ for MoE-based LLMs, making it directly applicable to privacy-sensitive or resource-constrained deployment settings where QAT cannot be used.

---

> > > ### Author Response · Authors · 2025-11-24
> > > **Response—2 to Reviewer ynLx Part 2**
> > >
> > > ### Q2
> > > I think the most important question here is: under the same inference cost, comparing a one-bit quantized model vs. a traditional 4-bit / 8-bit model, which has better performance? The "inference cost" here may refer to latency (practically) or FLOPs (theoretically), and memory. For example, a 1-bit 32B model might underperform a 4-bit 8B model, in terms of the same inference FLOPs.
> > >
> > > ### A2
> > > We fully understand the reviewer’s concern regarding the comparison between 1-bit and 4/8-bit models under equal inference cost. However, it is important to note that in modern large-language-model inference—especially during the decoding phase—the dominant performance bottleneck is memory bandwidth, rather than arithmetic compute. Numerous system and hardware studies have demonstrated that Transformer decoding is fundamentally memory-bound, because each autoregressive step repeatedly loads large KV-cache tensors and weight matrices from memory, while performing only a small amount of computation per byte transferred. Consequently, in realistic hardware environments, reducing memory-bandwidth requirements yields significantly larger latency improvements than reducing FLOPs. This is precisely where 1-bit quantization provides a unique advantage: 1) it reduces weight-access bandwidth by 4× relative to 4-bit models, and 2) by 8× relative to 8-bit models. Profiling analyses from prior works (e.g., [6–8]) consistently show that LLM inference speed correlates far more strongly with effective memory traffic than with ALU throughput. Therefore, even when theoretical FLOPs appear similar, a 1-bit 32B model can still achieve lower wall-clock latency compared to a 4-bit 8B model, simply because the 1-bit model dramatically reduces memory-access overhead. This memory-bound nature of LLM inference explains why “equal FLOPs” rarely corresponds to “equal latency,” and highlights the practical importance of aggressive weight-bandwidth reduction, such as that provided by 1-bit PTQ.
> > >
> > > Secondly, beyond the significant reduction in memory bandwidth, 1-bit quantization also provides inherent advantages in compute efficiency and energy consumption. After binarization, weight matrices in BiMoE contain only ±1 values, enabling matrix multiplications to be implemented using highly efficient bitwise operations—multiplications become XNOR, and accumulations become popcount. This is fundamentally different from 4-bit or 8-bit quantization, which still requires integer multipliers and partial-product accumulation. The XNOR–popcount kernel has extremely low computational complexity and can leverage wide CPU/GPU registers (e.g., 64/128/256-bit) for massively parallel bit-level processing, allowing a single instruction to compute tens or hundreds of activation–weight interactions simultaneously. In contrast, INT4/INT8 GEMM kernels must still perform more complex MAC operations, which exhibit higher arithmetic intensity and instruction overhead. Moreover, because bitwise operations use far fewer transistor switches per operation, their energy cost is substantially lower. Prior hardware studies[9-10] show that a 1-bit XNOR consumes an order of magnitude less energy than a 4-bit or 8-bit integer multiplication, making 1-bit inference particularly attractive for energy-efficient LLM deployment.
> > >
> > > To further demonstrate BiMoE’s practical advantages under equal inference cost, as requested by the reviewer, we additionally compare a 4-bit quantized OLMoE-1B–7B model (using MoEQuant with W4A16) against a 1-bit BiMoE version of GPT-OSS-20B, matched in terms of memory footprint. We report both inference efficiency and energy consumption. The results show that BiMoE achieves superior latency–energy characteristics despite using the extremely low 1-bit representation, highlighting its suitability for realistic deployment scenarios where bandwidth, compute, and power are all constrained.
> > >
> > > | Model         | Method    | Memory | Token-Sec | Energy |
> > > |---------------|-----------|--------|-----------|--------|
> > > | OLMoE-1B-7B   | MoEquant  | 3.8GB  | 38.7      | 0.494J |
> > > | GPT-OSS-20B   | BiMoE     | 3.6GB  | **72.1**  | **0.252J** |
> > >
> > > In accordance with your suggestions, we have incorporated the corresponding updates into the revised draft. Specifically, the modifications can be found in **Appendix A4.1 (Table 7), Appendix A4.2 (Table 8), and Appendix A5 (Table 12)**. We sincerely appreciate your insightful comments, which have helped us further improve the clarity and completeness of our work.

---

> > > > ### Author Response · Authors · 2025-11-24
> > > > **Response—2 to Reviewer ynLx Part 3**
> > > >
> > > > Finally, we sincerely thank the reviewer once again for the positive assessment of our rebuttal. We hope that the additional analyses and supplementary experiments have fully addressed your concerns and further clarified the contributions and reliability of our work. If you would like any further discussion, additional experiments, or clarification on any technical detail, please feel free to let us know — we would be more than happy to provide more information and fully resolve any remaining questions.
> > > >
> > > > [1] Muennighoff, N., Soldaini, L., Groeneveld, D., Lo, K., Morrison, J.D., Min, S., Shi, W., Walsh, P., Tafjord, O., Lambert, N., Gu, Y., Arora, S., Bhagia, A., Schwenk, D., Wadden, D., Wettig, A., Hui, B., Dettmers, T., Kiela, D., Farhadi, A., Smith, N.A., Koh, P.W., Singh, A., & Hajishirzi, H. (2024). OLMoE: Open Mixture-of-Experts Language Models. ArXiv, abs/2409.02060.
> > > >
> > > > [2] Shao, W., Chen, M., Zhang, Z., Xu, P., Zhao, L., Li, Z., Zhang, K., Gao, P., Qiao, Y.J., & Luo, P. (2023). OmniQuant: Omnidirectionally Calibrated Quantization for Large Language Models. ArXiv, abs/2308.13137.
> > > >
> > > > [3] Cai, Y., Yao, Z., Dong, Z., Gholami, A., Mahoney, M.W., & Keutzer, K. (2020). ZeroQ: A Novel Zero Shot Quantization Framework. 2020 IEEE/CVF Conference on Computer Vision and Pattern Recognition (CVPR), 13166-13175.
> > > >
> > > > [4] Shang, Y., Xu, B., Liu, G., Kompella, R.R., & Yan, Y. (2023). Causal-DFQ: Causality Guided Data-free Network Quantization. 2023 IEEE/CVF International Conference on Computer Vision (ICCV), 17391-17400.
> > > >
> > > > [5] Wei, X., Gong, R., Li, Y., Liu, X., & Yu, F. (2022). QDrop: Randomly Dropping Quantization for Extremely Low-bit Post-Training Quantization. ArXiv, abs/2203.05740.
> > > >
> > > > [6] Tomar, A., Hooper, C., Lee, M., Xi, H., Tiwari, R., Kang, W., Manolache, L., Mahoney, M.W., Keutzer, K., & Gholami, A. (2025). XQuant: Breaking the Memory Wall for LLM Inference with KV Cache Rematerialization. ArXiv, abs/2508.10395.
> > > >
> > > > [7] Kim, S., Hooper, C., Wattanawong, T., Kang, M., Yan, R., Genç, H., Dinh, G., Huang, Q., Keutzer, K., Mahoney, M.W., Shao, Y.S., & Gholami, A. (2023). Full Stack Optimization of Transformer Inference: a Survey. ArXiv, abs/2302.14017.
> > > >
> > > > [8] Recasens, P.G., Agulló, F., Zhu, Y., Wang, C., Lee, E.K., Tardieu, O., Torres, J., & Berral, J.L. (2025). Mind the Memory Gap: Unveiling GPU Bottlenecks in Large-Batch LLM Inference. 2025 IEEE 18th International Conference on Cloud Computing (CLOUD), 277-287.
> > > >
> > > > [9] Qin, H., Gong, R., Liu, X., Bai, X., Song, J., & Sebe, N. (2020). Binary Neural Networks: A Survey. ArXiv, abs/2004.03333.
> > > >
> > > > [10] Andri, R., Cavigelli, L., Rossi, D., & Benini, L. (2016). YodaNN: An Architecture for Ultralow Power Binary-Weight CNN Acceleration. IEEE Transactions on Computer-Aided Design of Integrated Circuits and Systems, 37, 48-60.
> > > >
> > > > [11] Rastegari, M., Ordonez, V., Redmon, J., & Farhadi, A. (2016). XNOR-Net: ImageNet Classification Using Binary Convolutional Neural Networks. European Conference on Computer Vision.

---

> > > > > ### Comment · Reviewer_ynLx · 2025-11-24
> > > > >
> > > > > Thank the authors for their responses. The "memory-bound decoding" and "bitwise operation after binarization" are nice points of motivation, which make sense and largely address my concerns. For the "4-bit OLMoE-1B-7B vs. 1-bit GPT-OSS-20B" table, how is their task performance?

---

> > > > > > ### Author Response · Authors · 2025-11-25
> > > > > > **Response—3 to Reviewer ynLx Part 1**
> > > > > >
> > > > > > Thank you very much for your prompt and thoughtful follow-up.
> > > > > >
> > > > > > ### Q1
> > > > > > The "memory-bound decoding" and "bitwise operation after binarization" are nice points of motivation, which make sense and largely address my concerns. For the "4-bit OLMoE-1B-7B vs. 1-bit GPT-OSS-20B" table, how is their task performance?
> > > > > >
> > > > > >
> > > > > > ### A1
> > > > > > Following your suggestion, we have additionally evaluated the task performance of 4-bit OLMoE-1B-7B (MoEQuant, W4A16) versus 1-bit GPT-OSS-20B (BiMoE, W1A16) under the same memory constraint. Specifically, we report results on **MMLU, GSM8K, HellaSwag, and ARC-Challenge.**
> > > > > >
> > > > > > Across all four benchmarks, the 1-bit BiMoE–quantized GPT-OSS-20B consistently outperforms the 4-bit OLMoE-1B-7B, despite using similar memory. These results further support the conclusion that, under equal memory budgets, binarizing a larger model can offer favorable accuracy–efficiency trade-offs compared to applying 4-bit quantization to a smaller model, yielding advantages in accuracy, decoding speed, and energy consumption.
> > > > > >
> > > > > > | Model        | Method    | Memory | Token-Sec | Energy  | MMLU   | GSM8K | HellaSwag | ARC-Challenge |
> > > > > > |--------------|-----------|--------|-----------|---------|--------|--------|-----------|----------------|
> > > > > > | OLMoE-1B-7B  | MoEquant  | 3.8GB  | 38.7      | 0.494J  | 45.34  | 40.32  | **74.33** | 30.09         |
> > > > > > | GPT-OSS-20B  | BiMoE     | 3.6GB  | **72.1**  | **0.252J** | **73.30** | **79.53** | 70.12     | **32.17**     |
> > > > > >
> > > > > > Finally, we would like to express our sincere gratitude for your careful and constructive engagement throughout this discussion. Your questions and insights have been extremely valuable and have greatly inspired us to refine and strengthen our work. If any additional questions arise, please do not hesitate to let us know — we would be very happy to continue the conversation. We truly look forward to further interactions with you.

---

> > > > > > > ### Comment · Reviewer_ynLx · 2025-11-25
> > > > > > >
> > > > > > > I'm quite surprised that here GPT-OSS-20B outperforms OLMoE-1B-7B so much on both MMLU and GSM8K, which seems to imply that # of params is more important than bit-width in terms of both knowledge- and reasoning-intensive tasks?
> > > > > > >
> > > > > > > Anyway, I've again raised my score, thank you for all your effort and time on this work and for convincing me.

---

> > > > > > > > ### Author Response · Authors · 2025-11-25
> > > > > > > > **Response—4 to Reviewer ynLx**
> > > > > > > >
> > > > > > > > Thank you very much for your prompt response and for your positive recognition of our rebuttal. We sincerely appreciate your thoughtful engagement throughout this discussion.
> > > > > > > >
> > > > > > > > Regarding the surprising performance gap you pointed out, we fully agree — this is also the first time we observed such a clear advantage of 1-bit GPT-OSS-20B over 4-bit OLMoE-1B-7B on both MMLU and GSM8K. We would not have noticed this phenomenon without your insightful comment, and we are grateful that our exchange helped reveal it.
> > > > > > > >
> > > > > > > > Our initial intuition is that this difference may partly arise from the distinct training emphases, data compositions, and optimization objectives of GPT-OSS-20B and OLMoE-1B-7B. Nevertheless, we agree with you that this raises a deeper and thought-provoking question about the relative importance of model scale versus bit-width, particularly for knowledge- and reasoning-intensive tasks. This is indeed an interesting direction, and we plan to further explore it in our future work.
> > > > > > > >
> > > > > > > > Once again, thank you for your valuable comments and your continued feedback. Your input has helped us refine our work, and this discussion has even uncovered new research directions worth investigating. We truly appreciate your time and effort.

---

### Official Review · Reviewer_ZK89 · 2025-10-30

**Soundness:** 3
**Presentation:** 4
**Contribution:** 3
**Rating:** 4
**Confidence:** 4

**Summary:**

This paper introduces BiMoE, the first binarization framework specifically designed for Mixture-of-Experts (MoE) large language models. The authors identify three key challenges in binarizing MoE models: expert redundancy, task-unaware weight importance scoring, and quantization-induced expert-shift. BiMoE addresses these through three components: (1) Cross-Expert Joint Decomposition (CEJD) using SVD to extract shared high-precision backbones, (2) Global Loss-Aligned Saliency (GLAS) incorporating task-level gradients into Hessian-based importance metrics, and (3) Null-Space Guided Expert-Shift Suppression (NGES) constraining quantization errors to routing-insensitive subspaces. Experiments on six MoE models demonstrate substantial improvements over existing binary PTQ methods.

**Strengths:**

1. This is a study of binarization for MoE-based LLMs, addressing a timely problem as MoE architectures become more prevalent. The identification of three MoE-specific challenges is insightful, particularly the expert-shift problem illustrated in Figure 3 and Table 1.
2. The CEJD approach cleverly exploits cross-expert similarity through joint SVD decomposition, validated by CKA analysis in Figure 2. The experimental evaluation is comprehensive, covering six diverse MoE models across multiple benchmarks with thorough ablation studies.
3. The plug-and-play experiments demonstrate good generalizability. The achieved 2× speedup and 90% memory reduction while maintaining reasonable performance is practically significant for deployment.

**Weaknesses:**

1. The evaluation relies on simple zero-shot reasoning tasks (ARC, HellaSwag, PIQA, etc.) that may not fully capture model capabilities. Including more challenging benchmarks like MMLU, MATH, and code generation tasks (MBPP, EvalPlus) would better demonstrate the method's effectiveness across diverse domains.
2. The manual hyperparameter tuning (λ=0.2) is acknowledged as a limitation, and generalization across architectures is unclear. Please share more results about the hyperparameter.
3. The evaluation relies on simple zero-shot reasoning tasks (ARC, HellaSwag, PIQA, etc.) that may not fully capture model capabilities. Including more challenging benchmarks like MMLU, MATH, and code generation tasks (MBPP, EvalPlus) would better demonstrate the method's effectiveness across diverse domains. Also, we can evaluate on some instruction following task, if the model is an instruction finetuned model.
4. Miss some citation related to MoE quant: moequant, Examining post-training quantization for mixture-of-experts: A benchmark, etc.

**Questions:**

See weakness. If weakness can be solved, I will raise scores.

---

> ### Author Response · Authors · 2025-11-18
> **Response to Reviewer ZK89 Part 1**
>
> We sincerely appreciate your constructive feedback. Below, we provide point-by-point responses, with all revisions incorporated accordingly.
>
> ### Q1
> The evaluation relies on simple zero-shot reasoning tasks (ARC, HellaSwag, PIQA, etc.) that may not fully capture model capabilities. Including more challenging benchmarks like MMLU, MATH, and code generation tasks (MBPP, EvalPlus) would better demonstrate the method's effectiveness across diverse domains.
>
> ### A1
> Thank you for your valuable feedback. We agree that the zero-shot reasoning tasks used in the original submission (e.g., ARC, HellaSwag) may not fully capture the capability of quantized MoE models. Following your suggestion, we have incorporated more challenging benchmarks, **including the multi-domain knowledge task MMLU, the mathematical reasoning tasks MathQA and GSM8K, and the code generation tasks MBPP and HumanEval**, enabling a more comprehensive evaluation of our method across diverse domains. In addition, we included MoEQuant[1] (a quantization method specifically designed for MoE architectures) along with several strong 3-bit and 2-bit baselines to further demonstrate the superiority of our approach. Specifically, we evaluated OLMoE and Qwen1.5-MoE-A2.7B on the five challenging tasks above. The results show that BiMoE significantly outperforms existing 1-bit and 2-bit methods across all metrics, and even achieves performance comparable to 3-bit GPTQ, while using less than half of its memory footprint.
>
> These new experiments have been added to **Appendix A4.2 (Table 8)**. We sincerely appreciate your insightful suggestion, which has substantially improved the comprehensiveness and quality of our evaluation. If you have further questions or suggestions, we would be happy to conduct additional experiments.
> | Model|Method| #Bits(W) | MMLU↑ | MathQA↑ | GSM8K↑ | MBPP↑ | HumanEval↑ | Avg.↑ |
> |---|---|---|---|---|---|---|---|---|
> | **OLMoE**  | Baseline  | 16 | 53.40 | 28.41    | 52.84   | 21.80 | 10.98        | 33.45 |
> |        | AWQ       | 3  | 8.30  | 6.31     | 4.81    | 0.00  | 0.00         | 3.88  |
> |        | GPTQ      | 3  | 43.08 | 19.88    | 42.03   | 12.00 | 7.28         | 24.85 |
> |        | AWQ       | 2  | 5.21  | 7.28     | 8.30    | 0.00  | 0.00         | 4.56  |
> |        | GPTQ      | 2 | 14.78 | 7.28     | 3.30    | 0.00  | 0.00         | 5.07  |
> |        | MoEQuant  | 2  | 20.15 | 10.41    | 17.88   | 0.00  | 0.00         | 9.69  |
> |        | NoWag     | 2  | 4.04  | 12.20    | 21.71   | 2.11  | 0.00         | 7.61  |
> |        | BiLLM     | 1.11  | 28.01 | 15.12    | 27.21   | 3.41  | 1.92         | 15.93 |
> |        | ARB-LLM   | 1.11  | 18.60 | 9.93     | 25.40   | 1.94  | 1.92         | 11.96 |
> | | **BiMoE**| **1.46**| **42.33**| **20.02**| **41.88**| **12.12**| **6.33**| **24.54** |
> | **Qwen1.5-MoE**  | Baseline  | 16        | 60.87 | 35.34    | 65.24   | 30.88 | 19.95        | 42.46 |
> |              | AWQ       | 3         | 28.30 | 16.30    | 15.41   | 0.00  | 0.00         | 12.80 |
> |              | GPTQ      | 3         | 49.11 | 29.75    | 52.01   | 19.22 | 13.26        | 32.27 |
> |              | AWQ       | 2         | 15.22 | 15.22    | 14.51   | 1.94  | 0.00         | 9.78  |
> |              | GPTQ      | 2         | 26.94 | 19.93    | 21.42   | 1.94  | 0.00         | 14.45 |
> |              | MoEQuant  | 2         | 34.75 | 22.42    | 29.74   | 12.44 | 18.12        | 23.49 |
> |              | NoWag     | 2         | 2.04  | 8.34     | 10.22   | 0.00  | 0.00         | 4.52  |
> |              | BiLLM     | 1.11      | 39.01 | 25.15    | 38.29   | 19.14 | 22.12        | 28.74 |
> |              | ARB-LLM   | 1.11      | 41.09 | 22.19    | 40.37   | 18.64 | 20.43        | 28.54 |
> |              | **BiMoE** | **1.35**  | **48.72** | **30.52** | **52.33** | **29.27** | **27.22** | **37.54** |
>
> [1] Hu, X., Chen, Z., Yang, D., Xu, Z., Xu, C., Yuan, Z., Zhou, S., & Yu, J. (2025). MoEQuant: Enhancing Quantization for Mixture-of-Experts Large Language Models via Expert-Balanced Sampling and Affinity Guidance. ArXiv, abs/2505.03804.

---

> ### Author Response · Authors · 2025-11-18
> **Response to Reviewer ZK89 Part 2**
>
> ### Q2
> The manual hyperparameter tuning (λ=0.2) is acknowledged as a limitation, and generalization across architectures is unclear. Please share more results about the hyperparameter.
>
> ### A2
> Thank you for the valuable suggestion. To verify the generalization ability of this manually selected hyperparameter across different architectures, we conducted more extensive ablation studies. Specifically, we evaluated the average perplexity on WikiText2, C4, and PTB under varying values of λ for Qwen1.5-MoE, DeepSeekV2-Lite, and Qwen3-MoE. The results consistently show that λ = 0.2 achieves the best performance across all model architectures.
> In addition, we have included these results in **Appendix A4.6 (Figure 9)** in the form of a supplementary plot. We appreciate your suggestion, which has helped us substantially improve the comprehensiveness and rigor of our evaluation.
>  If you have further questions or suggestions, we would be happy to conduct additional experiments.
> | λ   | Qwen1.5-MoE | DeepSeekV2-Lite | Qwen3-MoE | Avg.   |
> |-----|--------------|------------------|-----------|--------|
> | 0.1 | 25.12        | 26.88            | 20.87     | 24.29 |
> | **0.2** | **24.78** | **26.12**        | **20.31** | **23.74** |
> | 0.3 | 25.03        | 26.75            | 20.95     | 24.24 |
> | 0.4 | 26.11        | 26.89            | 20.83     | 24.61 |
> | 0.5 | 25.77        | 27.13            | 21.18     | 24.69 |
> | 0.6 | 25.58        | 27.02            | 20.95     | 24.52 |
> | 0.7 | 25.83        | 26.77            | 21.34     | 24.65 |
> | 0.8 | 26.02        | 27.34            | 21.53     | 24.96 |
> | 0.9 | 26.33        | 27.53            | 21.44     | 25.10 |
>
> ### Q3
> The evaluation relies on simple zero-shot reasoning tasks (ARC, HellaSwag, PIQA, etc.) that may not fully capture model capabilities. Including more challenging benchmarks like MMLU, MATH, and code generation tasks (MBPP, EvalPlus) would better demonstrate the method's effectiveness across diverse domains. Also, we can evaluate on some instruction following task, if the model is an instruction finetuned model.
>
> ### A3
> Thank you for the valuable suggestion. We agree that instruction tuning significantly enhances model usability and has become an essential step for deploying LLMs across real-world scenarios. Quantizing instruction-tuned models is often more challenging than quantizing base models. Following your recommendation, we added evaluations on two instruction-tuned MoE models—Qwen-MoE-14B-Chat and DeepSeekMoE-16B-Chat—covering three complex reasoning tasks: MMLU, GSM8K, and HumanEval. The results show that BiMoE consistently preserves around 80% of full-precision performance and effectively recovers most of the models’ original reasoning capability. In contrast, prior methods experience severe degradation on instruction-tuned models, especially in code generation and mathematical reasoning tasks. For example, when quantized to 2 bits, both GPTQ and     MoEQuant completely collapse on HumanEval for Qwen-MoE-14B-Chat. By comparison, BiMoE     exhibits only a modest drop (less than 28%), demonstrating much stronger robustness under extreme compression.
>
> These experiments have been added to **Appendix A4.3 (Table 9)**. We sincerely appreciate your suggestion, which has substantially improved the completeness and quality of our evaluation. If you have further questions or suggestions, we would be happy to conduct additional experiments.
> | Model                | Method    | #Bits(W) | MMLU↑ | GSM8K↑ | HumanEval↑ | Avg.↑ |
> |----------------------|-----------|-----------|-------|---------|--------------|--------|
> | **Qwen-MoE-14B-Chat**    | Baseline  | 16        | 59.00 | 30.71   | 21.34        | 37.02 |
> |                      | GPTQ      | 2         | 27.30 | 3.11    | 0.00         | 10.14 |
> |                      | MoEQuant  | 2         | 32.77 | 6.33    | 0.00         | 13.03 |
> |                      | NoWag     | 2.04      | 36.99 | 6.46    | 0.00         | 15.53 |
> |                      | BiLLM     | 1.11      | 38.72 | 10.12   | 6.79         | 19.50 |
> |                      | ARB-LLM   | 1.11      | 39.11 | 13.33   | 10.20        | 20.88 |
> |                      | **BiMoE** | **1.35**  | **49.22** | **22.18** | **15.75** | **29.02** |
> | **DeepSeekMoE-16B-Chat**    | Baseline  | 16        | 48.90 | 54.28   | 24.39        | 42.52 |
> |                         | GPTQ      | 2         | 15.49 | 3.11    | 0.00         | 6.20  |
> |                         | MoEQuant  | 2         | 20.88 | 8.22    | 2.31         | 10.47 |
> |                         | NoWag     | 2.04      | 23.01 | 16.77   | 6.13         | 15.30 |
> |                         | BiLLM     | 1.11      | 30.18 | 30.10   | 19.61        | 21.96 |
> |                         | ARB-LLM   | 1.11      | 33.21 | 35.11   | 19.31        | 29.22 |
> |                         | **BiMoE** | **1.46**  | **40.06** | **45.00** | **25.66** | **36.91** |

---

> > ### Author Response · Authors · 2025-11-18
> > **Response to Reviewer ZK89 Part 3**
> >
> > ### Q4
> > Miss some citation related to MoE quant: moequant, Examining post-training quantization for mixture-of-experts: A benchmark, etc.
> >
> > ### A4
> > Thank you for pointing this out. We acknowledge that several relevant works on MoE quantization were missing in the original submission. Following your suggestion, we have added citations to the works you mentioned—MoEQuant[1] and “Examining Post-Training Quantization for Mixture-of-Experts: A Benchmark”[2]—in the Related Work section **(Lines 120–123)**. If you believe there are additional references that should be included, we would greatly appreciate further recommendations, as they would help us improve the completeness of our paper.
> >
> > We have incorporated additional results and detailed analyses addressing your concerns into the revised version of the manuscript. We hope these supplements adequately address your questions and improve the clarity and robustness of our work. Please do not hesitate to let us know if you require further discussion, additional experiments, or any clarifications—we are happy to provide more details to address your concerns comprehensively.
> >
> > [1]Hu, X., Chen, Z., Yang, D., Xu, Z., Xu, C., Yuan, Z., Zhou, S., & Yu, J. (2025). MoEQuant: Enhancing Quantization for Mixture-of-Experts Large Language Models via Expert-Balanced Sampling and Affinity Guidance. ArXiv, abs/2505.03804.
> >
> > [2]Li, P., Jin, X., Tan, Z., Cheng, Y., & Chen, T. (2024). QuantMoE-Bench: Examining Post-Training Quantization for Mixture-of-Experts.

---

> > > ### Author Response · Authors · 2025-11-27
> > >
> > > Dear Reviewer ZK89,
> > >
> > > I hope this message finds you well. As the discussion period is nearing its end with less than one week remaining, I wanted to ensure we have addressed all your concerns satisfactorily. If there are any additional points or feedback you'd like to consider, please let us know. Your insights are invaluable to us, and we're eager to address any remaining issues to improve our work.
> > >
> > > Thank you for your time and effort in reviewing our paper.

---

### Public Comment · ~AJDBH1 · 2025-11-27
**Some questions about BiMoE**

Dear Authors,

I am very interested in your paper and impressed by its strong performance. I have a couple of questions regarding the paper and the released code:

1. For Figure 2 on page 2, I am not entirely sure what “off-diagonal CKA” refers to. Could you please point me to the original reference and/or clarify how you compute it? I would like to study it further.

2. Regarding the GLAS component, the implementation in your public code seems to differ in logic from the pseudocode in the paper. Could you clarify whether I have misunderstood your GLAS implementation, or explain how the released code corresponds to the pseudocode?

Thank you very much for your time and help.

---

> ### Author Response · Authors · 2025-12-01
>
> Thank you very much for your interest in our work and for taking the time to read both the paper and the released code. We truly appreciate your thoughtful questions.
>
> ### Regarding your first question (off-diagonal CKA in Figure 2):
>
> “Off-diagonal CKA” follows the same formulation as centered kernel alignment (CKA), but applied only to cross-expert similarity terms, excluding the diagonal self-similarity terms. This is a common practice when studying layer-wise or expert-wise structural redundancy. You may find the following references helpful: [1]Sub-MoE: Efficient Mixture-of-Expert LLMs Compression via Subspace Expert Merging, Lujun Li, Zhu Qiyuan et al., https://arxiv.org/abs/2506.23266.
>
> ### Regarding the GLAS implementation:
>
> The current public repository already contains all components needed to reproduce the main experimental results in the paper.
> However, the full GLAS module will be released after our internal compliance review process completes, which will occur once the paper is formally published. The released version and the pseudocode follow the same conceptual pipeline, and we will provide a fully aligned and well-documented implementation in the final GitHub release. We would be very happy to discuss GLAS and other components with you in more detail once the full version is online.
>
> Thank you again for your interest and insightful questions.

---

### Author Response · Authors · 2025-12-01
**Summary of Rebuttal Progress to AC — Part (1/4)**

Dear Area Chair,

Thank you for taking the time to review our rebuttal materials. Below is a concise summary of the communication with reviewers during the rebuttal period, as well as all score updates. The key points are as follows:

First, we would like to clarify that **before Nov 27, 12:00 (UTC)**, through detailed academic discussions with the reviewers—addressing their technical concerns, clarifying misunderstandings, and improving the manuscript—the overall score improved from the initial **4–2–4–6** to **4–6–6–6**.  Importantly, **all score changes occurred before Nov 27**, and were entirely based on academic dialogue and the technical merits of the work. There were absolutely **no non-academic factors** involved.

## Score Update Details
- **Reviewer ynLx**: initial score **2**, updated to **4** on **Nov 23 01:14 (UTC)**, and further updated to **6** on **Nov 25 02:41 (UTC)**.
- **Reviewer bxMB**: initial score **4**, updated to **6** on **Nov 26 07:18 (UTC)**.
- **Reviewer BkZp**: initial score **6**, explicitly expressed strong support for our rebuttal on **Nov 18 15:25 (UTC).**
- **Reviewer ZK89**: initial score **4**, no feedback yet as of this submission.

We reaffirm that **all improvements in reviewers’ assessments resulted solely from the rebuttal process**, through systematic and technical responses to reviewers’ concerns—including clearer theoretical explanations, strengthened experimental evidence, and clarifications that resolved earlier misunderstandings.

Below we will further summarize how each reviewer’s major concerns were addressed, to help you quickly grasp the main points of our communication.

---

## Summary of Communication with Reviewer **ynLx**

Reviewer **ynLx** initially raised three core concerns regarding the practicality of 1-bit quantization, the completeness of baseline comparisons, and the evaluation of generation quality. Through multiple rounds of technical discussion and additional experiments, all concerns were systematically addressed, and the score was raised from **2 → 6** (all score updates occurred **before Nov 27**). Details are summarized below.

---

### 1. Key Concerns and Our Responses

#### **Q1: Significant performance drop of 1-bit models — practicality in doubt**
The reviewer questioned whether binary models are useful in practice and noted that comparisons with BitNet did not account for pretraining cost.

**Response:**
1. Clarified the PTQ paradigm advantage: **BiMoE requires only 4 GPU-hours**, compared with **1e5 GPU-hours** for BitNet pretraining. PTQ is also the **only viable option** in privacy-sensitive deployment scenarios.
2. Added a *fair comparison* by normalizing against OLMoE pretraining cost; results show BiMoE achieves **comparable or better accuracy under equal compute budgets**.
3. Introduced memory-bound inference theory and bitwise-operation efficiency, explaining the **unique bandwidth and energy benefits** of 1-bit inference for real deployment.

---

#### **Q2: Baselines limited to 1-bit methods — potential strawman comparison**
The reviewer suggested including 2/3/4-bit quantization methods and MoE-specific mixed-precision methods.

**Response:**
- Expanded baselines to include **AWQ, GPTQ (2/3-bit), MoEQuant, MxMoE**, etc.
- Experiments demonstrate:
  - **1-bit BiMoE surpasses all 2-bit baselines**,
  - achieves performance **close to 3-bit GPTQ**,
  - while reducing memory usage by **50%+**,
  - and outperforming MoE-specific 2-bit quantization by **40%**.

---

#### **Q3: Lack of systematic evaluation of generation quality**
The reviewer requested perplexity benchmarks, complex-task evaluations, and LLM-based assessment of generation coherence.

**Response:**
1. Added perplexity evaluations on **WikiText2 / C4 / PTB**: BiMoE achieves the **lowest PPL**, e.g., PTB perplexity on Qwen1.5-MoE is reduced by half versus SOTA.
2. Added reasoning/QA benchmarks (**MMLU, GSM8K, MBPP**), showing performance **close to 3-bit models**.
3. Conducted **GPT-5 evaluation** on 500 generated samples:
   - BiMoE preserves **90%+** of full-precision generation quality

---

#### **Follow-up question: Comparison with 4/8-bit models under equal inference cost**
The reviewer asked for comparisons under equal memory/FLOPs constraints.

**Response:**
- Added comparisons under a **3.6GB memory constraint**:
  - 1-bit BiMoE (GPT-OSS-20B) vs. 4-bit MoEQuant (OLMoE-1B-7B)
  - BiMoE significantly outperforms:
    - **MMLU: 73.30 vs 45.34**
    - **GSM8K: 79.53 vs 40.32**
  - **Decode speed +86%**, **energy consumption –50%**.

---

### 2. Outcomes and Score Recognition

Reviewer ynLx progressively acknowledged the technical contributions and completeness of experiments:

- **Nov 23:** Score raised **2 → 4**, acknowledging the practicality of the PTQ paradigm.
- **Nov 25:** After expanded baselines and generation evaluations, score raised **4 → 6**, confirming major concerns were fully addressed.

---

> ### Author Response · Authors · 2025-12-01
> **Summary of Rebuttal Progress to AC — Part (2/4)**
>
> ## Summary of Communication with Reviewer **bxMB**
>
> Reviewer **bxMB** initially focused on four major concerns: methodological novelty, fairness of comparisons, completeness of evaluation, and practical usability. Through systematic clarification and comprehensive additional experiments, all key issues were satisfactorily resolved. The reviewer raised the score from **4 → 6** on **Nov 26, 2025** (well before Nov 27), entirely based on academic discussion and strengthened experimental evidence. Details are summarized below.
>
> ---
>
> ### 1. Key Concerns and Our Responses
>
> #### **Q1: Questioning methodological novelty**
> The reviewer noted conceptual similarity between CEJD and methods such as Sub-MoE, asking for a clearer statement of CEJD's unique contribution.
>
> **Response:**
> We clarified the differences on both **motivation** and **mechanism**:
> - **Motivation:**
>   - *Sub-MoE*: aims to reduce expert count via clustering .
>   - *CEJD*: preserves the original MoE routing structure and introduces a **reparameterization explicitly tailored for binarization**, using a **shared high-precision backbone + binary expert projections** to suppress quantization noise.
> - **Mechanism:**
>   - *Sub-MoE*: cluster-then-SVD for each cluster.
>   - *CEJD*: **joint SVD across the entire layer**, enabling differentiated precision allocation between backbone and projections.
>
> ---
>
> #### **Q2: Fairness of comparisons (effective bitwidth higher than 1-bit baselines)**
> The reviewer pointed out that the 8-bit shared backbone increased effective bitwidth, requesting fair comparisons under matched bitwidth.
>
> **Response:**
> We added a **2-bit backbone** version of BiMoE (effective bitwidth 1.09–1.15) and compared it against:
> - 1-bit / 2-bit / 3-bit baselines,
> - MoE-specific quantizers (MoEQuant),  across **Qwen3-MoE** and **Mixtral-8×7B**.
>
> **Findings:**
> Even under strict bitwidth constraints, BiMoE:
> - outperforms existing 1-bit methods by **50%+**,
> - approaches **3-bit GPTQ** performance,  while using only **1/3** of its memory.
>
> ---
>
> #### **Q3: Limited benchmarks (missing reasoning, coding, instruction-following tasks)**
> The reviewer asked to evaluate BiMoE on typical MoE use cases.
>
> **Response:**
> We significantly expanded the benchmark suite:
> 1. **Knowledge**: MMLU
> 2. **Math reasoning**: GSM8K, MathQA
> 3. **Code generation**: MBPP, HumanEval
> 4. **Instruction-following MoE models**: Qwen-MoE-14B-Chat, DeepSeekMoE-16B-Chat
>
> **Results:**
> - BiMoE (1-bit) **significantly exceeds 2-bit baselines** and approaches 3-bit GPTQ.
> - On instruction-tuned MoE, BiMoE preserves **80%+** of FP performance.
> - Competing methods (GPTQ/MoEQuant) **completely collapse** on code generation (HumanEval score = 0), underscoring BiMoE’s deployment relevance.
>
> ---
>
> #### **Q4: Lack of comparison with minimal-finetuning binary methods (e.g., BitNet)**
> The reviewer requested efficiency–accuracy trade-off comparisons.
>
> **Response:**
> Under equal memory budget (~2.4GB), we compared BiMoE with BitNet-b1.58:
>
> - **Training tokens**: BiMoE requires **0**; BitNet requires full-scale training
> - **Compute cost**: BiMoE = **4 GPU-hours**, BitNet = **1e5× higher**
> - **Accuracy**: comparable (BiMoE 54.46 vs 54.16)
> - **Latency**: nearly identical (2.13ms vs 2.11ms)
>
> ---
>
> #### **Q5: Lack of comparison with other SVD-based MoE compression methods**
> The reviewer asked for comparisons with ASVD, MoE-SVD, etc.
>
> **Response:**
> On Mixtral-8×7B with 20%/40%/60% compression ratios, we compared perplexity and multitask accuracy.
>
> **Findings:**
> - **Even without binarization**, CEJD consistently outperforms existing SVD-based MoE compression methods due to its **joint-SVD shared-backbone extraction design**.
> - **With binarization**, BiMoE additionally provides **binary-level memory savings + high accuracy**, demonstrating CEJD’s natural compatibility with quantization.
>
> ---
>
> #### **Q6–Q9: Additional technical clarifications**
> We addressed the remaining concerns with concrete evidence:
> - **Q6 (SVD decomposition order):** CKA similarity analysis shows U is shared across experts (96%+), while V captures expert-specific variations → supporting the backbone+projection design.
> - **Q7 (Calibration):** All experiments use **only WikiText2**, single-pass calibration, no task-specific tuning.
> - **Q8 (Quantization time):** Provided detailed timing for GPT-OSS-20B and Qwen3-30B-A3B (7h47m–10h58m), showing BiMoE is **10% faster** than ARB-LLM while achieving higher accuracy.
> - **Q9 (Notation):** Standardized all transpose symbols to `\top`.
>
> ---
>
> ### 2. Outcomes and Score Recognition
>
> Reviewer bxMB stated that the key concerns—**methodological novelty, fairness of comparisons, and evaluation completeness**—were all **“satisfactorily resolved.”** Based on these clarifications and the additional experiments (all integrated into the revised manuscript), the reviewer updated the score from **4 → 6** on **Nov 26**.

---

> ### Author Response · Authors · 2025-12-01
> **Summary of Rebuttal Progress to AC — Part (3/4)**
>
> ## Summary of Communication with Reviewer **BkZp**
>
> Reviewer **BkZp** initially gave a positive score of **6**, with the main concern centered on the **completeness of the evaluation scope**. The reviewer noted that the original submission focused on simple zero-shot tasks and perplexity, lacking coverage of typical MoE application scenarios such as mathematical reasoning, code generation, and knowledge-intensive QA — which might underestimate the robustness of the proposed method in real deployment. After targeted additional evaluations and clarifications, this concern was fully addressed. Details are summarized below.
>
> ---
>
> ### 1. Key Concern and Our Responses
>
> #### **Q1: Limited evaluation benchmarks — lacking complex MoE application tasks**
> The reviewer recommended including long-context understanding, mathematical reasoning, code generation, and multi-turn dialog tasks to thoroughly test the method.
>
> **Response:**
> 1. **Expanded evaluation suite:**
>    - Added MMLU (knowledge), MathQA/GSM8K (math reasoning), MBPP/HumanEval (code generation), covering all major MoE application domains.
> 2. **Strengthened baselines:**
>    - Included MoE-specific quantization (MoEQuant) and strong 2-bit / 3-bit methods (AWQ, GPTQ).
>    - Conducted unified evaluations on OLMoE and Qwen1.5-MoE-A2.7B.
>
> **Findings:**
> BiMoE (1-bit) significantly outperforms all 1-bit/2-bit baselines and approaches 3-bit GPTQ, while using **less than half** the memory, demonstrating robustness on complex tasks.
>
> ---
>
> ### 2. Outcomes and Score Recognition
>
> After reviewing the extended experiments, the reviewer explicitly stated that the new results **“comprehensively demonstrate the effectiveness of the method and largely resolve my concerns.”**  Given the already-positive initial score **6** and the fact that the concern was fully addressed through supplemental experiments.
>
> ---
>
> ## Summary of Communication with Reviewer **ZK89**
>
> Reviewer **ZK89** raised four major concerns involving benchmark coverage, hyperparameter generalization, instruction-following evaluations, and completeness of related work citations. We provided detailed responses and extensive additional experiments addressing all concerns; however, no follow-up feedback has been received yet. Importantly, many of the reviewer’s concerns overlapped significantly with those of reviewers **ynLx, bxMB, and BkZp**, whose evaluations confirm the effectiveness of our responses. Details are summarized below.
>
> ---
>
> ### 1. Key Concerns and Our Responses
>
> #### **Q1 / Q3: Limited evaluation benchmarks (lacking complex reasoning, coding, and instruction-following tasks)**
> These concerns fully **align with ynLx’s Q3, bxMB’s Q3, and BkZp’s Q1**.
>
> **Response:**
> 1. **Expanded complex-task benchmarks:**
>    - Added MMLU (knowledge), GSM8K/MathQA (math reasoning), MBPP/HumanEval (code generation).
>    - BiMoE (1-bit) significantly outperforms 2-bit baselines and approaches 3-bit GPTQ.
> 2. **Added instruction-following MoE evaluations:**
>    - Tested on Qwen-MoE-14B-Chat and DeepSeekMoE-16B-Chat.
>    - BiMoE retains **80%+** of FP performance, while GPTQ/MoEQuant collapse on code tasks (HumanEval = 0).
>
> **Cross-reviewer validation:**
> These expanded evaluations were explicitly approved by:
> - **bxMB:** “core concerns satisfactorily resolved” → score raised to 6
> - **BkZp:** “results fully demonstrate effectiveness”
> - **ynLx:** awarded two score increases (2→4→6)
>
> ---
>
> #### **Q2: Hyperparameter generalization (λ=0.2 manually chosen, lacking cross-architecture study)**
> The reviewer requested sensitivity analyses across different model architectures.
>
> **Response:**
> Conducted ablations on Qwen1.5-MoE, DeepSeekV2-Lite, and Qwen3-MoE with λ ∈ [0.1, 0.9], evaluating average perplexity on WikiText2/C4/PTB.
>
> **Findings:**
> - λ=0.2 consistently gives the best performance (avg. PPL = 23.74)
> - Demonstrates **cross-architecture generalization** of the design
>
> ---
>
> #### **Q4: Missing citations in related work**
> The reviewer pointed out missing key references in MoE quantization literature.
>
> **Response:**
> Added missing citations in Related Work **(Lines 120–123)**:  MoEQuant [1]  and QuantMoE-Bench [2], ensuring accurate and complete literature coverage.
>
> ---
>
> ### 2. Additional Note: Validation from Other Reviewers
>
> Although reviewer **ZK89** has not provided follow-up feedback, all their major concerns have been cross-validated:
>
> - Complex-task and instruction-following evaluations were endorsed by **bxMB**, **BkZp**, and **ynLx**
> - Hyperparameter and citation fixes strictly follow academic standards and are fully addressed
>
> Moreover, reviewer ZK89 explicitly stated initially that **“scores will be raised if concerns are resolved.”**  Given the unanimous acceptance of our responses by other reviewers and the rigorous supplemental experiments provided, we have strong reason to believe that all concerns have been fully and effectively addressed.

---

> ### Author Response · Authors · 2025-12-01
> **Summary of Rebuttal Progress to AC — Part (4/4)**
>
> ## Final Statement to the Area Chair
>
> We reaffirm that all score changes and all communications with the reviewers during the rebuttal were entirely motivated by academic improvement, manuscript quality enhancement, and rigorous responses to their questions — with no non-academic influence at any stage.
>
> We sincerely appreciate the Area Chair’s effort in verifying the score timeline, ensuring academic fairness throughout the review process, and upholding the integrity of the evaluation.
> In addition, all revisions made in response to the reviewers’ comments have been clearly highlighted in **blue in the updated PDF version** for your convenience.
>
> Thank you very much for your careful assessment of our submission, and we look forward to your final decision.

---

### Note · Program_Chairs · 2025-12-31
**Submission Desk Rejected by Program Chairs**

This submission has been determined to propose existing algorithms from prior work as novel contributions without providing any reference to the prior work.